# Arrayed CRISPR libraries for the genome-wide activation, deletion and silencing of human protein-coding genes

Arrayed CRISPR libraries extend the scope of gene-perturbation screens to non-selectable cell phenotypes. However, library generation requires assembling thousands of vectors expressing single-guide RNAs (sgRNAs). Here, by leveraging massively parallel plasmid-cloning methodology, we show that arrayed libraries can be constructed for the genome-wide ablation (19,936 plasmids) of human protein-coding genes and for their activation and epigenetic silencing (22,442 plasmids), with each plasmid encoding an array of four non-overlapping sgRNAs designed to tolerate most human DNA polymorphisms. The quadruple-sgRNA libraries yielded high perturbation efficacies in deletion (75–99%) and silencing (76–92%) experiments and substantial fold changes in activation experiments. Moreover, an arrayed activation screen of 1,634 human transcription factors uncovered 11 novel regulators of the cellular prion protein PrP$^C$, screening with a pooled version of the ablation library led to the identification of 5 novel modifiers of autophagy that otherwise went undetected, and 'post-pooling' individually produced lentiviruses eliminated template-switching artefacts and enhanced the performance of pooled screens for epigenetic silencing. Quadruple-sgRNA arrayed libraries are a powerful and versatile resource for targeted genome-wide perturbations.

Genetic screens are well-established tools of biomedical research[1,2] and have been substantially expanded by CRISPR-mediated techniques, which now allow for gene ablation (CRISPRko), activation (CRISPRa), interference (CRISPRi) and epigenetic silencing (CRISPRoff)[3,4]. Most genome-wide CRISPR screens use pooled libraries based on single-guide-RNA (sgRNA) vectors, where all vectors are mixed in 1–2 pools. These libraries have enabled the study of cell-autonomous phenotypes, including cell survival, gene expression, drug resistance and many other phenotypes selectable by drugs or by phenotype[3,4]. However, many important characteristics of cell biology, including interactions through which certain cells cause other cells to exhibit an induced trait, or the secretion of bioactive molecules, cannot be addressed using pooled CRISPR libraries. Furthermore, pooled libraries perform suboptimally in genome-wide high-content optical screens[3,5–7].

Arrayed sgRNA libraries, which target genes one by one in distinct wells, are applicable to almost all screenable phenotypes[8], but their construction is laborious. There are only few commercial and academic[9–11] arrayed CRISPRko libraries with poorly documented effectiveness, and no genome-wide arrayed CRISPRa library is available, precluding the study of phenotypes through activation screens. Furthermore, existing arrayed libraries suffer from several limitations. First, synthetic crRNA libraries are limited to easily transfectable cells and do not allow for the selection of transfected cells. Second, plasmid-based libraries featuring one sgRNA per vector exhibit low and heterogeneous gene-perturbation efficiency[12]. Third, sgRNAs driven by a single promoter may not be effective in the cells of interest. Fourth, the sgRNA design algorithms used by most existing libraries are based on the hg38 or earlier versions of human reference genomes[13,14].

✉e-mail: jiang-an.yin@uzh.ch; adriano.aguzzi@uzh.ch

However, the genome of cells used for gene-perturbation screens, such as patient-derived induced pluripotent stem cells (iPSCs)[15], may diverge from the reference genome, leading to impaired sgRNA function.

In this Article, we describe a new generation of highly active, versatile CRISPR arrayed libraries that overcome the limitations of existing libraries and allow for multiple genetic and epigenetic gain/loss-of-function modalities, thereby enabling the investigation of hitherto unexplored biological space. We confirm that targeting each gene with multiple sgRNAs improves the potency of gene perturbation[9,16,17], and show experimentally that optimal gene perturbation[14] is achieved with quadruple-guide RNAs (qgRNAs).

Because traditional DNA cloning[17] requires gel purification, colony picking and Sanger sequencing, it is unsuitable for the high-throughput generation of arrayed qgRNA plasmids. We therefore developed ALPA (for 'automated liquid-phase assembly') cloning, which assembles four sgRNAs targeting the same gene, driven by four distinct promoters, into a single vector in a high-throughput manner cost-effectively. We devised a custom algorithm to design non-overlapping sgRNAs tolerant to most common genetic polymorphisms. With these methods, we constructed a deletion and a combined activation/silencing library (termed T.spiezzo and T.gonfio) consisting of 19,936 and 22,442 plasmids and targeting 19,820 and 19,839 human protein-coding genes, respectively. We showcase the practical utility of T.spiezzo and T.gonfio by performing large-scale gene activation, silencing and ablation screens in a broad spectrum of use cases, thereby identifying a wealth of genetic modifiers that had gone undetected with existing tools.

## Results

### The ALPA cloning method

Conventional DNA cloning yields heterogeneous recombination products, requiring the isolation and verification of clonal bacterial colonies streaked on semisolid culture media. These steps are not easily automatable and preclude the simultaneous generation of large numbers of plasmids. Here we describe the ALPA cloning procedure, which allows for one-pot plasmid assembly and bacterial transformation. Thanks to a dual antibiotic selection in the precursor vector (ampicillin) and the final plasmid (trimethoprim), ALPA cloning selectively enriches the desired plasmids to levels comparable to single-colony picking.

We used ALPA cloning to assemble four sgRNAs, each followed by a distinct variant of trans-activating CRISPR RNA (tracrRNA) and driven by a different ubiquitously active type III RNA polymerase promoter (human U6, mouse U6, human H1 and human 7SK), into a single vector[14,18,19] (Fig. 1a and Extended Data Fig. 1). The qgRNA vector includes puromycin and TagBFP cassettes flanked by lentiviral long terminal repeats and PiggyBac (PB) transposon elements, enabling multiple routes of selection and transduction[11] (Fig. 1a). The four sgRNAs were individually synthesized as 59-meric oligonucleotide primers comprising the 20-nucleotide protospacer sequence and a constant region, including amplification primer annealing sites (Extended Data Fig. 1). In three distinct polymerase chain reactions (PCRs), the primers were mixed with the corresponding constant-fragment templates to produce three individual amplicons. These amplicons and the digested empty vector (pYJA5) contain directionally distinct overlapping ends (approximately 20 nucleotides), enabling Gibson assembly[20] (Fig. 1a and Extended Data Fig. 1).

In the precursor vector pYJA5, the β-lactamase gene (AmpR) providing ampicillin resistance flanked by two *Bbs*I restriction sites was removed to minimize the size of final qgRNA plasmids for the subsequent Gibson assembly steps. A trimethoprim-resistant dihydrofolate reductase gene (TmpR) was incorporated into the first amplicon between sgRNA1 and sgRNA2, enabling a selection switch from ampicillin to trimethoprim[17,19] (Fig. 1a and Extended Data Fig. 1).

To test the accuracy of ALPA cloning, we cloned the sgRNA-containing amplicons into pYJA5 using Gibson assembly (Fig. 1b) and performed PCR on single colonies with primers flanking the qgRNA insert (Fig. 1a). All PCR products from the tested colonies showed the expected size of 2.2 kb, suggesting the correct assembly of the backbone and fragments (Fig. 1b). We then sequenced single colonies from eight independent cloning procedures (≥22 colonies per procedure). All colonies showed the desired antibiotic selection switch, and each procedure resulted in 83–93% of colonies showing correct qgRNA sequences (Fig. 1c). Repetitive DNA sequences may lead to inappropriate recombination. To minimize this effect, each of the four sgRNAs is driven by a different Pol-III promoter and followed by a distinct tracrRNA. In the eight cloning trials described above, 0–10% of tested colonies harboured recombined plasmids (Fig. 1c). The frequency of plasmids bearing point mutations in the cloning trials was 3–14% (Fig. 1c). Such mutations may result from errors in oligonucleotide synthesis or from DNA mismatch tolerance by the Taq DNA ligase during Gibson assembly[20,21]. Four of the eight cloning trials were repeated three times; the percentages of colonies with correct, recombined or mutated plasmids were similar to the previous trials (Fig. 1d). Hence, ALPA cloning generates high-quality plasmids without requiring the isolation of single bacterial colonies.

For high-throughput cloning, ALPA cloning steps were performed in 384-well plates. Reaction products were transferred to deep-96-well plates for transformation and amplification in recombination-deficient chemically competent *E. coli*. Magnetic bead-based plasmid minipreps were performed in the same microplates using custom-made equipment (Methods), enabling the construction of >42,000 individual plasmids (~2,000 plasmids per week with two full-time equivalents) with a yield of ~25 µg per plasmid (Fig. 1e).

### Efficiency and robustness of qgRNAs in gene activation and ablation

We tested the efficiency of ALPA-cloned qgRNA plasmids for gene activation and ablation in HEK293 (human embryotic kidney) cells. For CRISPRa, we co-transfected the CRISPR activator dCas9-VPR with sgRNA-expressing plasmids targeting the genes *ASCL1*, *NEUROD1* and *CXCR4*, which have low, moderate and high baseline expression, respectively[16]. Compared with individual sgRNAs cloned into a pYJA5-derived vector modified for one sgRNA insertion, qgRNA vectors massively increased target gene activation (Fig. 1f), consistent with the synergistic gene activation efficiency observed with three sgRNAs[16].

To explore the universality of the enhanced gene activation observed with qgRNAs, we conducted an unbiased examination of 12 model protein-coding genes (*IL1R2*, *IL1B*, *MYC*, *KLF4*, *NANOG*, *ZFP42*, *HBG1*, *POU5F1*, *LIN28A*, *TERT*, *SOX2* and *VEGFA*) used in an earlier study[22]. Many qgRNAs exhibited superior activation, and none was inferior to the best-performing sgRNAs (Fig. 1f and Extended Data Fig. 2a). The extent of gene activation was highest for genes with low basal expression levels (Extended Data Fig. 2b). Moreover, qgRNAs could efficiently activate long non-coding RNAs (*TINCR*, *LINC00925*, *LINC00514* and *LINC00028*)[22] (Extended Data Fig. 2c). Hence, the qgRNA strategy achieves robust gene activation of a wide variety of protein-coding and non-coding genes.

Furthermore, we used the cell-surface proteins CD2, CD4 and CD200 in HEK293 cells to test if the qgRNA design might reduce the considerable cell-to-cell heterogeneity afflicting gene-activating single sgRNAs[23]. Single sgRNAs induced variable, mostly low gene activation, whereas qgRNA transduction into HEK293 cells stably expressing doxycycline-inducible dCas9-VPR led to robust cell-surface expression of CD2, CD4 and CD200, with improved separation of activated cells from the non-targeting (NT) controls, as reflected by their superior *Z*′ factors (Extended Data Fig. 2d,e).

We assessed the CRISPRko efficacy by live-cell antibody staining of the cell-surface molecules CD47, IFNGR1 and MCAM[24]. For each gene, 12 single sgRNAs from widely used resources[14,25,26] were tested. Single sgRNAs showed variable ablation efficiencies (5–85% for *CD47*, 1–76% for *IFNGR1* and 6–85% for *MCAM*). By contrast, the respective qgRNA

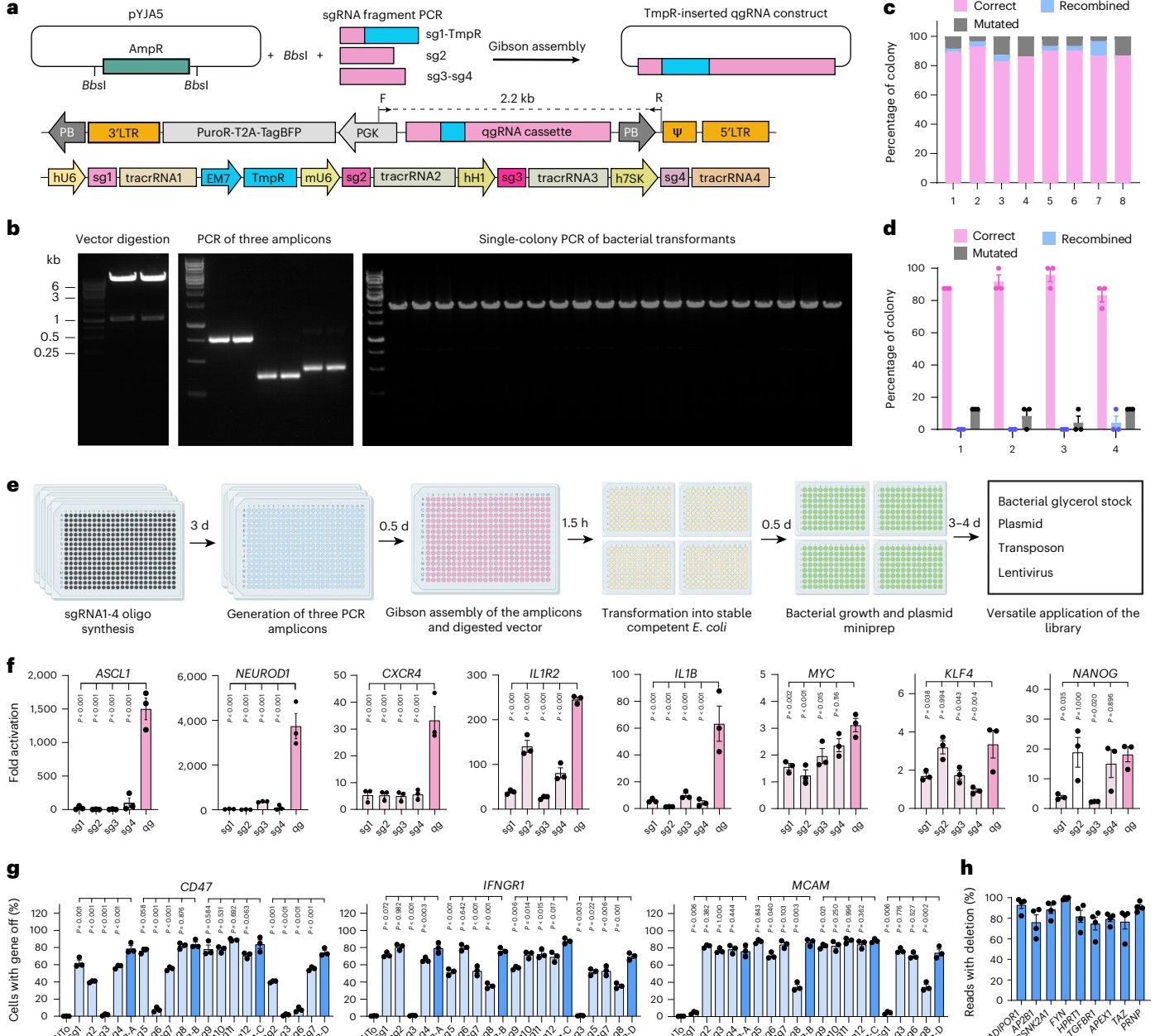

**Fig. 1 | ALPA cloning and gene activation and ablation with qgRNA. a**, Cloning strategy. The ampicillin resistance gene (AmpR) was removed from the vector pYJA5. sgRNA1–4 and the trimethoprim resistance gene (TmpR) were fused with three distinct PCR amplicons. All elements were Gibson-assembled to form the qgRNA-pYJA5 plasmid, and transformants were selected with trimethoprim. The detailed structure of qgRNA-pYJA5 full plasmid and qgRNA cassette are depicted. LTR, long terminal repeat; Ψ, packaging signal sequence; PB, piggyBac transposon element; PuroR, puromycin resistance element; hU6, mU6, hH1 and h7SK are ubiquitously expressed RNA polymerase-III promoters; sg, sgRNA. F and R arrows: forward and reverse primers used for single-colony PCRs, Sanger and NGS. **b**, Representative pYJA5 restriction fragments, 3-fragment PCRs and single-colony PCR of ALPA cloning products after transforming into *E. coli* and trimethoprim selection. *Bbs*I digestion of pYJA5 yielded an ~1 kb band of the AmpR element and ~7.6 kb band of the linearized vector (left). After PCR with the corresponding sgRNA primers, the three amplicons showed the expected size of 761 bp, 360 bp and 422 bp on agarose gels, respectively (middle). Single-colony PCR with primers flanking the qgRNA cassette of ALPA cloning products in transformed bacteria plate consistently yielded the expected size (2.2 kb, right). **c**, Percentage of correct, recombined and mutated qgRNA plasmids in 8 independent ALPA cloning experiments with distinct qgRNA sequences

(≥22 colonies were tested in each experiment). **d**, Percentage of correct, recombined and mutated qgRNA plasmids in four ALPA cloning experiments. Each dot represents an independent biological replicate consisting of eight colonies (*n* = 24; mean ± s.e.m.). **e**, Timeline of ALPA cloning in high-throughput format (h, hours; d, days). Created with BioRender.com. **f**, Gene activation (qRT-PCR) in HEK293 cells 3 days post-transfection with dCas9-VPR and single (sg1–4) or four sgRNA (qg) plasmids. Additional genes are shown in Extended Data Fig. 2a. Dots (here and henceforth): independent experiments (mean ± s.e.m.). **g**, Gene ablation efficiency by single sgRNAs versus qgRNAs in HEK293 cells via co-transfection with the Cas9 plasmid. 12 single sgRNAs (sg1–12) from the Brunello, GeCKOv2 and TKOv3 libraries were tested; qgRNA plasmids (qg-A,B,C,D) were assembled with the random combination of sg1–4, sg5–8 and sg9–12, and the 4 least effective single sgRNAs among the 12 sgRNAs, respectively. Outcomes were re-plotted for the four least effective single sgRNAs along with the respective qg-D. hNTo, NT control plasmid; cell-surface proteins were stained with fluorescent-conjugated antibodies and analysed via flow cytometry. **h**, qgRNA plasmids robustly ablated genes inadequately disrupted by single sgRNAs. Single sgRNAs were assembled into qgRNA plasmids and co-transfected with the Cas9 plasmid into HEK293 cells (as in **g**). In **f** and **g**, *P* values were determined by one-way ANOVA with Dunnett's multiple comparisons test.

plasmids showed ablation efficiencies of >70% (Fig. 1g and Supplementary Fig. 1a). Even the combination of the four least effective single sgRNAs into qgRNAs resulted in robust gene ablation (qg-D in Fig. 1g). To further test the efficiency and robustness of the qgRNA approach for gene ablation, we examined 9 genes (*ADIPOR1*, *AP2B1*, *CSNK2A1*, *FYN*, *HPRT1*, *TGFBR1*, *APEX1*, *TAZ* and *PRNP*) for which single sgRNAs have low or moderate ablation efficiency[27]. Single-molecule real-time (SMRT) long-read sequencing of PCR amplicons from genome-edited cells showed nucleotide deletions in 75–99% of the sequencing reads (Fig. 1h). Agarose gel electrophoresis showed conspicuous deletions of the genomic region between the sgRNA cutting sites for all nine genes (Extended Data Fig. 2f).

Gene ablation with qgRNA plasmids may induce multiple DNA double-strand breaks, resulting in toxicity. We tested this by transducing equal numbers of dTomato-expressing wild-type and enhanced green fluorescent protein (EGFP)-Cas9-expressing HEK293 cells (Extended Data Fig. 2g) with NT qgRNAs (hNTo-1 and hNTo-2) or qgRNAs against essential (*CHERP* and *RPL37*) or non-essential (*FBLN1*, *SLC9A9*, *FAM174A*, *AK5*, *CLCN5*, *SFXN3*, *GSTM1* and *SPRR4*) genes (Extended Data Fig. 2h). After 2 weeks in culture, transduction with qgRNAs targeting essential genes led to the profound depletion of EGFP. However, no EGFP depletion was observed in cells transduced with NT qgRNAs or with qgRNA vectors targeting non-essential genes (Extended Data Fig. 2h). We conclude that qgRNA ablation plasmids do not impose detectable toxicity onto genome-edited HEK293 cells.

### Lentiviral packaging and delivery of the qgRNA vectors

We devised a lentiviral production protocol in 384-well plates in HEK293T cells (Extended Data Fig. 2i and Supplementary Information), enabling the conversion of 2,000–4,000 qgRNA plasmids per week into lentiviruses. This method achieved ≥50% cell transfection efficiency, a median titre of $5.5 \times 10^6$ lentiviral transducing units (TU) per ml in raw culture-medium supernatants, and 81.5% of plates exhibiting titres exceeding $1 \times 10^6$ TU ml$^{-1}$ in ≥90% of wells on each plate (Extended Data Fig. 2j). Viral transduction efficiently delivered qgRNAs to the human lymphocyte-related cell lines THP-1 and ARH-77, the human neuroblastoma cell line GIMEN, the human glioblastoma cell line U251-MG and patient-derived iPSCs, as indicated by the fraction of TagBFP$^+$ cells (Extended Data Fig. 2k). We then examined the efficiency of gene activation in iPSC-derived neurons (iNeurons, which stably express dCas9-VPR) using lentivirus-mediated delivery of the qgRNA vector. We observed conspicuous gene activation of all tested genes ranging from ~2-fold to ~10,000-fold (Extended Data Fig. 2l).

### Updated algorithms for generic, specific and synergistic sgRNA selection

To enable gain-of-function and loss-of-function arrayed CRISPR screens, we generated genome-wide activation ('T.gonfio', meaning swelling up) and deletion ('T.spiezzo', meaning breaking in two) arrayed libraries for human protein-coding genes using ALPA cloning (Fig. 1e). We used the Calabrese and hCRISPRa-v2 sgRNA sequences[14,28] for T.gonfio, and the Brunello and TKOv3 sgRNA sequences[14,26] for T.spiezzo as a baseline to generate our arrayed libraries with an algorithm to select the optimal combination of four sgRNAs (Fig. 2a and Source Data File 1).

Common DNA polymorphisms in human genomes, such as single-nucleotide polymorphisms (SNPs), affect 0.1% of the genome and may reduce CRISPR efficacy[15]. Except for the TKOv3 library, most CRISPR libraries did not consider DNA polymorphisms when selecting sgRNA sequences, hampering their usage on primary patient-derived cells. We obtained a dataset derived from 13,200 whole genomes and 64,600 exomes from a total of 77,781 individuals (http://db.systemsbiology.net/kaviar/) providing the coordinates of genetic polymorphisms aligned to the hg38 reference genome. Guide RNAs were flagged as unsuitable if the genomic coordinates of the 20-nucleotide protospacer sequence or the 2 guanine nucleotides

of the protospacer adjacent motif (PAM) were affected by a polymorphism with a frequency higher than 0.1%. The GuideScan algorithm (http://www.guidescan.com) can predict off-target effects of sgRNAs with high accuracy, showing a strong correlation with the unbiased genome-wide off-target assay GUIDE-Seq[29,30]. Guide RNAs with GuideScan scores exceeding 0.2 are generally considered specific[30], and we imposed this constraint for sgRNA selection in our libraries.

Previous libraries chose top-ranking sgRNA sequences based on high on-target efficacy scores and low predicted off-target effects, yet this could result in the selection of overlapping sgRNAs whose target positions differed only by a few nucleotides. This was particularly common in CRISPRa libraries, owing to the limited target window for sgRNAs upstream of the transcription start site (TSS). We found that four non-overlapping sgRNAs (spaced ≥50 nucleotides apart) resulted in conspicuously higher gene activation than qgRNA combinations that did not meet this criterion, suggesting that spatially unconstrained binding of sgRNA-dCas9-VPR complexes is strongly synergistic (Extended Data Fig. 3a). Since we generated our libraries using the Gibson assembly method, if two or more sgRNAs share identical subsequences of 8 nucleotides or more, the prevalence of correct plasmids decreased because of recombination between identical sequences among the four sgRNAs (Extended Data Fig. 3b,c).

The efficacy of CRISPRa-mediated gene activation relies on sgRNAs targeting a window of 400 base pairs (bp) upstream of TSS of a gene[14,31]. Many genes have more than one TSS that may exhibit different activities in different cell models[14,32]. We designed the T.gonfio library to target each major TSS with an individual qgRNA plasmid, except when TSSs were spaced <1,000 bp apart. Because some genes or TSSs did not allow for four sgRNAs fulfilling these requirements, we supplemented the above-mentioned libraries with sgRNAs from the CRISPick web portal (https://portals.broadinstitute.org/gppx/crispick/public), which designs sgRNAs with the same algorithm as the Calabrese and Brunello libraries. After filtering with the above constraints, all possible combinations of four sgRNAs targeting a gene/TSS were ranked by their aggregate specificity score, enabling the selection of sgRNA sequences with minimized potential off-target effects (Fig. 2a).

### Features of the T.spiezzo and T.gonfio libraries

The T.gonfio and the T.spiezzo libraries include 22,442 plasmids and 19,936 plasmids covering 19,839 and 19,820 human protein-coding genes, respectively. Each library contains 116 NT control plasmids and is organized into thematic sublibraries (Source Data File 1). The transcription factor (TF), secretome and G protein-coupled receptor sublibraries were defined according to current gene catalogues[33,34]. Other sublibraries were based on categories defined by the pooled library hCRISPRa-v2 (ref. 28). The sgRNAs selected with our algorithm originated mostly from previously published libraries (Fig. 2b and Source Data File 1). T.gonfio covers 17,528 genes targeted at a single TSS and 2,311 genes targeted at ≥2 TSSs (Extended Data Fig. 3d). Among the 19,820 genes targeted by the T.spiezzo library, the expected deletions in the human genome range from 19 to >10$^5$ bases (Extended Data Fig. 3d). By excluding sgRNAs with GuideScan scores <0.2, we enriched for specificity without sacrificing the predicted efficacy (Fig. 2c,d). Both T.spiezzo and T.gonfio were designed to improve targeting by avoiding genetically polymorphic regions (Fig. 2e and Extended Data Fig. 3e). Our sgRNA selection algorithm maximized the proportion of sgRNAs spaced ≥50 nucleotides apart (Fig. 2f), thus increasing targeting efficacy. Furthermore, we avoided qgRNA combinations that shared subsequences of ≥8 bp (Extended Data Fig. 3f), thus ensuring minimal chances for sgRNA recombination.

We avoided sgRNAs with multiple perfect genomic matches wherever possible. However, when targeting families of closely related, paralogous genes, there were often no specific sgRNAs to choose from. For simplicity, we created a separate qgRNA plasmid for each protein-coding gene that possessed its own unique Entrez gene

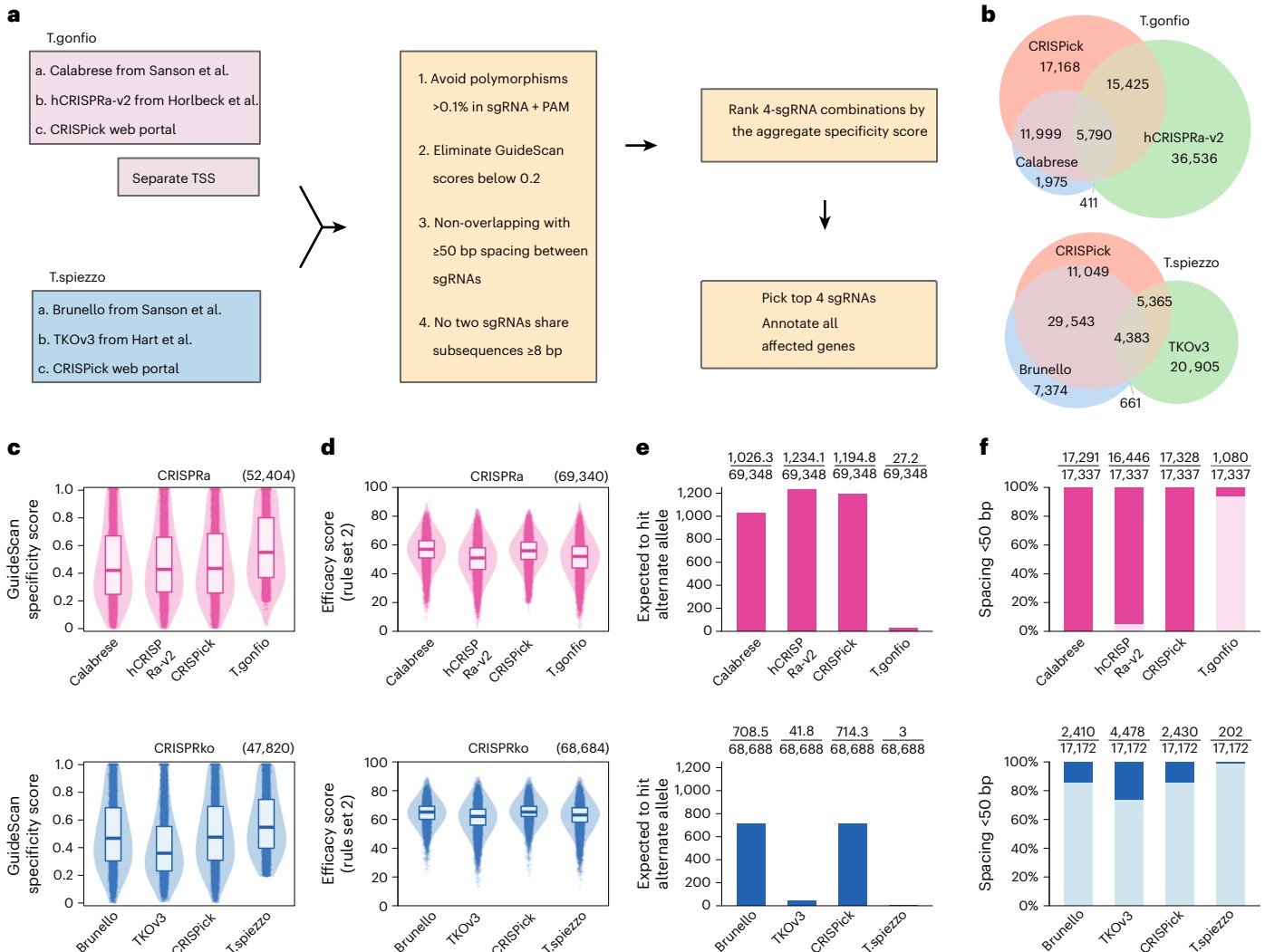

**Fig. 2 | sgRNA selection and features of the T.spiezzo and T.gonfio libraries.** **a**, A large pool of potential sgRNAs was explored by combining existing CRISPR libraries and the output of the CRISPick system. Each sgRNA was annotated with data on genetic polymorphisms affecting the target region and with GuideScan specificity scores. For T.gonfio, individual qgRNA plasmids were designed to target each alternative TSS. The best combination of four non-overlapping sgRNAs was chosen (Source Data File 1). **b**, Numbers of sgRNAs of the T.spiezzo and T.gonfio libraries originating from existing libraries and resources. **c**, Cross-library comparison of GuideScan specificity scores for the individual

sgRNAs. The top 4 sgRNAs from each library were included, based on their original ranking, for genes that are present in all source libraries. **d**, Cross-library comparison of predicted sgRNA efficacy scores. **e**, Comparison of the number of sgRNAs expected to target alternate alleles of genetically polymorphic regions, based on their prevalence in humans. Only polymorphisms with a frequency >0.1% are considered. **f**, Percentage of qgRNA combinations where ≥1 pair of sgRNAs is spaced fewer than 50 bp apart, potentially leading to interference. In **c** and **d**, the box plot represents the median and interquartile range.

identifier. In T.spiezzo, such unspecific sgRNAs were mostly excluded, whereas in T.gonfio, the proportion of sgRNAs with off-site targets (0.8%) was comparable to the reference pooled libraries because the window of optimal activity around the TSS constrained the choice of target sites (Extended Data Fig. 3g). CRISPR activation of unintended genes may also occur if two genes are located on opposite strands of the genome and share a bidirectional promoter region. Such effects are unavoidable; indeed, when considering a window of 1 kb surrounding the TSS, around 20% of CRISPRa sgRNAs affected additional genes in all examined libraries, including T.gonfio (Extended Data Fig. 3h). All sgRNAs that affect any genes other than the intended gene have been annotated (Extended Data Fig. 3i and Source Data File 1).

### Sequencing the T.spiezzo and T.gonfio libraries

For quality control, we amplified the qgRNA expression cassettes in each well with barcoded primers and subjected pools of amplicons (2.2 kb) to SMRT long-read sequencing (Extended Data Fig. 4a).

To estimate technical errors, 74 single-colony-derived, fully sequenced plasmids bearing distinct qgRNA sequences were included in each sequencing round. The median read count (at CCS7 quality) per plasmid was 86, with ≥10 reads for 98.7% and ≥1 read for 99.9% of plasmids, respectively (Extended Data Fig. 4b).

Mutations, deletions and recombination can occur in plasmids constructed by Gibson assembly. Because ALPA cloning does not rely on colony picking, these alterations may affect a fraction of the plasmid pool in each well. This heterogeneity was quantified by a linked analysis of all four promoter, protospacer and tracrRNA sequences using single-molecule long-read sequencing. Guide RNAs were considered correct if the 20-nucleotide sgRNA and tracrRNA sequences were present and error-free. When considering the median across all wells in the libraries, the percentage of reads with at least one, two, three or four correct sgRNAs was 98%, 94%, 92% and 78% for the T.gonfio library, and 98%, 92%, 89% and 76% for the T.spiezzo library, respectively (Fig. 3a). At the 5th percentile (that is, worse than 95% of wells in the library),

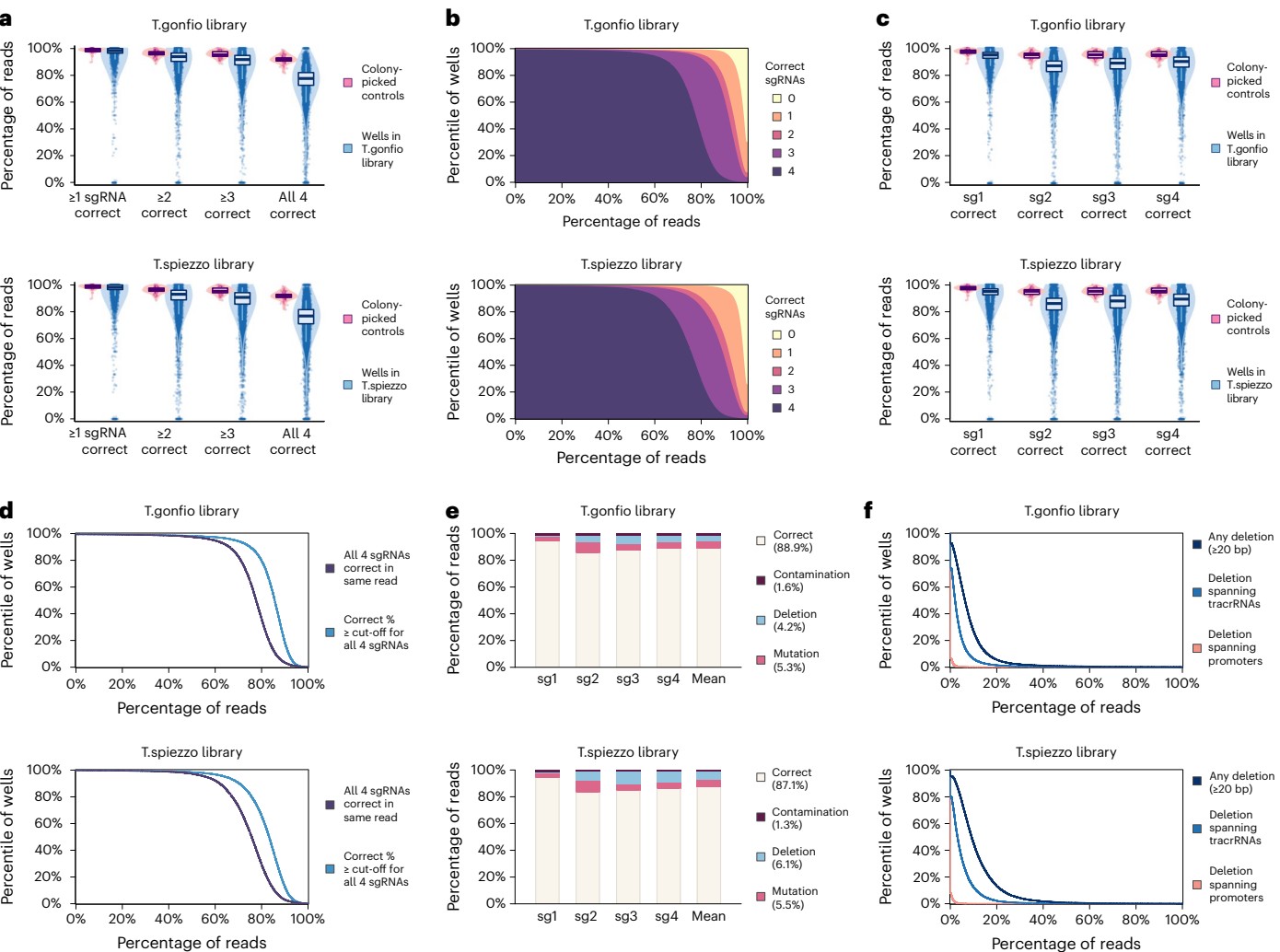

**Fig. 3 | Genome-wide sequencing of the T.spiezzo and T.gonfio libraries.**
**a**, Reads with 0, 1, 2, 3 or 4 correct sgRNAs for each well in the T.spiezzo and T.gonfio libraries (quantitative SMRT long-read sequencing). To assess technical errors, we added barcoded amplicons of 74 single-colony-derived, sequence-validated qgRNA plasmids as internal NGS controls. **b**, Cumulative distribution of the percentage of reads with 0–4 correct sgRNAs in each well of the T.spiezzo and T.gonfio libraries. **c**, Percentage of correct sgRNA-1, sgRNA-2, sgRNA-3 and sgRNA-4 cassettes among the plasmid pools in each well of the T.spiezzo and

T.gonfio libraries. **d**, Percentage of reads with four entirely correct sgRNAs in the same vector (black) and minimum percentage threshold passed by each of the four sgRNAs individually (blue) considering the entire pool of plasmids in each well. **e**, Mean percentage of mutations, deletions and cross-well contaminations in the T.spiezzo and T.gonfio libraries. **f**, Cumulative distribution of plasmids with recombination in each well of the T.spiezzo and T.gonfio libraries. In **a** and **c**, the box plot denotes the median and interquartile range; the whiskers indicate the 5th and 95th percentiles.

the fraction of reads with ≥3 correct sgRNAs was 77% for the T.gonfio library and 71% for the T.spiezzo library (Fig. 3b). Overall, 99.7% of wells passed the minimal quality standard of >50% reads with at least one correct sgRNA (76 wells failed to meet this standard, including 38 wells with zero CCS7 reads). We thus observed acceptable error rates for the vast majority of wells in both libraries.

When considering the four sgRNAs individually, the median percentage of correct reads was ≥85% for all four sgRNAs in both the T.gonfio and T.spiezzo libraries (Fig. 3c). While 65% of wells in the T.gonfio library had ≥75% reads with four entirely correct sgRNAs (within the same read), when each sgRNA is considered separately, 90% of wells were ≥75% correct for each of the four individual sgRNAs (Fig. 3d). Hence, mutations may be compensated for by other clones in the same well. In a more stringent analysis where sgRNAs were considered correct only if the preceding promoter sequence was ≥95% correct, these percentages remained similar (Extended Data Fig. 4c,d).

Incorrect sgRNAs were classified as contaminated (matching sgRNAs from other wells), deleted or mutated (Fig. 3e). Contaminations were rare (1.6% and 1.3% contaminating sgRNAs in T.gonfio and

T.spiezzo, respectively). Large deletions (≥50% of sgRNA and tracrRNA) affected 4.2% and 6.1% sgRNAs in T.gonfio and T.spiezzo, respectively. Deletions spanning two tracrRNAs affected 4.1% of reads, whereas 0.1% of reads contained deletions spanning two promoters (Fig. 3f). The mean percentage of plasmids with deletions affecting ≥1 sgRNA was 8.1% in T.gonfio and 11.4% in T.spiezzo, whereas mutations affected 5.3% and 5.5% of sgRNAs in T.gonfio and T.spiezzo, respectively. These estimates include PCR- and sequencing-derived errors and may over-estimate the error rate. Crucially, off-target activities were acquired in only <0.1% of mutated sgRNAs (Extended Data Fig. 4e,f). Perfectly correct sequences were observed in 88.9% and 87.1% of sgRNAs for T.gonfio and T.spiezzo, respectively. We conclude that ALPA cloning resulted in the generation of these qgRNA libraries with low overall error rates.

## Benchmarking qgRNA ablation plasmids in cells and organoids
Next, we sought to benchmark the T.spiezzo library against existing CRISPR reagents using several gene delivery methods (transduction, transfection and electroporation) with human HCT116 colon

cancer cells, iPSCs and kidney organoids (Extended Data Fig. 5a). We chose epithelial cell adhesion molecule (EPCAM), cell-surface glycoprotein CD44 and phosphatidylinositol glycan anchor biosynthesis class A (PIGA) as targets, based on their detectability with live-cell immunostaining and flow cytometry quantification (Extended Data Fig. 5b). First, we transduced Cas9-expressing HCT116 (HCT116-Cas9) or doxycycline-inducible Cas9-expressing iPSC (iPSC-iCas9) cells with either the lentivirally packaged qgRNA vector or with a pool of four individually packaged sgRNAs (ThermoFisher) at a multiplicity of infection (MOI) of 5. Transduction of both reagents resulted in a time-dependent reduction of EPCAM and CD44 expression. The T.spiezzo lentiviruses achieved higher ablation efficiency in both cell models at 4–8 days post-transduction (Extended Data Fig. 5c). Next, we transfected HCT116-Cas9 cells (iPSCs were not transfected owing to their poor transfectability) with our qgRNA plasmids or a pool of four individual synthetic sgRNAs (Integrated DNA Technologies). While the synthetic sgRNAs showed more rapid ablation than the qgRNA plasmids at day 4 post-transfection, both reagents resulted in a similar reduction of EPCAM and CD44 detection at day 8 (Extended Data Fig. 5d). Next, we electroporated HCT116-Cas9 and iPSC-iCas9 cells with either the qgRNA vectors or a pool of four individual synthetic sgRNAs (Integrated DNA Technologies). Both reagents were active in HCT116-Cas9 cells, but the four synthetic sgRNAs accomplished a faster and more efficient reduction of EPCAM and CD44 than the qgRNA vectors (Extended Data Fig. 5d). In iPSC-iCas9 cells, electroporation of synthetic sgRNAs showed <50% knockout efficacy, whereas the qgRNA vector resulted in fast and highly efficient editing (Extended Data Fig. 5d).

To assess the efficiency of the qgRNA approach in complex cellular models, we used an inducible Cas9 iPSC line[35] to generate nephron progenitor cells and further differentiated them into kidney organoids following established protocols[36]. The PIGA gene, which is essential for the synthesis of glycosylphosphatidylinositol anchors, was targeted, and its editing efficiency was assessed by staining with non-toxic, fluorescently labelled aerolysin (FLAER assay)[11]. Progenitor cells were transduced with the lentivirally packaged qgRNA vector or with a pool of four individually packaged sgRNAs (ThermoFisher) at increasing volumes of viral supernatant, and after 48 days, the organoids were dissociated into single cells and subsequently stained with FLAER. Lentiviruses carrying the qgRNA vector showed high ablation efficiency at lower lentiviral volumes than the respective four individually packaged sgRNAs, despite showing similar levels of p24 (viral coat protein), indicative of comparable viral titres (Extended Data Fig. 5c). These results indicate equal or superior gene perturbation with T.spiezzo compared with existing CRISPR reagents in difficult-to-manipulate cell models.

## An arrayed CRISPR activation screen for TFs controlling PrP$^C$ expression

The cellular prion protein PrP$^C$, encoded by the PRNP gene, is required for the development of prion diseases[37,38]. Previous genome-wide microRNA and siRNA screens have uncovered a complex pattern of regulated expression of PrP$^C$ (refs. 39,40). However, these screens failed to identify any TFs regulating PrP$^C$ expression. We therefore measured PrP$^C$ expression in an arrayed activation screen by time-resolved fluorescence resonance energy transfer (TR-FRET) using a pair of antibodies binding distinct domains of PrP$^C$ (Fig. 4a).

An arrayed T.gonfio sublibrary encompassing all human TFs ($n = 1,634$) was packaged into lentiviral vectors and transduced into U251-MG human glioblastoma cells stably expressing the CRISPR activator dCas9-VPR (MOI = 3). Experiments were performed in triplicate in 384-well microplates, each including 14 wells with NT and 14 PRNP-targeting controls. Cells were lysed 4 days post-transduction; one replicate plate was used to determine cell viability, and two replicates were used to assess PrP$^C$ levels by TR-FRET (Fig. 4a). Levels of PrP$^C$ in NT and PRNP-targeting controls showed that 19 and 4 plates

had a $Z'$ factor[41] of 0–0.5 and >0.5, respectively, whereas 1 plate had a $Z' < 0$ (Fig. 4b,c). The Pearson correlation coefficient ($R^2$) between duplicates was 0.77 (Extended Data Fig. 5e). Hit calling was based on an absolute log$_2$ fold change (FC) of ≥1 and a P value of ≤ 0.05 and resulted in 24 and 12 genes upregulating and downregulating PrP$^C$, respectively (Fig. 4d and Source Data File 2). All 36 hits exhibited conspicuous activation in response to their respective qgRNAs (Extended Data Fig. 5f), underscoring the robust performance of our library and bolstering our confidence in the hits identified by this screen. We then repeated the TR-FRET assay with these candidates and reproduced the effects for 19 up- and 10 downregulators (log$_2$ FC ≥ 0.5) (Fig. 4e). Western blotting for PrP$^C$ confirmed these effects for 11 of the 20 most pronounced modifiers (Fig. 4f,g and Supplementary Figs. 2–4). One of these was the PBX1 homeobox gene previously identified as a modifier of sporadic Creutzfeldt–Jakob disease in a GWAS study[42] ($P = 0.004$; Source Data File 2).

We then assessed PRNP mRNA levels following the individual activation of the 11 modifiers. Four upregulators, HNF4A, PPARA, ZNF81 and KLF12, increased PRNP mRNA by 26–161%, and three downregulators, CHCHD3, BNC2 and TFEB, decreased PRNP mRNA expression by 22–57% (Fig. 4h). Intriguingly, NEUROD4, PBX1 and ZBTB38 had little to no effect on PRNP mRNA but exerted a pronounced impact on PrP$^C$ protein expression (Fig. 4f–h). Hence, certain TFs appear to control PRNP transcription, whereas others may influence its expression indirectly. Elucidating the latter mechanisms may deliver new insights into prion biology.

To assess the reproducibility of arrayed screens, we conducted a second arrayed TF PrP$^C$ screen over a year later, using an entirely different set of antibodies and lentiviruses. The genes identified in the initial screen exhibited substantial overlap with those identified in the subsequent screen (Extended Data Fig. 5g and Source Data File 2). Besides reinforcing the findings of our initial screen, these data underscore the general robustness of arrayed CRISPR activation screens with the T.gonfio library.

## Novel modifiers of autophagy uniquely identified by the pooled T.spiezzo library

The qgRNA plasmids outperformed single sgRNAs from which they were assembled, suggesting that they could also improve the performance of pooled screens, and the reduced complexity of T.gonfio and T.spiezzo (4–10-fold lower than that of 1-sgRNA libraries) may enable leaner, more sensitive and less expensive screens. We tested this presumption by performing a genome-wide CRISPR screen aimed at identifying modulators of autophagy[43,44] with a pool of all 19,820 T.spiezzo qgRNA plasmids, with the Brunello library[14], and by comparing the results with a previous screen performed with the Cellecta library[43]. Each library was packaged into lentiviruses and transduced (MOI = 0.4) into the epithelial cell line H4 cells stably expressing Cas9 and a GFP-tagged version of the SQSTM1 autophagy reporter. Cells were selected with puromycin for 3 days and maintained in non-selective medium for 7 days. Then, GFP$^{high}$ or GFP$^{low}$ (upper or lower quartile of GFP fluorescence, respectively) cells were separated and collected by fluorescence-activated cell sorting (FACS) (Fig. 5a, Extended Data Fig. 6a and Supplementary Fig. 1b). Genomic DNA was isolated and the abundance of sgRNAs was determined by Illumina next-generation sequencing (NGS).

When packaging pooled libraries into lentiviral particles, reverse transcriptase-mediated template switching often generates chimeric products with unpredictable sequences[45–47]. By sequencing sgRNA2 and sgRNA3 sequences from individual qgRNA plasmids in pools of transduced cells, we found that the percentage of intersected reads (correct alignment and linkage of sgRNA2 and sgRNA3) from the T.spiezzo was around 70% (Extended Data Fig. 6b), indicative of considerable lentiviral template switching. For all further analysis with pooled libraries, we only used reads that aligned correctly and where both sgRNAs

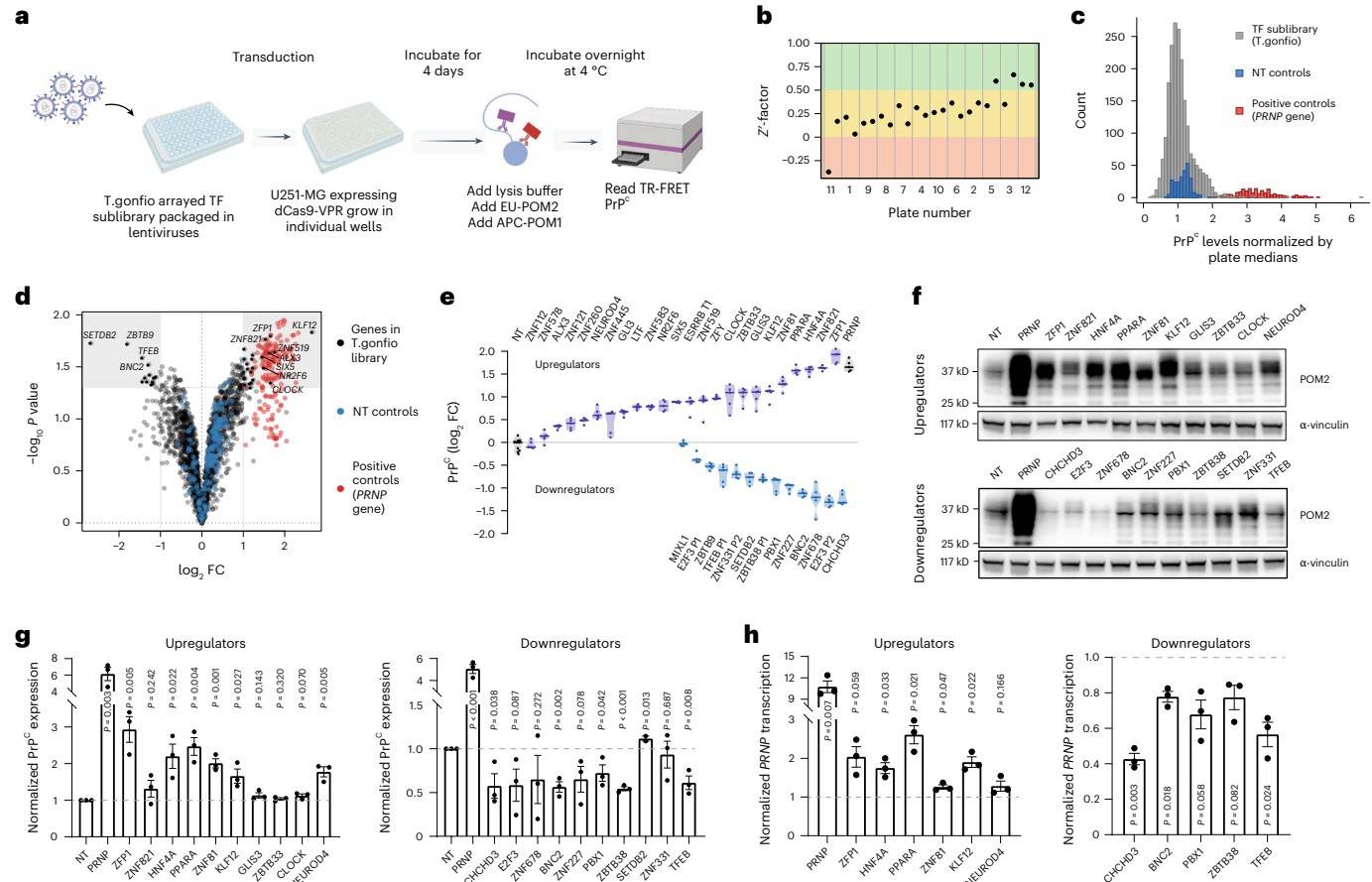

**Fig. 4 | TFs regulating PrP$^C$ expression identified by an arrayed activation screen with T.gonfio. a**, Schematic of the primary PrP$^C$ TF sublibrary screen. Created with BioRender.com. **b**, Z′ factor of each plate from the primary screen. **c**, Distribution of positive controls (qgRNA targeting *PRNP*), negative controls (qgRNA NT control) and samples. **d**, Volcano plot showing −log$_{10}$ P values and log$_2$ FC values across the T.gonfio TF sublibrary. The 36 candidate genes and *PRNP* are depicted in black and red, respectively. **e**, Repetition of the 36 candidate genes in their regulation of PrP$^C$ levels (as in **a**); NT, NT control. P1 or P2 indicates the qgRNA plasmid targeting TSS1 or TSS2 transcriptional start site of a defined gene, respectively (*n* = 5 technical repeats). The central line represents the median. **f**, An example of western blotting analysis of PrP$^C$ expression after

individual activation of the top 10 most pronounced candidate genes from the up- or downregulators confirmed in **e**; POM2, primary antibody against PrP$^C$, α-vinculin, primary antibody against vinculin (internal control) (*N* = 3 biological repeats, except *N* = 2 for *SETDB2*). **g**, Quantification of PrP$^C$ levels after individual activation of the top 10 most pronounced candidate genes from the up- or downregulators as described in **f**. The result was normalized to NT control. **h**, Quantification of *PRNP* transcription levels after individual activation of the confirmed candidate genes from the up- or downregulators as described in **g** (*N* = 3 biological repeats). The result was normalized to NT control. For **g** and **h**, dots: independent experiments (mean ± s.e.m.). *P* values were determined by two-tailed Student's *t*-test.

mapped to the same plasmid. This method does not detect template switches occurring in sgRNA1 and sgRNA4. However, the success of the screens reported here suggests that the randomness of the switches and the high number of targeted cells (300 cells per qgRNA) attenuate the impact of such switches. Thus, the sgRNA2–sgRNA3 sequencing strategy provides an acceptable compromise between quality assurance and cost-effectiveness.

By comparing the representational differences of sgRNA between GFP$^{high}$ and GFP$^{low}$ samples treated with T.spiezzo, Brunello or Cellecta, we identified many well-characterized modulators of autophagy, including *ATG5*, *ATG7*, *BECN1*, *WIPI2* and *PIK3C3* (red dots in Fig. 5b,c and Source Data File 3). Among the core autophagy-related genes, T.spiezzo, Brunello and Cellecta identified 51, 25 and 8 genes in the corresponding screens respectively at an absolute log$_2$ FC ≥1 (Source Data File 3). Further, Gene Ontology (GO) enrichment analysis of the 200 most enriched genes in GFP$^{high}$ samples showed that T.spiezzo identified more autophagy-related pathways than the other libraries (Extended Data Fig. 6c–e). Importantly, the enrichment of validated autophagy modulators was significantly higher for T.spiezzo (Fig. 5d and Extended Data Fig. 6f–m). To directly compare representational

changes between all three screens, we selected the 100 genes with the highest and lowest FC for each screen and intersected the lists to produce a set of shared genes. The T.spiezzo library identified all shared genes with a higher FC compared with Brunello and Cellecta (Fig. 5e).

In addition, T.spiezzo (but neither Brunello nor Cellecta) identified certain genes that had not been highlighted as autophagy relevant by GO analysis (grey dots in Fig. 5b,c). To validate these potential novel modifiers of autophagy, we focused on genes with log$_2$ FC ≥5 in the T.spiezzo screen and ≤1 in the Brunello and Cellecta screens (Source Data File 3). Among the top lists, we found genes (*CFLAR*, *PEX1*, *VAMP1*, *LAMP1*, *USP36* and *SHC1*) that had been reported to modulate autophagy[48–54], supporting the idea that some of these genes may be genuine autophagy regulators.

We investigated the regulation of autophagy by the six genes mentioned above and by ten genes with no reported association with autophagy (*FOXD4L1*, *SLC9A2*, *RBBP8*, *VRK1*, *LCE1C*, *SRFBP1*, *ATP8B2*, *HNRNPM*, *ATXN10* and *GRIN2A*). We first repeated the measurement of GFP-SQSTM1 signal intensity in GFP$^{high}$ and GFP$^{low}$ cell populations after ablating each of these 16 genes. We observed a prominent shift of GFP-SQSTM1 intensity towards GFP$^{high}$ for all 16 genes tested (Fig. 5f

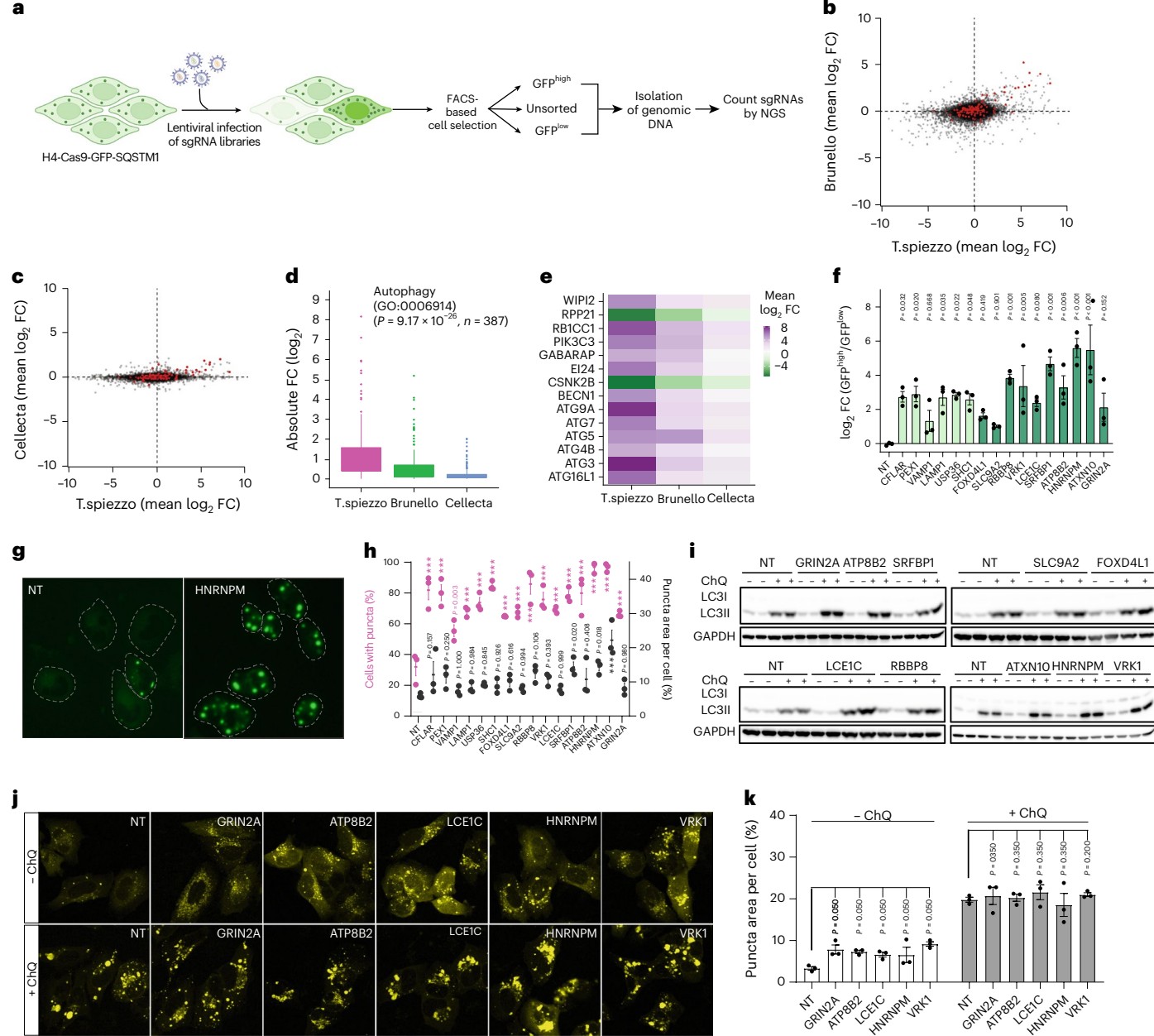

**Fig. 5 | Novel autophagy modifiers identified by pooled T.spiezzo screen.**
**a**, H4-Cas9-GFP-SQSTM1 cells were transduced with genome-wide pooled lentiviral sgRNA ablation libraries (T.spiezzo and Brunello) and selected with puromycin. Cells in the uppermost and lowermost fluorescence quartile were collected by FACS and sgRNAs were quantified by sequencing genomic DNA. Created with BioRender.com. **b,c**, Average $\log_2$ FC in GFP$^{high}$ and GFP$^{low}$ samples from T.spiezzo versus Brunello (**b**) and T.spiezzo versus Cellecta (**c**). Autophagy-relevant genes (autophagy, GO:0006914) are highlighted in red. **d**, Autophagy genes enriched in GFP$^{high}$ cells from the T.spiezzo, Brunello and Cellecta screens. The box plot represents the interquartile range. **e**, Heatmap showing the highest and lowest mean $\log_2$ FC (GFP$^{high}$ versus GFP$^{low}$) of genes identified among the top 100 genes in the T.spiezzo, Brunello and Cellecta screens. **f**, Quantification of $\log_2$ FC of cell count in GFP$^{high}$ versus GFP$^{low}$ populations transduced with T.spiezzo qgRNA lentivirus against NT control or each of the 16 genes selected for validation. The boundaries for GFP$^{high}$ and GFP$^{low}$ cell populations were set according to the NT condition. Dots (here and henceforth): independent experiments, mean ± s.e.m. P values were determined by one-way ANOVA with Dunnett's multiple comparisons test. **g**, An example of GFP-SQSTM1 puncta in

H4-Cas9-GFP-SQSTM1 cells transduced with T.spiezzo qgRNA lentivirus against NT or HNRNPM (see Extended Data Fig. 6o for all genes tested). Dashed lines: cell contours according to the cytosolic GFP signal. **h**, Percentage of cells with GFP-SQSTM1 puncta (purple) and puncta area (black) in H4-Cas9-GFP-SQSTM1 cells transduced with T.spiezzo qgRNA lentivirus against 16 genes selected for validation. P values were determined by one-way ANOVA with Dunnett's multiple comparisons test, ***$P < 0.001$. **i**, Immunoblotting of LC3-II from H4-Cas9 cells transduced with T.spiezzo qgRNA lentivirus against 10 possible autophagy modulators in the absence (−) or presence (+) of ChQ (100 μM, 6 h). GAPDH: loading control. Two biological repeats were assessed for each condition.
**j**, Representative micrographs of YFP-LC3 in H4-Cas9-cells transduced with T.spiezzo qgRNA lentivirus against NT or each of the 5 genes selected for further validation. H4-Cas9 cells were cultured and treated as described in **a** and transduced with YFP-LC3 lentiviruses 48–60 h before examining YFP-LC3 puncta. **k**, Quantification of puncta area of YFP-LC3 of cells and conditions described in **j**. $N = 3$ biological repeats. P values were determined by a one-tailed Mann–Whitney test.

and Extended Data Fig. 6n). In parallel, we assessed the cytosolic distribution of the puncta of GFP-SQSTM1 by microscopy. Ablation of each gene significantly increased the percentage of cells exhibiting puncta of GFP-SQSTM1 over that of NT controls, and ablation of *SRFBP1*, *HNRNPM* or *ATXN10* enlarged the size of the puncta (Fig. 5g,h and Extended Data Fig. 6o).

The convergence between the primary screen and the validation experiments encouraged us to confirm the potential novel autophagy regulators with orthogonal methods. We thus analysed the turnover of a second autophagy marker, LC3-II, by western blotting in the presence or absence of 100 μM chloroquine (ChQ), which blocks autophagic flux by preventing autolysosome maturation[55]. Compared with vehicle treatment, ChQ (6 h) conspicuously increased the levels of LC3-II in all samples. Ablation of *GRIN2A*, *ATP8B2*, *LCE1C*, *HNRNPM* or *VRK1* further increased LC3-II levels in ChQ-treated cells consistently compared with ChQ-treated NT control cells, whereas the other genes showed no or inconsistent effects (Fig. 5i, Extended Data Fig. 6p and Supplementary Fig. 5). We then ablated these five genes individually and examined their effect on the level and distribution of the YFP-LC3 autophagy reporter[56]. All five genes, when ablated, significantly increased the puncta area of YFP-LC3 over NT controls but did not further modulate the area of YFP-LC3 puncta after ChQ treatment (Fig. 5j,k). We infer that the pooled T.spiezzo screen identified *GRIN2A*, *ATP8B2*, *LCE1C*, *HNRNPM* and *VRK1* as novel bona fide autophagic flux modulators.

### T.gonfio library for targeted epigenetic silencing

CRISPR-mediated targeted epigenetic silencing[57,58] is a useful alternative to gene ablation, especially for cells sensitive to DNA breakage such as iPSCs[59]. Targeted gene silencing by CRISPRoff is robust and persists even after iPSC differentiation into neurons[58]. We found that 96.8% of the T.gonfio sgRNAs coincide with the targeting window for CRISPRoff, whereas most of the remaining 3.2% of sgRNAs fall within 100 bp of it (Fig. 6a). This encouraged us to examine the effectiveness of the T.gonfio plasmids for CRISPRoff.

qgRNA plasmids targeting the TSSs of the cell-surface proteins ITGB1, CD81 and CD151 were co-transfected into HEK293T cells with a CRISPRoff plasmid or, as a control, a CRISPRoff mutant encoding a catalytically inactive DNA methyltransferase. A pool of three single sgRNA plasmids used in the CRISPRoff study[58] was used as reference for each gene. In all cases, the qgRNA plasmids achieved similar gene silencing to the CRISPRoff pool (Fig. 6b,c). We then used three cell-surface proteins, CD47, IFNGR1 and MCAM, to compare the silencing efficacy of CRISPRoff with T.spiezzo-induced gene ablation. The T.spiezzo qgRNAs with Cas9 induced 80–90% gene ablation, and the T.gonfio plasmids with CRISPRoff induced a similar extent of gene silencing (Fig. 6d,e).

To test the generalized suitability of T.gonfio-mediated epigenetic silencing, we benchmarked T.gonfio against a refined pooled dual-sgRNA CRISPRoff library, which prioritizes guides according to a more expansive set of screening data[58,60]. The TagBFP-expressing qgRNA T.gonfio (pool of 22,442 plasmids) and the GFP-expressing dual-sgRNA CRISPRoff libraries were packaged into lentiviral particles and transduced into HEK293T cells (MOI = 0.3–0.4). After puromycin selection, cells were transfected with a CRISPRoff plasmid expressing the red fluorescent protein mScarletI. Two days later, GFP+/TagBFP+ mScarletI+ cells were sorted and maintained without antibiotic selection for ≥10 cell divisions, allowing for the ablation of cells with silenced essential genes. Guide abundances were compared between cells at baseline (immediately before the transfection of CRISPRoff-mScarletI) and at the endpoint (after sorting and passaging) (Fig. 6f and Supplementary Fig. 1c,d). The phenotype score γ, defined as (log₂ (sgRNA enrichment))/(number of cell doublings), was calculated for each gene (Fig. 6f). The dual-sgRNA CRISPRoff library induced a similar growth defect for the set of essential genes as in the original CRISPRoff study[58] (Fig. 6g, Extended Data Fig. 7a and Source Data File 4). While the fitness effects of the pooled T.gonfio and CRISPRoff libraries were similar, the

T.gonfio library produced a narrower distribution of phenotype scores for both essential genes and non-essential genes (Fig. 6g and Extended Data Fig. 7a). Accordingly, the strictly standardized mean difference (SSMD) score for T.gonfio was higher than that for CRISPRoff (Fig. 6h), indicating a stronger separation between essential and non-essential genes. Furthermore, T.gonfio showed a better accuracy and a lower false-positive rate for essential genes (Fig. 6i), presumably owing to the enhanced power of the quadruple-guide strategy. Hence, T.gonfio can be usefully deployed for arrayed epigenetic silencing screens.

### Template-switching elimination enhances screen performance

Vectors containing multiple sgRNAs are highly performant. However, they are prone to lentiviral template switching, which occurs during lentiviral assembly and uncouples the sequencing barcodes from the respective sgRNAs. We observed 20–30% template switches between sgRNA2 and sgRNA3, implying even higher rates across the entire qgRNA cassette (Fig. 6j and Extended Data Fig. 6b). Mixing lentiviral particles that had been grown in separate wells ('post-pooling') eliminated template switching as expected (Fig. 6j). We thus compared CRISPRoff essentialome screens performed with a post-pooled T.gonfio library (Extended Data Fig. 2i) and a pre-pooled T.gonfio library where qgRNA plasmids were pooled before lentiviral packaging (Fig. 6f, Extended Data Fig. 7b and Source Data File 4).

The post-pooled library enhanced the dropout of essential genes compared with the pre-pooled screen, while no difference was observed for non-essential genes (Fig. 6k,l). Furthermore, the post-pooled library exhibited improved discrimination between essential and non-essential genes, as evidenced by higher SSMD values (Fig. 6m). We conclude that post-pooling, by eliminating lentiviral template switching, enhances the performance of genome-wide pooled screens.

## Discussion

Genome-wide CRISPR phenotypic screens are clarifying fundamental biological phenomena[3,4]. Arrayed CRISPR libraries expand their scope to the study of complex, non-autonomous phenotypes. The libraries described here enable disparate modalities of gene perturbation, including activation, silencing and ablation. Importantly, arrayed CRISPR screens can be less laborious than generally presumed, and can be performed rapidly by standardizing workflows with inexpensive automation steps.

### The ALPA cloning method

Covering the entire protein-coding genome with ablating, activating and silencing tools required the generation of >42,000 individual plasmids. This would have required prohibitive resources, since agar plating, colony picking and gel extraction of desired DNA fragments[17] are time-consuming and refractory to automation. The ALPA plasmid construction method radically simplifies and parallelizes the cloning procedure, enabling the generation of ~2,000 individual plasmids per week. The principles underlying ALPA cloning are adaptable to any large-scale molecular cloning scenarios, including, for instance, the saturation mutagenesis of proteins. Hence, ALPA cloning is a cost-effective high-throughput molecular cloning method, which may considerably expand the generation of large-scale plasmid collections.

### Efficacy of CRISPR-based gene perturbation

Although the algorithms for predicting sgRNA activity are continuously improving[13], the efficacy of gene-perturbation screens can be jeopardized by suboptimal guide efficiency, leading to false-negative hit calls. Pooled libraries can partially circumvent this by increasing the number of sgRNA vectors targeting the same gene. This strategy is cost-inefficient for arrayed genome-wide libraries. Thus, augmenting the efficiency and robustness of gene perturbation is key for high-performance arrayed genome-scale libraries.

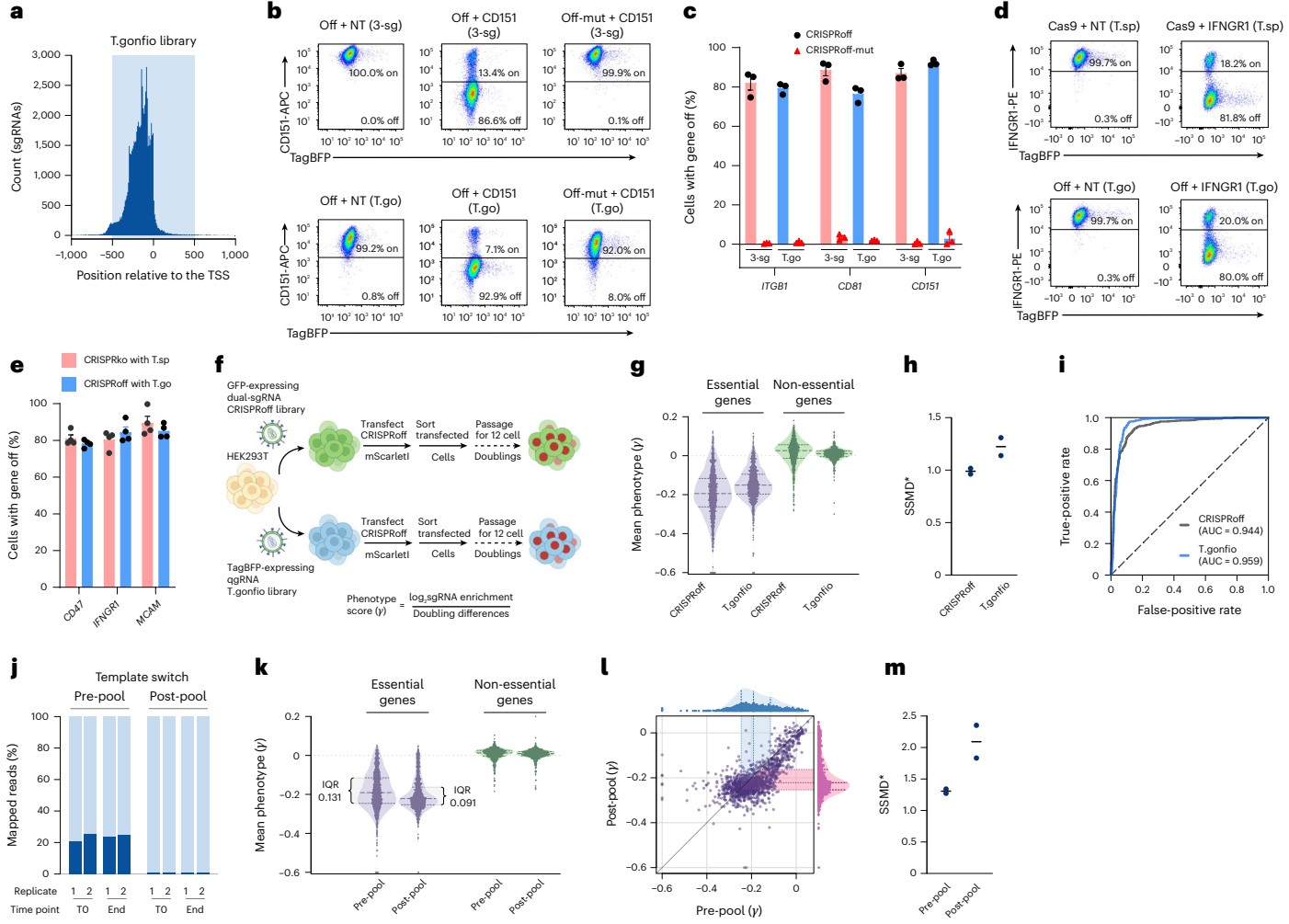

**Fig. 6 | Targeted epigenetic silencing (CRISPRoff) with the T.gonfio library.**
**a**, Alignment of the T.gonfio sgRNA sequences to the CRISPRoff targeting window. **b**, An example of flow cytometry measurement of CD151 in HEK293T cells after exposure (10 days) to pools of three separated sgRNAs (3-sgRNA, used in the CRISPRoff study) or the respective T.gonfio qgRNA. 'Off' represents the CRISPRoff plasmid; 'off-mut', CRISPRoff mutant carrying a catalytically inactive version of the DNA methyltransferase. **c**, Quantification of the percentage of cells with ITGB1, CD81 and CD151 silencing 10 days after co-transfection of the CRISPRoff or CRISPRoff-mutant plasmids with pools of three single sgRNAs (3-sg) or the qgRNA plasmid from the T.gonfio library in HEK293T cells. Each dot represents an independent biological repeat of the assay. Data are presented as mean ± s.e.m. **d**, An example of a flow cytometry measurement of the percentage of cells with IFNGR1 silencing 10 days post-CRISPR knockout with the qgRNA plasmid from the T.spiezzo library or CRISPRoff with the qgRNA plasmid from the T.gonfio library in HEK293T cells. **e**, Quantification of the percentage of cells with CD47, IFNGR1 and MCAM silencing 10 days post-CRISPR knockout with qgRNA plasmids from the T.spiezzo library or CRISPRoff with qgRNA plasmids from the T.gonfio library in HEK293T cells. Each dot represents a biological repeat of the assay. Data are presented as mean ± s.e.m. **f**, A schematic of pairwise pooled genome-wide CRISPRoff screens to determine the effectiveness of T.gonfio for gene silencing at the genome scale in HEK293T cells. Created with BioRender.com. **g**, Mean phenotype scores ($\gamma$) for essential and non-essential genes from screens with the CRISPRoff or the T.gonfio library. $N = 2$ biological repeats. The dashed lines indicate the median and interquartile range. **h**, SSMD* value measuring the separation between essential and non-essential genes for each repeat of the screens performed with the CRISPRoff or T.gonfio pool libraries. **i**, Plots of true and false positive rates for the distinction between essential and non-essential genes for screens with the CRISPRoff or the T.gonfio library. AUC, area under the curve. **j**, Lentiviral template switching between sgRNA2 and sgRNA3 in pre-pooled versus post-pooled T.gonfio libraries. T0 and end: baseline and endpoint as defined in the text. **k**, Mean phenotype scores ($\gamma$) for essential versus non-essential genes in screens with pre-pooled versus post-pooled T.gonfio libraries (2 biological repeats); IQR, interquartile range. **l**, Mean phenotype scores ($\gamma$) for essential genes in post-pooled versus pre-pooled screens. IQRs for each screen were depicted. **m**, SSMD* between essential and non-essential genes for screens performed with the pre-pooled versus post-pooled library. In **g**, **k** and **l**, values above 0.2 or below −0.6 were set to 0.2 and −0.6, respectively.

The integration of qgRNA into each targeting vector substantially improves CRISPR-mediated gene activation, ablation and silencing.

## Versatility of qgRNA-based libraries

To ensure the broadest possible adaptability to multiple experimental model systems (immortal cell lines, iPSCs, organoids or primary cells), we included two selection markers in our qgRNA vector (puromycin resistance and TagBFP) and the motifs necessary for lentiviral packaging and transposon-mediated integration. All these features were inherited from the lenti-PB vector (Methods); our results confirm the findings of previous studies with this vector. Furthermore, each sgRNA is driven by a different housekeeping promoter, ensuring activity in the broadest range of cells and tissues and minimizing the risk of transcriptional silencing by promoter methylation. The sgRNA selection algorithm was tuned to identify the least polymorphic regions of each gene, thereby extending the likelihood of perturbation to patient-derived cells that may substantially differ from the human reference genome.

Unexpectedly, the T.gonfio library performs very well in epigenetic silencing (CRISPRoff) screens. This opens the possibility of performing both loss-of-function and gain-of-function screens using the same library in cell lines expressing the appropriate dCas9 proteins, further saving time, cost and labour in the execution of gene-perturbation screens. Genome-wide pairwise screens showed that T.spiezzo and T.gonfio provided increased signal-to-noise ratios over the existing pooled libraries, Brunello[14] and CRISPRoff[58]. Gene ablation by enzymatically active Cas9 can induce genotoxic double-strand DNA breaks. Although we did not observe qgRNA-associated genotoxicity in T.spiezzo-treated HEK293 cells, other studies suggest that toxicity can be cell line and genomic region dependent. In such contexts, T.gonfio-mediated epigenetic silencing may offer advantages over T.spiezzo-mediated gene ablation.

The flexibility of arrayed libraries allows for 'post-pooling', that is, mixing individually produced lentiviral particles. This eliminates lentiviral template switching, improves the signal-to-noise ratio and increases the sensitivity of pooled screens. Hence, the arrayed production of lentiviruses, particularly upon automation of its most laborious steps such as cell seeding and dispensing of transfection reagents, can enable the execution of pooled screens with improved performance. Lastly, our libraries have a much smaller size compared with existing libraries that include up to 10 guides per gene. This can not only reduce workload and cost but enables screens when cell numbers are limiting—which is a frequent problem with human primary cells.

### Quality and homogeneity of the qgRNA vectors

By substantially lowering the percentage of incorrect plasmid assemblies, ALPA cloning eliminates the necessity of isolating clonal bacterial colonies. Consequently, each ALPA cloning reaction product generally represents a polyclonal pool of plasmids. This source of variability was quantitatively assessed by sequencing: in the average well, 90% of the plasmid population contained ≥3 intact sgRNAs. Some qgRNA plasmids showed mutations in the region of the sgRNA and tracrRNA sequences, possibly originating from oligonucleotide synthesis and Taq DNA ligase reactions. The mutation rate found in our sgRNA vector sequences is consistent with the expected error rate occurring during Gibson assembly, leading to mutations in approximately 10% of the plasmids[20].

Despite the use of four different promoters and tracrRNA variants, we observed a recombination between sgRNA expression cassettes in ~10% of reads, resulting in a deletion of the intervening sequence. However, 85% of recombining plasmids retained ≥1 correct sgRNA sequence, and in the median well, 99.7% of reads had at least one sgRNA + tracrRNA module that was 100% correct. The average percentage of entirely correct protospacer and tracrRNA sequences for each of the four sgRNAs was ~90%.

Mutated sgRNAs may target genomic sites illegitimately and induce off-target effects. This affected <0.5% of mutated sgRNAs and targeted additional genes in only ~0.1% of cases (Extended Data Fig. 4e,f). Hence, the sequence alterations in ALPA-cloned plasmid pools had no practical effect apart from a slight reduction in the number of active sgRNAs. Notably, the 74 single-colony-derived and Sanger sequencing-confirmed control plasmids showed several errors attributable to PCR or sequencing steps, suggesting that the error rates reported for T.gonfio and T.spiezzo are likely overestimates. Moreover, PiggyBac transposition can introduce large numbers of qgRNA cassettes into each cell, thereby causing hyperelevation of qgRNA expression and, potentially, substantial off-target effects.

### An arrayed assessment of TFs regulating PrP^C expression

The cellular prion protein PrP$^C$ encoded by the *PRNP* gene is required for the development and pathogenesis of prion disease, as evidenced by studies that mice devoid of PrP$^C$ coding gene *PRNP* are resistant to prion infection[37] and neural tissues lacking PrP$^C$ are resistant to

scrapie-induced toxicity[61]. Thus, genes or drugs modifying the level of PrP$^C$ expression are of general interest for both the basic biological understanding and therapeutic treatment of prion diseases. We have previously identified a wealth of modulators of PrP$^C$ expression using arrayed genome-wide screens with microRNAs[40] and siRNAs[39]. However, these screens did not identify any TFs regulating PrP$^C$ expression, and the transcriptional regulation of *PRNP* has remained largely unexplored. Our arrayed CRISPR activation screen focusing on the TF sublibrary has thus partially filled the gap by successfully identifying 11 TFs regulating PrP$^C$ expression, and these TFs provide an opportunity for further study to close the gap.

### Identification of autophagy modifiers

Autophagy is a highly regulated biological process that includes the formation of autophagosomes, autophagosomal engulfment of cytoplasmic components and organelles, fusion of autophagosomes with lysosomes to produce autolysosomes, and the degradation of intra-autophagosomal components by lysosomal hydrolases[62]. Approximately 200 genes have been identified as core components of autophagic flux[63].

Dysfunction of autophagy has been implied in diseases, including neurodegenerative disease, diabetes, tumours and immune diseases[63]. The accumulation of GFP-tagged SQSTM1 provides a reliable proxy of autophagic activity, motivating us to compare the sensitivity and specificity of a pooled T.spiezzo version with the two pooled CRISPRko libraries, Brunello and Cellecta. Not only did T.spiezzo exhibit superior sensitivity compared with the Brunello and Cellecta libraries, but also it identified several novel autophagy modifiers that were not uncovered by the other libraries, five of which were subsequently validated using orthogonal methods. Beyond contributing to our understanding of autophagy and providing a wealth of new potential therapeutic targets, these results add to the evidence that the T.spiezzo and T.gonfio libraries represent powerful tools for genome-wide interrogations in both arrayed or pooled modalities.

### The effect of lentiviral template switch in pooled screens

Template switching is a ubiquitous artefact of pooled lentiviral vectors. While it is most prominent in vectors featuring multiple sgRNAs, it can affect also single sgRNAs whose identities are inferred by their associated barcodes. Its pervasiveness can impair the sensitivity of pooled screens and, in the worst case, cause incorrect hit calling. The arrayed production of individual lentiviral vectors allows for 'post-pooling' and, consequently, for pooled libraries completely devoid of any lentiviral template switch. A direct comparison of the pre-pooled and post-pooled versions of the T.gonfio library showed that post-pooling improved the phenotypic score of screens while not radically changing the ranking of identified hits (Fig. 6k,l), perhaps because of the strong selective pressure occurring in essentialome screens. The high yield of novel genetic modifiers of autophagy in our pooled ablation screen suggests that similar considerations also apply to the pre-pooled version of the T.spiezzo library.

## Methods

### DNA constructs

The DNA constructs (except for the qgRNA expression plasmids, whose construction by the ALPA cloning method was described below) used in the study originated from Addgene-deposited plasmids, including lentiCas9-Blast (#52962), SP-dCas9-VPR (#63798), lenti-dCas9-VPR-Blast (#96917), PB-TRE-dCas9-VPR (#63800), psPAX2 (#12260), VSV-G (#8454), lentiGuide-Hygro-eGFP (#99375), lentiGuide-Hygro-dTomato (#99376) and pLVX-LC3-YFP (#99571). The CRISPRoff and CRISPRoff-D3A mutant plasmids were gifted by L. A. Gilbert and J. S. Weissman from University of California, San Francisco. The transposase plasmid was gifted by E. Zuo from Agricultural Genomics Institute at Shenzhen Chinese Academy of Agricultural

Sciences. The pYJA5, CRISPRoff-mScarletI and single sgRNA plasmids were constructed in-house.

The pYJA5 construct was created by modifying the lenti-PB vector[11] (a gift from A. Bradley) in two steps. First, the DNA fragment flanked by the recognition sites for the restriction enzymes *Mlu*I and *Age*I in the lenti-PB vector was replaced by a synthesized DNA fragment that included the human U6 promoter and the fourth variant of tracrRNA, as well as an ampicillin resistance gene (β-lactamase expression cassette). Two *Bbs*I (type II restriction enzyme) recognition sites flanking the β-lactamase expression cassette were introduced into the new fragment to facilitate the removal of the β-lactamase expression cassette. In a second step, the original ampicillin resistance (β-lactamase) expression cassette in the lenti-PB vector was removed between the two *Bsp*HI restriction enzyme recognition sites. After its removal, we inserted a qgRNA expression cassette containing a trimethoprim resistance gene (dihydrofolate reductase), enabling ALPA cloning. Furthermore, all *Bsm*BI recognition sites were mutated. Detailed sequences of the pYJA5 and qgRNA-pYJA5 constructs are included in Supplementary Information. The CRISPRoff-mScarletI plasmid was constructed by replacing the TagBFP cDNA fragment with the mScarletI cDNA sequences. The original CRISPRoff plasmid DNA (except the region encoding TagBFP) and mScarletI cDNA from the pmScarlet-i_C1 plasmid (#85044) were amplified by PCR with the Phusion high-fidelity DNA polymerase (New England Biolabs M0530L). Both amplicons were assembled by Gibson assembly to produce the desired CRISPRoff-mScarletI plasmid. Single sgRNAs were cloned into the pYJA5-modified vector individually via a previously established method[64].

All plasmids were sequence-confirmed. The backbone vector pYJA5 (#217778), two NT control plasmids of each library (#217779-217782, which can be used as ready-to-go controls and provide the three constant regions for the three-fragment PCRs for qgRNA cloning) and the CRISPRoff-mScarletI plasmid (#217783) were deposited at Addgene.

### Quantification of gene activation by real-time quantitative PCR

HEK293 cells were seeded in 24-well plates at a density of $4.0 \times 10^5$ cells per well. On the second day, cells at 80–90% confluency were co-transfected with 0.25 µg of single-sgRNA (or qgRNA) plasmids and 0.25 µg of dCas9-VPR plasmids using Lipofectamine 3000 (Thermo Fisher Scientific). Three days post-transfection, cells were lysed, and their total RNA was isolated using TRIzol Reagent (Thermo Fisher Scientific) according to the manual. For iPSC-derived neurons, which stably express dCas9-VPR, qgRNA lentiviruses were transduced at an MOI of 1.4. Seven days post-transduction, cells were lysed, and RNA extracted using the same method as described for the HEK293 cells. RNA (600 ng) was reverse transcribed into cDNA via the QuantiTect Reverse Transcription Kit (Qiagen). Real-time quantitative PCR was done with SYBR green (Roche) according to the manufacturer's instructions with the primer sets detailed in Supplementary Information. *GAPDH*, *ACTB* and *HMBS* were used as controls.

### Gene activation-associated cell–cell heterogeneity assay

Polyclonal HEK293 cells stably expressing doxycycline-inducible dCas9-VPR (PB-TRE-dCas9-VPR, hygromycin selection) were generated by co-transfection of the PB-TRE-dCas9-VPR (500 ng) and transposase (500 ng) plasmids with Lipofectamine 3000 in 6-well plates. Cells were split on the following day and transferred to medium containing hygromycin (100 µg ml⁻¹, GIBCO) for 5–7 days. Then, cells were split and transduced with single sgRNAs, qgRNAs or NT control (hNTa) lentiviruses at an MOI of 0.3. After 2 days, cells were split with puromycin-containing medium (1 µg ml⁻¹, GIBCO), selected for 5–7 days with medium change every other day and seeded in 6-well plates at a density of $2.0 \times 10^5$ cells per well. Cells were grown for 3 days in doxycycline hyclate-containing medium (1 µg ml⁻¹, Sigma-Aldrich), and the medium was refreshed every day.

At day 3 post-doxycycline induction of dCas9-VPR expression, cells were trypsinized and resuspended as a single-cell suspension in Cell Staining Buffer (BioLegend 420201) at a density of $6 \times 10^5$ cells in 100 µl. Cells were pre-incubated with 5 µl of Human TruStain FcX (Fc Receptor Blocking Solution, BioLegend 422301) per 100 µl of cell suspension for 5–10 min at room temperature. Afterwards, APC-conjugated antibodies to human CD2 (BioLegend 300214), CD4 (BioLegend 357408) or CD200 (BioLegend 329208) were added and incubated on ice for 15–20 min in the dark. After 20 min, cells were washed twice with 2 ml of Cell Staining Buffer by centrifugation at $350 \times g$ for 5 min. The cell pellet was resuspended in 400 µl of Cell Staining Buffer, transferred to FACS tubes, centrifuged at 1,000 rpm for 1 min and finally analysed with a Fortessa (BD) LSR II analyser at the FACS core facility of the University of Zurich. NT sgRNA-treated cells were used as controls.

### Quantification of gene ablation efficiency by live-cell antibody staining

To examine the gene ablation efficiencies of *CD47*, *IFNGR1* and *MCAM* in HEK293 cells, HEK293 cells at 80–90% confluency were co-transfected with the lentiCas9-Blast (250 ng per well) and sgRNA plasmids (250 ng per well) using Lipofectamine 3000 in 24-well plates. Twenty-four hours post-transfection, cells were split to puromycin (1 µg ml⁻¹)-containing medium for 72 h and cultured in medium without selection for around 1 week. During the whole procedure, cells were maintained at a confluency of no more than 80%. On the day of the assay, cells were trypsinized and single cells were stained with the respective CD47 (BioLegend 323124), IFNGR1 (BioLegend 308606) or MCAM (BioLegend 361016) antibody and analysed using a similar approach as described for the gene activation-associated cell–cell heterogeneity assay mentioned above. NT sgRNA-treated cells were used as controls.

To examine gene ablation efficiency of qgRNA and other reagents in various cell types and organoids, HCT116-Cas9 cells were grown in DMEM medium (GIBCO) supplemented with 10% FBS (GIBCO) and 1× penicillin/streptomycin (GIBCO) in a humidified incubator at 37 °C with 5% $CO_2$. iPSC-iCas9 cells were cultured in mTeSR (Stem Cell Technologies) supplemented with penicillin/streptomycin (GIBCO) and doxycycline (200 ng ml⁻¹; Clontech) on laminin-521 (Biolamina) coated plates at 37 °C and 5% $CO_2$. For routine maintenance, approximately 70% confluent cultures were dissociated into single cells with TrypLE (GIBCO) and seeded at a seeding density of 10,000–25,000 cells per cm² in mTeSR supplemented with 2 µM ROCK inhibitor Y27632 (Tocris) onto laminin-521 coated plates. After 24 h, the medium was replaced with mTeSR without ROCKi followed by daily medium changes. Kidney organoid differentiation and maintenance were performed as described[35].

For transduction, the MOI of lentivirus was 5 and the gene knockout efficiency was measured 4 or 8 days (HCT116-Cas9 or iPSC-iCas9) or >14 days (NPC-iCas9) after transduction. For transfection, HCT116-Cas9 or iPSC-iCas9 cells were transfected at 60–80% confluency with Lipofectamine 2000 using the qgRNA plasmids (5 µg) or synthetic guides (10 µM) complexed with the tracrRNA following the manufacturer's protocols (ThermoFisher and IDT), respectively. Gene ablation efficiency was measured 4 or 8 days after transfection. For nucleofection, $2 \times 10^5$ HCT116-Cas9 and iPSC-iCas9 were resuspended in 20 µl SE cell line nucleofection solution (Lonza) (HCT116-Cas9) or P3 primary cell nucleofection solution (Lonza) (iPSC-iCas9). Cells were mixed and incubated at room temperature for 2 min in PCR tubes. The qgRNA plasmid (5 µg) or synthetic guide RNAs (10 µM) were mixed and the cell/reagent/nucleofection mix was transferred to nucleofection cuvette strips (Lonza). Cells were electroporated using a 4D nucleofector (4D-Nucleofector Core Unit, Lonza, AAF-1002B; 4D-Nucleofector X Unit, AAF-1002X; Lonza). Programmes were adapted for the different cell types (HCT116-Cas9, EN-113; iPSC-iCas9, CD-118). After nucleofection, transfected cells were transferred in culture plates containing pre-warmed cell-specific growth media. Gene ablation efficiency was

measured 4 or 8 days post-nucleofection. All gene ablation efficiencies were assessed by the aforementioned live-cell staining method with 1 µl of Alexa488 anti-human EPCAM (Abcam ab112067) or Alexa647 anti-mouse/human CD44 (BioLegend 103018).

For the quantification of gene ablation efficiency in organoids, NPCs were transduced with lentiviruses carrying the qgRNA plasmid or a pool of four individual lentiviruses each carrying an sgRNA targeting *PIGA*. After 46 days post-transduction, organoids were dissociated into single cells and stained with FLAER-488 reagent (Biozol) in 3% BSA (blocking solution) according to the manufacturer's protocol. Subsequently, the percentage of FLAER-negative cells in each condition was analysed using a Fortessa FACS analyser (BD).

## Quantification of gene editing efficiency via SMRT long-read sequencing

HEK293 cells were cultured, transfected and maintained in the same manner as for the gene ablation efficiency measurements by the live-cell antibody staining assay. On the day of the assay, cells were collected for genomic DNA isolation using the DNeasy Blood and Tissue Kit (Qiagen 69506). Barcoded primers flanking qgRNA targeting sites were synthesized to amplify the genome-edited region of the corresponding genes using the Phusion high-fidelity DNA polymerase (New England Biolabs M0530L). For each PCR reaction of 50 µl volume, 150 ng genomic DNA, 0.5 µl Phusion DNA polymerase, 10 µM forward/ reverse primers, 10 mM dNTP and 10 µl 5× Phusion HF buffer were used, and the following temperature conditions: initial denaturation at 98 °C for 30 s, 36 cycles at 98 °C for 10 s, 60 °C for 30 s, and 72 °C for 30 s per kb, and final extension at 72 °C for 6 min. PCR products were purified with gel extraction using the NucleoSpin gel and PCR clean-up kit (Macherey Nagel 40609.250). Purified PCR amplicons were pooled in approximately equimolar amounts (determined by Nanodrop) and subjected to SMRT long-read sequencing. The individual reads were demultiplexed and aligned to their corresponding amplicon sequences amplified from cells treated with NT sgRNAs. Gene editing efficiency was calculated as the percentage of mutated reads compared with the corresponding NT controls. For each gene, their Fwd bc1, bc2, bc3 and bc4 primers were designed with 4 distinct 10 bp barcodes to index the 4 biological repeats, and their Rev primers bc1 and bc2 were designed with 2 distinct 10 bp barcodes to index NT and qgRNA plasmids transfected cells, respectively. The barcoded primers (high-performance liquid chromatography purified grade) are detailed in Supplementary Information.

## Cell-growth competition assay

HEK293 cells stably expressing dTomato were generated by transducing the lentiGuide-Hygro-dTomato lentivirus. HEK293-Cas9-EGFP cells were generated by co-transducing HEK293 cells with lentiCas9-Blast and lentiGuide-Hygro-eGFP lentiviruses. Non-transduced cells were eliminated with hygromycin (100 µg ml⁻¹, GIBCO) and blasticidin (10 µg ml⁻¹, GIBCO). Polyclonal cells of both lines were used in the assay. The two stable cell lines were then mixed and seeded in a ratio of around 1:1. The next day, cells were analysed with a Fortessa (BD) LSR II analyser at the core facility centre of the University of Zurich to validate the ratio of EGFP/dTomato of the starting cells. Afterwards, cell mixtures were seeded on 12-well plates and transduced at an MOI of 0.3 with lentivirus containing different qgRNA knockout plasmids targeting different genes or NT controls. Two days post-transduction, cells were split with puromycin (1 µg ml⁻¹, GIBCO)-containing medium and selected for 5–7 days with medium change every other day. At day 14 after the lentiviral transduction of sgRNAs, cells were collected, resuspended in PBS, transferred to FACS tubes, centrifuged at 1,000 rpm for 1 min and analysed with a Fortessa (BD) LSR II analyser at the core facility centre of the University of Zurich to determine the ratio of EGFP/dTomato positive cells. The final ratio was normalized to the starting ratio measured on day 2 after seeding the cell mixtures.

## Lentiviral packaging

For production of individual or small numbers of lentiviral vectors, HEK293T cells were grown at 80–90% confluency in DMEM + 10% FBS medium on poly-D-lysine-coated 24-well plates and transfected with the 3 different plasmids (sgRNA plasmid, psPAX2 and VSV-G; ratios 5:3:2) with Lipofectamine 3000 for lentivirus production. After incubation for 6 h or overnight, the medium was changed to virus harvesting medium (DMEM + 10% FBS + 1% BSA). The supernatant containing the lentiviral particles was collected 48–60 h after the change to virus harvesting medium. Suspended cells or cellular debris was pelleted with centrifugation at 1,500 rpm for 5 min, and then the supernatant was collected and stored at −80 °C.

For the titration of lentiviral particles, $2 \times 10^5$ HEK293T cells were grown in 24-well plates and infected by adding small volumes ($V$) of the above-mentioned viral supernatant (such as 3 µl). A representative batch of cells was used to determine the cell count at the time of infection ($N$). Seventy-two hours after infection, the cells were collected and analysed by flow cytometry to quantify the fraction of infected cells (BFP-positive). The percentage of positive cells ($P$) is then used to calculate the titre ($T$) of the virus according to the following formula: $T = \frac{P \times N}{V}$. The details of high-throughput lentiviral production in 384-well plates are described in Supplementary Information.

## ALPA cloning high-throughput generation of libraries

Twenty-nucleotide sgRNA sequences were incorporated into oligonucleotide sequences with appended constant sequences and synthesized in 384-well plates using the high-affinity purification method by Sangon Biotech. The oligonucleotides were diluted with ddH₂O to a working concentration of 4 µM. Then the C1, M and C2s PCRs were performed in 10 µl of PCR reaction per well individually in 384-well plates with the Integra ViaFlo 384-well pipetting system. PCR plates were tightly sealed and centrifuged at 2,000 rpm for 2 min and placed in thermocyclers with the following programme: preheat the lid at 99 °C; initial denaturation at 98 °C for 30 s, 36 cycles comprising 98 °C for 10 s, 60 °C for 30 s, and 72 °C for 25 s, and final extension at 72 °C for 5 min, followed by cooldown to 20 °C. PCR products were diluted with the addition of 9 µl of ddH₂O. The success of PCR on each plate was confirmed by DNA agarose gel electrophoresis of several random samples on the plate. Then 2 µl, 1 µl and 1 µl of C1, M and C2s PCR diluted products, respectively, were added to a mixture containing 1 µl (120 ng) of pYJA5 *Bbs*I-digested purified vector and 5 µl of 2× homemade HiFi Gibson master mix for the Gibson assembly. The mixture was incubated in the thermocycler at 50 °C for 1 h and then used for the transformation of competent cells or stored immediately at −20 °C. A detailed method is provided in Supplementary Information.

## Transformation of ALPA cloning product into *E. coli* competent cells

Transformation was carried out in 96-well deep-well plates (2.3 ml, Axygen P-DW-20-C) in the cold room. Gibson mix (5 µl per well) from the 384-well plate was transferred into four 96-well plates and spun down to the bottom of each well. Competent cells (50 µl per well) (prepared in-house from NEB stable competent *E. coli* cells, New England Biolabs, C3040I) were dispensed and mixed twice with the Gibson mix. The plates were then kept immersed in ice for 30 min. Heat shock was performed for 30 s at 42 °C by placing the plate in a water bath. Plates were placed back on ice for 5 min. Homemade SOC medium (300 µl) (0.5% yeast extract, 2% tryptone, 10 mM NaCl, 2.5 mM KCl, 10 mM MgCl₂, 10 mM MgSO₄ and 20 mM glucose) was added to the plate and incubated for 1 h at 37 °C under shaking at 900 rpm using a thermo-shaker. Since the NEB stable competent *E. coli* cells harbour the tetracycline resistance and to minimize any potential contamination, after incubation with SOC medium, 900 µl (per well) of Terrific Broth (TB) medium (https://openwetware.org/wiki/Terrific_Broth) containing 15 µg ml⁻¹ trimethoprim and 15 µg ml⁻¹ tetracycline was added to the

transformation mix and incubated at 30 °C under shaking at 900 rpm for 40–48 h. Bacteria were stored in 16.7% (v/v) glycerol/medium in 96-well plates (300 μl final storage volume) and 384-well plates (150 μl final storage volume) at −80 °C.

### Magnetic bead-based 96-well plasmid miniprep of ALPA cloning product

Fifty microlitres of the Gibson assembly product-transformed bacteria was transferred into 1.2 ml of TB medium (with 15 μg ml⁻¹ trimethoprim and 15 μg ml⁻¹ tetracycline in a 96-well deep-well plate) immediately before the storage of the bacteria and grown at 30 °C at 900 rpm for 40–48 h. The bacteria were then subjected to in-house magnetic bead-based plasmid miniprep procedures, which were adapted from the canonical plasmid miniprep protocols[65]. Bacteria were pelleted at 4,000 rpm for 10 min and resuspended in 200 μl of P1 buffer (50 mM glucose, 10 mM EDTA and 25 mM Tris (pH 8.0)), and subsequently lysed in 200 μl of P2 buffer (0.2 M NaOH and 1% SDS (w/v)), and the lysis mixture was neutralized in 200 μl of P3 buffer (3 M KOAc, pH 6.0) and subjected to centrifugation at 4,000 rpm for 10 min at 4 °C. The supernatant (400 μl) was transferred into a new deep-well plate and 1,000 μl of cold absolute ethanol was added, mixed and centrifuged at 4,000 rpm for 10 min at 4 °C. The supernatant was discarded and 50 μl of ddH$_2$O was added to the plasmid pellet and mixed to dissolve the plasmids. Beads buffer (75 μl) (2.5 M NaCl, 10 mM Tris base, 1 mM EDTA, 3.36 mM HCl, 20% (w/v) PEG8000 and 0.05% (w/v) Tween 20) and 50 μl of SpeedBeads magnetic carboxylate modified particles (GE Healthcare 65152105050250, 1:50 dilution in beads buffer) were added to the plasmids, mixed and incubated for 5 min on an in-house design and 3D printed magnetic rack (blueprint of the design, resin for printing and magnets used are available here: https://www.dropbox.com/scl/fi/xx8uung6pte5d4086alr2/3D-printing-software-and-resin-used.zip?rlkey=umij8xhnb13tgbrwwykkyumtq&e = 2&dl=0) to separate the beads from the supernatant. The beads were then washed twice with 70% ethanol and dried in a water bath (65 °C). Plasmid DNA was then eluted by 150 μl of sterile Tris-EDTA buffer (1 mM EDTA and 10 mM Tris-HCl (pH 8.0)) from the beads at 65 °C for 10 min and transferred to a new low-profile 96-well plate. To ensure that the full cloning procedure was correct, two wells of plasmids from each 96-well plate were subjected to Sanger sequencing.

### In silico design of the qgRNA libraries

Guide RNAs from the Calabrese[14] and hCRISPRa-v2 (ref. 28) libraries and the TKOv3 (ref. 26) and Brunello[14,66] libraries were adopted and formed the basis for the T.gonfio and T.spiezzo libraries. sgRNAs from the CRISPick tool (https://portals.broadinstitute.org/gppx/crispick/public, last accessed in April 2020) were also incorporated to ensure optimal coverage of difficult-to-target and newly annotated genes. We adopted the alternative TSS definitions from the hCRISPRa-v2 library[28]. Additional TSSs were targeted by a separate set of qgRNAs if the FANTOM5 scores indicated conspicuous transcriptional activity and if they were spaced more than 1 kb apart from the primary TSS.

We avoided sgRNAs containing genetic polymorphisms with frequencies greater than 0.1% in both the 20-nucleotide target sequence and the 2 guanosine nucleotides of the NGG PAM. Variant frequencies were derived from the Kaviar database[67], which includes curated genomic data on single nucleotide variants, indels and complex variants from over 77,000 individuals (including over 13,000 whole genomes, only variants seen more than 3 times, version 160204-hg38, last accessed on 7 August 2019). To select a four-guide combination with minimal off-target effects, the specificity scores for each sgRNA from libraries were calculated using the GuideScan[68] tool (version 2018-05-16). For each guide, potential off-target sites were weighted by their CFD (cutting frequency determination) scores[66], and CFD scores were aggregated into a single score using the formula $1/(1 + \text{sum of CFD scores from all off-target sites})$[69]. Because the GuideScan database did

not contain all sgRNAs, the aggregate specificity scores for the remaining sgRNAs were calculated using the CRISPOR[70,71] tool (version 4.97).

The potential qgRNA combinations were then ranked, using the criteria as follows: (1) maximize the number of sgRNAs (from zero to four) that fulfil the following minimal requirements—the sgRNA can be mapped to a defined genomic location in the reference genome with an N(GG) PAM; there are no overlaps with frequent genetic polymorphisms (>0.1%); the specificity score (considering up to 3 mismatches) is at least 0.2; and for T.spiezzo only, the guide conforms to the criteria of ref. 72; (2) maximize the number of sgRNAs with exactly one perfect match location in the reference genome; (3) minimize the number of overlaps between two neighbouring sgRNAs spaced fewer than 50 bp apart; (4) minimize the number of sgRNAs derived from the CRISPick sgRNA Designer tool, rather than the previously published libraries; (5) for T.gonfio, minimize the number of sgRNAs derived from the 'supplemental 5' rather than 'top 5' sgRNAs for the hCRISPRa-v2 library, and for T.spiezzo, minimize the number of CRISPick-derived sgRNAs ranked outside the top 10; and (6) maximize the aggregate specificity score from all 4 guides. (7) Exclude any combination with two or more sgRNAs sharing identical subsequences of 8 or more base pairs, minimizing potential DNA recombination during Gibson assembly. The highest-ranked four-guide combination was chosen.

To facilitate focused screens of a subset of the genome, we divided the entire set of protein-coding genes into mutually exclusive sublibraries. The TFs, G protein-coupled receptors and secretome sublibraries were based on recent publications[33,73,74], and the other seven thematic sublibraries were adopted largely from the hCRISPRa-v2 library[28], except that the sublibraries were updated to incorporate a small number of additional transmembrane receptors, transporters, kinases and phosphates, using GO terms (exported from BioMart[75] on 25 March 2020) and a list of membrane proteins provided by the Human Protein Atlas project[34] (last accessed on 11 March 2020). The detailed methods for the sgRNA design of the libraries and calculation of potential off-target sites of the sgRNAs are provided in Supplementary Information.

### In silico comparison of the CRISPR libraries

To compare in silico characteristics of existing libraries and the qgRNA libraries, the top four guides for each gene were selected. Whereas the Brunello[14,66] and TKOv3 (ref. 26) libraries were designed to contain four sgRNAs per gene, the Calabrese library[14] was divided by the authors into Set A and Set B, each containing three sgRNAs per gene. To define the top four sgRNAs, the sgRNAs from Set A were supplemented with a randomly selected sgRNA from Set B. For the hCRISPRa-v2 (ref. 28) and CRISPick tool, the four highest-ranked sgRNAs were chosen. Since the libraries differed in the genes that they covered, and different genes vary in the availability of potential sgRNAs with high predicted activity and specificity, only genes present in all libraries were used for benchmarking.

Furthermore, for genes for which the T.gonfio and hCRISPRa-v2 libraries included more than one TSS, only the sgRNAs targeting the main TSS were included, defined as the TSS with the highest score in the FANTOM5 dataset, or—if data were unavailable for that gene—the most upstream TSS. To compare the expected number of sgRNA binding sites affected by genetic polymorphisms, the frequency of the most common polymorphism overlapping each sgRNA was considered. This is a conservative estimate, since SNPs with frequencies below 0.1% were excluded. In the case of multiple SNPs overlapping with an sgRNA, only the most frequent was considered. Owing to linkage disequilibrium between SNPs affecting the same sgRNA, a precise estimation of the total probability of overlaps with polymorphisms would require access to the individual-level sequencing data underlying the SNP databases.

### SMRT long-read sequencing of the libraries

Library plasmids were diluted to a concentration of 1.3 ng μl⁻¹, and 0.5 μl of each diluted plasmid was used as template for per 10 μl PCR reaction

(16 cycles) using the high-fidelity Phusion DNA polymerase with barcoded primers that uniquely identified each plasmid on a 384-well plate (primer information is provided in Supplementary Information). The amplified region (2,225 bp) encompassed the entire qgRNA expression cassette (containing all four promoter, guide RNA and tracrRNA sequences, as well as the trimethoprim resistance element) and was flanked by two 10 bp paired barcode sequences. Then PCR amplicons from each 384-well plate were pooled down and purified by magnetic beads. Next, individual purified pools were uniquely barcoded, purified and further pooled with near-equimolar concentrations of DNA amplicons and subjected to SMRT sequencing using the PacBio Sequel IIe instrument.

Consensus reads were generated from raw subreads with SMRT Link software (Pacific Biosystems) using standard settings. Barcode demultiplexing of the consensus reads was performed using SMRT Link software (version 9.0.0.92188) with two consecutive calls to the 'lima' tool for plate and well barcodes, respectively. Only consensus reads that retained at least 80% of both barcode sequences with a quality score of at least 25 were included in the analysis. This was done by specifying the following non-default parameters to the SMRT Link 'lima' tool: '–min-score 25 –min-ref-span 0.8 –min-scoring-regions 2'. For plate barcodes, the '–same' parameter was given to indicate that the same barcode was present on either side of the read, whereas the '–different' parameter was given for well barcodes, which were flanked by one row and one column barcode. The data file containing the sequences and metadata for all consensus reads was converted from .bam to .sam format using SAMtools (version 1.9). Additional filtering for read and barcode quality was performed using custom R scripts. For each consensus read, the number of full passes (number of complete subreads) and the estimated read quality were extracted from the metadata of the .sam file and integrated with data from the two '.lima.report' files generated by demultiplexing the plate and well barcodes.

The 'ScoreCombined' and 'ScoreLead' values from the '.lima. report' file had to reach a minimum of 60 and 30, respectively. We further filtered by read quality of 7 full passes, a read quality of 0.9999 and a mean per-base Phred quality score of 85 (out of a maximum achievable score of 93). For 99.9% of plasmids in our libraries, at least one consensus read fulfilled these criteria. If there were no consensus reads for a specific well, we relaxed the requirements to include reads with at least three full passes and a read quality of at least 0.99; these criteria were used for 15 plasmids. Twenty-three plasmids in our libraries (0.05%) were not represented by any reads.

To quantify the percentage of correct guide RNA sequences and to identify well-to-well contaminations, each read was searched for the combined sgRNA and tracrRNA sequences in the forward and reverse directions, and all perfect matches were counted. To further characterize incorrect sequences, each consensus read was aligned to the correct barcoded reference sequence for that well. Alignment was done with the 'pairwiseAlignment' function of the Biostrings package from the Bioconductor project (version 2.54.0) with a gap opening penalty of 30. Once all reads were aligned to a unified set of coordinates, the sequences corresponding to each combined sgRNA and tracrRNA region were extracted. Each region was classified as (a) entirely correct, (b) a contamination (if it was a perfect match for an sgRNA sequence from another plasmid), (c) a large deletion (if more than 50% of the aligned sequence was composed of gaps) or (d) a mutation (all other alterations). For each well, aggregate statistics were computed on the number of correct sgRNA modules, cross-well and cross-plate contaminations, and the number of deletions affecting each element of the qgRNA expression cassette.

### Arrayed activation screen for PrP^C regulating TFs

The screen was performed as previously described[39]. Five thousand cells (per well) of U-251 MG stably expressing dCas9-VPR were seeded in 30 µl of medium into white 384-well cell culture plates (Greiner Bio-One 781080). The plates were incubated in a rotating tower incubator (LiCONiC StoreX STX). Twenty-four hours later, cells were transduced with lentiviruses containing the qgRNA vector targeting each TF at an MOI of 3. Each plate contained 14 wells with NT and another 14 wells with *PRNP*-targeting controls. Plates were further incubated for 4 days. Experiments were performed in triplicate; one replicate was used to determine cell viability using CellTiter-Glo (Promega) according to the manuals using the EnVision plate reader (PerkinElmer). The other two replicate plates were used to assess PrP^C levels by the TR-FRET method[39]. Cell culture medium was removed by inverting the plates, and cells were lysed in 10 µl of lysis buffer (0.5% Na-deoxycholate (Sigma-Aldrich), 0.5% Triton X (Sigma-Aldrich)), supplemented with EDTA-free cOmplete Mini Protease Inhibitors (Roche) and 0.5% BSA (Merck). Following lysis, plates were incubated on a plate shaker (Eppendorf ThermoMixer Comfort) for 10 min (4 °C, 400 rpm shaking conditions) before centrifugation at 1,000 × g for 1 min and incubated at 4 °C for 2 additional hours. Following incubation, plates were centrifuged once more under the same conditions mentioned above and 5 µl of each FRET antibody pair was added (2.5 nM final concentration for the donor and 5 nM for the acceptor, diluted in 1× Lance buffer (PerkinElmer)). POM1 (binding to amino acid residue a.a. 144–152) and POM2 (binding to a.a. 43–92) (ref. 76), targeting different epitopes of PrP^C, were coupled to a FRET donor, europium (EU) and a FRET acceptor, allophycocyanin (APC), respectively, following previously reported protocols[77]. Plates were centrifuged once more and incubated overnight at 4 °C. Then, TR-FRET measurements were read out using previously reported parameters[77] on an EnVision multimode plate reader (PerkinElmer). The first and last columns of each 384-well plate were reserved for blanks (wells containing only one of the antibodies, or buffer only), which were used to calculate net FRET values and for background subtraction as previously described[77].

For plate-wise quality control, the separation between positive (*PRNP*) and NT controls was assessed using the $Z'$ factor $Z' = 1 - \frac{3 \times (\mathrm{SD_{pos}} + \mathrm{SD_{NT}})}{|\mathrm{mean_{pos}} - \mathrm{mean_{NT}}|}$, where $\mathrm{SD_{pos}}$ and $\mathrm{SD_{NT}}$ denote the standard deviation of positive and NT controls, respectively[78]. To obtain logarithmized and normalized PrP^C expression values ($\log_2$ FC values), raw values were $\log_2$ transformed and normalized by subtracting the median expression value of all genes on that plate. Mixed-moment estimates of sample SSMD values were computed using the formula $\frac{\bar{d}}{\sqrt{w_i s_i^2 + w_0 s_0^2}}$, and $t$ values were calculated as $\frac{\bar{d}}{\sqrt{(w_i s_i^2 + w_0 s_0^2)/n}}$ (ref. 41). In our case, $\bar{d}$ was the mean of normalized expression values of the two replicates for each plasmid. The weights ($w_i$ and $w_0$) were both set to 0.5; the variable $s_i^2$ referred to the variance of the two replicates for each plasmid, and $s_0^2$ was the median of all variances for pairs of NT control wells on replicate plates. Two-sided *P* values were derived from sample $t$ values using the Student's $t$-distribution with one degree of freedom.

Primary hits were further repeated with the same FRET-based assay in 384-well plates with 5 technical repeats. Then the respective top 10 hits of PrP^C upregulators and downregulators were individually confirmed in 6-well plates in U-251MG dCas9-VPR cells with their corresponding qgRNA lentiviruses 4 days post-viral transduction and 1 µg ml⁻¹ puromycin selection via western blotting. The POM2 primary antibody against PrP^C was used for the assay. Vinculin was used as a loading control. The levels of PrP^C were quantified by ImageJ and normalized to vinculin.

### Pooled genome-wide CRISPR knockout autophagy screens

The H4-Cas9-GFP-SQSTM1 and H4-Cas9 cells generated in the previous study[43] were used and maintained in DMEM + 10% FBS, 1% L-glutamine and 1% penicillin/streptomycin at 37 °C with 5% $CO_2$. sgRNA plasmids of pooled T.spiezzo and an optimized Brunello library that covers 18,360 genes (5 sgRNAs per gene whenever possible, split into two sgRNA sub-pools named CHIP1 to CHIP6)[79] were amplified and packaged into lentiviral particles using HEK293T cells. H4-Cas9-GFP-SQSTM1 cells

were infected at an MOI of 0.4 with at least 300 coverage of each sgRNA during the screen. Twenty-four hours post-infection, cells were selected with puromycin (2 µg ml⁻¹, GIBCO) for 3 days and then maintained in culture and split as needed to ensure that confluence did not exceed 90% for a further 7 days. Then cells were collected and resuspended at a density of 30 million cells per ml, and live and single cells were sorted (BD ARIA III) from the lower GFP quartile (GFP$^{low}$) or from the upper GFP quartile (GFP$^{high}$) and subsequently fixed in 4% PFA in PBS. For each screen, 20–25 million cells were isolated by FACS in the GFP$^{low}$ and GFP$^{high}$ category. Twenty-five million unsorted cells were also collected as an input sample.

For all samples, genomic DNA (gDNA) was isolated using phenol chloroform extraction. In short, cells were resuspended in 5 ml TNES (10 mM Tris-Cl pH 8.0, 100 mM NaCl, 1 mM EDTA, 1% SDS) and incubated overnight at 65 °C to reverse PFA crosslinks. After allowing the samples to cool, samples were incubated with 100 µl RNase A (QIAGEN) for 30 min at 37 °C, followed by addition of 100 µl of proteinase K (QIAGEN) for 1 h at 45 °C. PCIA (5 ml) (phenol/chloroform/isoamyl alcohol pH 8) (ThermoFisher) were added and samples were vortexed and spun at a max speed for 2 min, and the aqueous phase was transferred to 5 ml of PCIA. Samples were vortexed again and spun at a max speed for 2 min. The aqueous phase was transferred to 4.5 ml of chloroform. A third time, samples were vortexed and spun at a max speed for 2 min, and the aqueous phase was transferred to 400 µl of 3 M Na-acetate pH 5.2. Later, 10 ml of 100% EtOH was added, samples were mixed, and DNA was precipitated for 1 h on ice, followed by spinning at a max speed for 10 min. EtOH was decanted and pellets were washed with 10 ml of 70% EtOH. Finally, samples were spun at a max speed for 10 min, and pellets were air-dried and resuspended in 1 ml of nuclease-free water.

For gDNAs of GFP$^{high}$, GFP$^{low}$ and unsorted samples from the T.spiezzo pooled screen, we developed an Illumina sequencing strategy. In brief, sgRNA2 and sgRNA3 from the qgRNA library were amplified (595 bp) by PCR using the NEB Q5 DNA polymerase (New England Biolabs) with a universal P7 primer and individual P5 primer, each with a unique index. We designed a custom sequencing primer (Read-1 primer) for read 1 detecting the sequence of sgRNA2 and designed a custom index sequencing primer (Index-1 sequencing primer) detecting the sequence of sgRNA3, and an index sequencing primer (Index-2 sequencing primer) for demultiplexing samples. For each PCR, the amplification parameters are 98 °C for 30 s, 98 °C for 10 s, 55 °C for 30 s, 72 °C for 120 s and 72 °C for 120 s. We used 30–60 µg of gDNA from each sample and performed 16–31 individual PCR reactions, each with 2 µg gDNA as the input, which were pooled and reamplified with the same primers. An illustration of the sequencing strategy and the information of primers are provided in Supplementary Information. The sequencing library preparations for the Brunello and Cellecta library screens were performed according to the published protocols[14,66]. All library samples were sequenced on an Illumina NovaSeq 6000 using S1 flowcells and the XP workflow.

### Pooled autophagy screen data analysis

All FASTQ files were aligned using bowtie2 with the following parameters: –L10 –N0 –iS,1,2; the seed length is 10; zero mismatches allowed in seed alignment; the interval between seeds is 1 + 2 * sqrt (read length), which is smaller than the default (–iS,1,2.5). This is adapted to the short sequences (20 nt) that are being aligned and provided better results than standard bowtie2 settings (which are for longer reads of around 50 nt). For T.spiezzo, two bowtie2 indices were built, one for sgRNA2 and one for sgRNA3. For the Brunello library, two bowtie2 indices were built, one for CHIP1–3 and one for CHIP4–6 of the Brunello library. For the Cellecta library data[43], historical counts from the original screen were used, obtained with bowtie1.

All alignments were written to SAM files, which were then further processed with samtools (statistical analysis, filtering and counting). Counts were obtained by counting the reference names, that is, sgRNA

annotations from SAM files. For T.spiezzo, each sample was processed to most accurately reflect the Illumina-based sequencing protocol that was used: read 1 of each sample should only be from the sgRNA2, that is, the reference sequence should contain an sg2 in its name; reference sequences of the second read should contain an sg3 in their name; owing to the alignments using two separate indices for the FASTQ files from reads 1 and 2, all alignments from reads 1 and 2 must contain sgRNA2 and sgRNA3 sequences, respectively. The intersection of sgRNA2 first reads and sgRNA3 second reads provides the names of all valid reads across the FASTQ files for read 1 and read 2. FASTQ files are filtered to these common read names (using the -N option to samtools) and unaligned reads are discarded (-F4 option to samtools). Because of this filtering, the number of reads from the respective SAM files is the same. In addition, the two corresponding reads are then checked for targeting the same gene and discarded if this is not the case. The reported valid guide rate takes into account correct sequences and the cohesiveness of both sgRNA sequences.

FCs were estimated using edgeR. These were obtained for the comparison of the GFP$^{high}$ versus the GFP$^{low}$ samples in all three screens used for analysis (T.spiezzo, Brunello and Cellecta). Enrichment analysis was performed using a thresholded approach (overrepresentation analysis (ORA)). The top or bottom 200 genes with the highest or lowest FC were used for ORA. Gene sets from the biological process branch of GO were used as reference sets. Enrichment calculations were done using the clusterProfiler package in R. Box plots for enriched gene sets and/or gene sets related to autophagy were prepared using the absolute FC. This was done to be able to quantify the effect of the screening library via analysis of variance (ANOVA) modelling. In short, the absolute FC was modelled as a linear function of gene and screening library ((abs(logFC) - gene + screen) R formula), and the significance of the screen coefficient was determined using ANOVA (aov in R).

### Pooled autophagy screen hits validation

Among the top hits uniquely identified by the T.spiezzo pooled library, ten potential novel regulators together with six known autophagy regulators were selected for further validation. Lentiviruses of the corresponding qgRNA plasmids targeting the 16 genes or NT controls were individually packaged, titrated and transduced to H4-Cas9-GFP-SQSTM1 cells at an MOI of around 0.3 in 24-well plates. Then cells were split and selected in 2 µg ml⁻¹ puromycin medium for 3 days and further cultured in normal medium without antibiotic selection for a week in duplicates. One replicate of cells was analysed by flow cytometry to quantify the log$_2$ FC of cell numbers in GFP$^{high}$ versus GFP$^{low}$ populations, as in the primary pooled screen. The other replicate of cells was imaged under a fluorescent microscope to analyse the distribution of GFP-SQSTM1 puncta. NT-treated cells were used as controls for both conditions. The area of GFP-SQSTM1 puncta was determined with CellProfiler, and the percentage of cells with GFP-SQSTM1 puncta was analysed with ImageJ (NIH).

To further confirm the potential novel regulators of autophagy, H4-Cas9 cells were individually transduced with qgRNA lentiviruses against these genes or NT and cultured the same as the GFP-SQSTM1 assay in 10 cm dishes in duplicate. Six hours before collection of samples, one replicate of cells was treated with vehicle (medium) and one replicate was treated with a final concentration of 100 µM ChQ. At the endpoint, cells were washed twice with cold PBS before adding ice-cold 1× RIPA buffer (Cell Signaling) with protease (cOmplete, Roche) and phosphatase inhibitors (PhosSTOP, Roche) directly to the well. After cell scraping, the suspension was collected in a tube and incubated on ice for 30 min before centrifugation at a maximum speed for 10 min. The supernatant containing cell lysates was collected in a new tube and the protein concentration assessed through the BCA assay (Pierce). The total protein (5 µg) was separated in a 12% Bis-Tris gel with MOPS running buffer and subsequently transferred to a PVDF membrane. The following antibodies were used for immunoblotting: LC3b (Cell

Signaling #3868) and GAPDH (Cell Signaling #2118). The experiments were repeated twice. The intensities of LC3-II were quantified with ImageJ and normalized to GAPDH intensities.

For the YFP-LC3 reporter assay, H4-Cas9 cells were cultured following a similar protocol as used for the GFP-SQSTM1 assay. Forty-eight to sixty hours before the experiment, 2,000 gene-edited or NT-treated H4-Cas9 cells were seeded into an 8-well chamber, and 6 h later (or the next day), they were transduced with YFP-LC3 lentiviruses at an MOI of 3. Two days later, each well of cells was treated with vehicle or 100 μM of ChQ for 6 h, before cells were imaged with a Leica LAS X confocal imaging system. The puncta area of YFP-LC3 was analysed in a similar manner to that of the GFP-SQSTM1 experiments.

### CRISPRoff tests with individual qgRNA plasmids

Individual qgRNA plasmids for CRISPRoff were assessed in HEK293T cells using plasmid transient transfection. A pool of three single sgRNAs (150 ng) or individual qgRNA plasmids (150 ng) were co-transfected with the CRISPRoff (300 ng) or CRISPRoff-D3A mutant (300 ng) plasmids into HEK293T cells in 24-well plates with Lipofectamine 3000 once the cells reached 80–90% confluency. The second day, cells were split into 1 μg ml$^{-1}$ puromycin containing medium and selected for 3 days to eliminate non-transfected cells. Then cells were maintained in normal growth medium without antibiotic selection for 1 week. Cells were trypsinized and single live-cell solutions were stained using the same method as described for the gene ablation efficiency assay with the corresponding fluorophore-conjugated primary antibodies and analysed with flow cytometry. Gene ablation efficiencies of qgRNA plasmids from the T.spiezzo library were determined as described above. NT sgRNA-treated cells were used as controls.

### Genome-wide CRISPRoff screens

The T.gonfio pooled library was amplified with Lucigen E. cloni 10G Elite electrocompetent cells through electroporation. In brief, 150 ng plasmid of the library was added to 200 μl competent cells, followed by 4 aliquots with 50 μl loaded into each electroporation cuvette (0.1 cm gap) for further electroporation using the Gene Pulser Electroporation System (voltage in 1,600 V, capacitance in 25 μF and resistance in 200 Ω). Electroporated bacteria were mixed with chilled recovery medium, followed by shaking at 250 rpm for 1 h at 37 °C. Then all transformations were pooled and distributed in 4 pre-warmed 24.5 cm$^2$ bioassay plates (trimethoprim resistance) and grown at 30 °C for 16 h. Finally, bacteria in plates were collected for plasmid extraction with the QIAGEN Endo-Free Plasmid Maxi Kit (catalogue number 12362). Amplified sgRNA libraries were packaged into lentiviruses with HEK293T cells followed by titration with flow cytometry analysis as described above. For the dual-sgRNA CRISPRoff library, DH5α competent cells ($3 \times 10^8$ cfu μg$^{-1}$ in 75 μl) were used to transform 100 ng plasmids to amplify the library as previously described[58].

The CRISPRoff genome-wide dropout screen was performed as described[58]. In brief, 60 million HEK293T cells for each library were seeded in 2× T300 culturing flasks, followed by virus transduction at an MOI of 0.3 with a coverage of about 1,000 cells per plasmid 6 h post-seeding. Two days later, the cells were passaged with a 1:3 ratio and maintained for 4 days in the presence of puromycin, during which the percentage of GFP-positive (for dual-sgRNA library) or BFP-positive (for T.gonfio 4-sgRNA library) cells was monitored with flow cytometry analysis until a proportion of 90% was achieved. Next, 60 million cells were seeded in 2× T300 flasks without puromycin. About 24 h later, cells with a confluency of about 80% (T0 time point) in each flask were transfected with 57 μg of the CRISPRoff-mScarletI plasmid using Lipofectamine 3000. On the second day, transfected cells were passaged at a 1:2 ratio for cell sorting. One day later, 27 million cells double-positive for mScarletI-GFP (for the CRISPRoff library screen) or mScarletI-TagBFP (for the T.gonfio library screen)

were sorted for each library, seeded in T300 flasks and maintained for 10–12 cell passages (T10 time point). Sixty million cells for each time point were collected for gDNA extraction and subsequent sequencing analyses.

PCR amplicons of the screen with the CRISPRoff library were prepared according to established methods[58]. PCR amplicons of the T.gonfio library screen were prepared the same as the pooled screen for autophagy using the T.spiezzo pooled library. For each sample of the CRISPRoff screens, 130 μg of gDNA was used in 65 × 50 μl PCR reactions (2 μg gDNA per 50 μl PCR reaction), and all PCR products from the same sample were pooled and purified. All PCR amplicons from the two libraries were sequenced on an Illumina NextSeq500 device at the Functional Genomics Center Zurich.

### CRISPRoff screen data analysis

Sequences were mapped to sgRNAs from the reference CRISPR libraries. If no perfect match was found, we relaxed the search criteria to include sgRNAs with a maximum edit distance of 1 (a single-nucleotide mismatch, insertion or deletion was tolerated). Mapped sgRNA sequences were allocated to plasmids. For a few genes in the T.gonfio and dual-sgRNA CRISPRoff libraries, the same sgRNAs were included for multiple plasmids. The inclusion of unspecific sgRNAs is sometimes inevitable when targeting genes with closely related paralogues. In such cases, the information from neighbouring mapped sgRNA sequences was used for disambiguation, whenever possible. Sometimes, however, sgRNAs had to be assigned to multiple plasmids. When computing plasmid counts, these were weighted by the number of possible plasmids, resulting in fractional counts. This was separate from the handling of reads with template switches, which were excluded from further analysis.

For the normalization of plasmid counts, only reads with two mapped sgRNAs without a template switch were included. For each screen, raw read counts were adjusted to account for any differences in the number of reads per sample using the median ratio method[80]. Initially, the geometric mean across samples was calculated for each plasmid. Next, counts were divided by the geometric mean across samples to produce a matrix of count ratios. A sample-specific size factor was computed, defined as the median of count ratios across all plasmids. Finally, raw counts were divided by the sample-specific size factor to produce normalized counts. Because the geometric mean is defined as the antilog of the sum of log values divided by the number of samples, and the logarithm of zero is undefined, only plasmids with non-zero counts for all samples were used for estimating the size factor. Sample size factors ranged from 0.81 to 1.21. Both the T.gonfio and dual-sgRNA CRISPRoff libraries contained multiple plasmids for genes with major alternative TSSs. To ensure a fair comparison between libraries, only one plasmid per gene was included, and guide RNAs targeting the principal TSS were preferred. The primary TSS was defined by the activity score from the FANTOM5 project[81]. The sgRNA sequences of the dual-sgRNA CRISPRoff library were aligned to the hg38 reference genome and re-annotated with additional information, including Entrez gene identifiers and their location relative to the TSS. To define the gene sets used for the comparison, we downloaded the 'common_essentials.csv' and 'nonessentials.csv' data files from the Public 20Q2 release of the Cancer Dependency Map (DepMap, https://depmap.org/portal/download/all/).

The robust estimate of the SSMD[41,82] was computed as a measure of the separation between essential and non-essential genes as follows: $\mathrm{SSMD}* = \frac{\mathrm{median}_{ne} - \mathrm{median}_e}{1.4826\sqrt{\mathrm{MAD}_{ne} + \mathrm{MAD}_e}}$, where $\mathrm{MAD}_{ne}$ and $\mathrm{MAD}_e$ denote the median absolute deviation of non-essential and essential genes, respectively.

### Reporting summary

Further information on research design is available in the Nature Portfolio Reporting Summary linked to this article.

## Data availability

The complete sgRNA sequences, metadata and annotation for our T.spiezzo and T.gonfio libraries are included as Source Data File 1. The count and fold-change data from the CRISPR screens are available as Source Data Files 2–4. All additional experimental and sequencing data are available from the corresponding authors on reasonable request. Source data are provided with this paper.

## Code availability

For the construction of our arrayed CRISPR libraries, we wrote a custom code pipeline, available at https://github.com/Lukas-1/CRISPR_4sgRNA. All code is based on the R statistical programming environment, version 3.6.3, and Bioconductor suite, version 3.10.0. For the quality-control analysis with single-molecule sequencing of plasmids and for the arrayed and pooled CRISPRoff screens, the analysis was performed with custom code available at https://github.com/Lukas-1/CRISPR_4sgRNA/tree/master/6)%20Individual%20experiments using the R statistical programming environment, version 3.6.3.

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

## Acknowledgements

We thank P. Hsu (University of California, Berkeley), J. Doench (the Broad Institute of MIT and Harvard) and J. Corn (ETH Zurich) for their help and suggestions; A. Bradley (University of Cambridge) for sharing the Lenti-PB plasmid; E. Zuo (Agricultural Genomics Institute at Shenzhen Chinese Academy of Agricultural Sciences) for sharing the transposase plasmid; D. M. Sabatini for suggestions on autophagy-hits validation; and L. A. Gilbert, G. C. Pommier (University of California, San Francisco) and J. S. Weissman (Massachusetts Institute of Technology) for providing reagents and advice for experiments on targeted epigenetic silencing (CRISPRoff). We thank K. Maggi, S. Sellitto and G. Ehmer for technical assistance, A. Bratus-Neuenschwander and W. Qi, and S. Kreutzer and her team at the Functional Genomics Center Zurich (FGCZ) for support with SMRT sequencing and Illumina sequencing, respectively, and M. Wickert at the Cytometry Facility of University of Zurich for technical assistance, as well as M. Avar and D. Heinzer for supporting C.T. with the PrP$^C$

FRET assays. A. Aguzzi discloses support for the research described in this study from institutional core funding by the University of Zurich and the University Hospital of Zurich, and grants from the Nomis Foundation, the Swiss National Research Foundation (grant ID 179040 and grant ID 207872, Sinergia grant ID 183563), the Swiss Personalized Health Network (SPHN, 2017DRI17), an Advanced Grant of the European Research Council (ERC Prion2020 no. 670958), the HMZ ImmunoTarget grant, the Human Frontiers Science Program (grant ID RGP0001/2022), the Michael J. Fox Foundation (grant ID MJFF-022156), Swissuniversities (CRISPR4ALL) and a donation from the estate of Hans Salvisberg. J.-A.Y. discloses support for the research described in this study from the postdoc grant Forschungskredit from University of Zurich and the Career Development Awards grant of the Synapsis Foundation—Alzheimer Research Switzerland ARS (grant ID 2021-CDA02). A.D., J.T., S.R.-N. and P.H. disclose support for the research described in this study from the DZNE. Y.W. and L.Y. were supported by China Scholarship Council. A. Armani is the recipient of Marie Curie postdoctoral fellowship (MSCA GF, 101033310).

## Author contributions

J.-A.Y. designed, supervised and coordinated the research, invented the ALPA cloning method, set up and performed most of the experiments, and wrote the paper. L.F. developed the qgRNA selection algorithm with input from J.-A.Y., analysed the features of the libraries, the SMRT sequencing and the TF PrP$^C$ and CRISPRoff screens, and wrote the paper. M.C.S. performed the benchmarking experiment of qgRNA and the autophagy screens with the assistance of B.K., and wrote the paper. T.L. performed the qgRNA knockout efficiency and RT-qPCR assays, and the CRISPRoff screens. C.T. performed the TF PrP$^C$ screen and validated the hits with help of J.-A.Y. and T.L., and wrote the paper. A.D. performed gene activation experiment in iNeurons and packaged the T.gonfio TF lentiviruses with J.T. and S.R.-N. A.S. prepared reagents and performed three-fragment PCRs and Gibson assembly reactions of the ALPA cloning of the libraries. Y.W. and L.Y. performed the competent cell production and transformation, bacterial glycerol stock and miniprepped of the libraries. D.L.V. prepared the PCR amplicons for SMRT sequencing of the libraries. J.G. performed the knockout efficiency and CRISPRoff efficiency assays, and part of the CRISPRoff screens. E.D.C. produced the Taq DNA ligase for Gibson assembly reaction. K.G. performed the iNeuron RT-qPCR assay and assisted ALPA cloning. A. Armani performed the LC3-II western blot and analyses. E.J.O., J.J. and J.H. performed the Illumina sequencing of the autophagy screens, and F.N. and M.P. analysed the results. V.B. performed the cell–cell heterogeneity and the qgRNA toxicity assays, and virus titration in 384-well plates. R.R. performed partially plasmid/lentiviral preparation and RT-qPCR assays. F.B. helped with autophagy hits validation and virus titration. M.S.B. performed the puncta area analysis of GFP-SQSTM1 and YFP-LC3. S.B. assisted the assay for gene activation efficiency. M.L. assisted the flow cytometry analyses. S.H. supported the production of Taq DNA ligase and PrPc-detecting antibodies. M.K. designed sgRNAs for the first trial ALPA cloning and offered feedback for the project. L.P. and D.H. supervised the research. P.H. supervised the research and planning and the execution of the experiments at DZNE. A. Aguzzi conceived the primary idea of generating arrayed libraries, supervised the planning and the execution of the experiments, coordinated the activities of the research team, and wrote the paper with input from all authors.

## Competing interests

J.-A.Y., L.F. and A. Aguzzi are listed as inventors on a patent (Molecular Cloning Method and Vector Therefore, WO/2023/089153) owned by the University of Zurich, whose claims are supported by the present study. L.P. has ownership interest in Sagimet Biosciences, Apricot Therapeutics and Element Biosciences. M.K. is an inventor on US Patent 11,254,933 related to CRISPRi and CRISPRa screening, serves on the Scientific Advisory Boards of Engine Biosciences, Casma Therapeutics, Cajal Neuroscience, Alector and Montara Therapeutics, and is an advisor to Modulo Bio and Recursion Therapeutics. All authors with the affiliation 'Novartis Institutes for Biomedical Research' are employees of Novartis Pharma AG and may own stock in the company. The other authors declare no competing interests.

## Additional information

**Extended data** is available for this paper at https://doi.org/10.1038/s41551-024-01278-4.

**Correspondence and requests for materials** should be addressed to Jiang-An Yin or Adriano Aguzzi.

Jiang-An Yin [1,8] ✉, Lukas Frick[1,8], Manuel C. Scheidmann[2], Tingting Liu[1], Chiara Trevisan [1], Ashutosh Dhingra[3], Anna Spinelli[1], Yancheng Wu [1], Longping Yao[1], Dalila Laura Vena [1], Britta Knapp[2], Jingjing Guo[1], Elena De Cecco[1], Kathi Ging[1], Andrea Armani [1,4], Edward J. Oakeley [2], Florian Nigsch [2], Joel Jenzer[2], Jasmin Haegele[2], Michal Pikusa[2], Joachim Täger[3], Salvador Rodriguez-Nieto[3], Vangelis Bouris[1], Rafaela Ribeiro[1], Federico Baroni[1], Manmeet Sakshi Bedi[2], Scott Berry[5,6], Marco Losa [1], Simone Hornemann [1], Martin Kampmann[7], Lucas Pelkmans[5], Dominic Hoepfner[2], Peter Heutink[3] & Adriano Aguzzi[1] ✉

[1]Institute of Neuropathology, University of Zurich, Zurich, Switzerland. [2]Novartis Institutes for Biomedical Research, Novartis Campus, Basel, Switzerland. [3]German Center for Neurodegenerative Diseases (DZNE), Tübingen, Germany. [4]Department of Biomedical Sciences, University of Padua, Padova, Italy. [5]Department of Molecular Life Sciences, University of Zurich, Zurich, Switzerland. [6]EMBL Australia Node in Single Molecule Science, School of Medical Sciences, University of New South Wales, Sydney, Australia. [7]Institute for Neurodegenerative Diseases, Department of Biochemistry and Biophysics, University of California San Francisco, San Francisco, CA, USA. [8]These authors contributed equally: Jiang-An Yin, Lukas Frick. ✉e-mail: jiang-an.yin@uzh.ch; adriano.aguzzi@uzh.ch

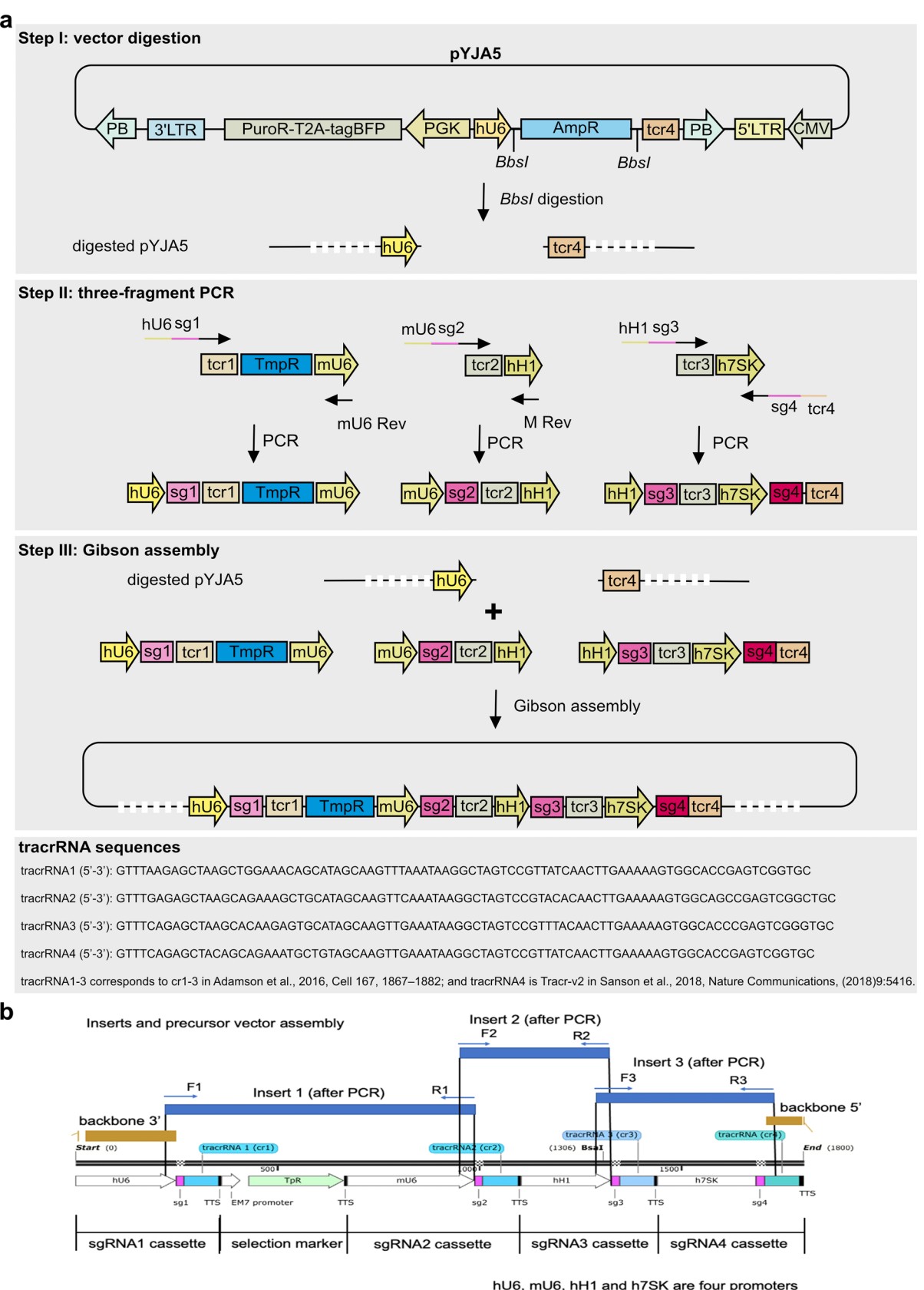

**tracrRNA sequences**

tracrRNA1 (5'-3'): GTTTAAGAGCTAAGCTGGAAACAGCATAGCAAGTTTAAATAAGGCTAGTCCGTTATCAACTTGAAAAAGTGGCACCGAGTCGGTGC

tracrRNA2 (5'-3'): GTTTGAGAGCTAAGCAGAAAGCTGCATAGCAAGTTCAAATAAGGCTAGTCCGTACACAACTTGAAAAAGTGGCAGCCGAGTCGGCTGC

tracrRNA3 (5'-3'): GTTTCAGAGCTAAGCACAAGAGTGCATAGCAAGTTGAAATAAGGCTAGTCCGTTTACAACTTGAAAAAGTGGCACCCGAGTCGGGTGC

tracrRNA4 (5'-3'): GTTTCAGAGCTACAGCAGAAATGCTGTAGCAAGTTGAAATAAGGCTAGTCCGTTATCAACTTGAAAAAGTGGCACCGAGTCGGTGC

tracrRNA1-3 corresponds to cr1-3 in Adamson et al., 2016, Cell 167, 1867–1882; and tracrRNA4 is Tracr-v2 in Sanson et al., 2018, Nature Communications, (2018)9:5416.

**Extended Data Fig. 1 | Details of ALPA cloning method. a**, Step-by-step construction of qgRNA plasmids using ALPA cloning method and sequence information of tracrRNA1-4 (tcr1-4). **b**, Zoom-in illustration of homologous ends overlapping among the three amplicons and the digested vector pYJA5.

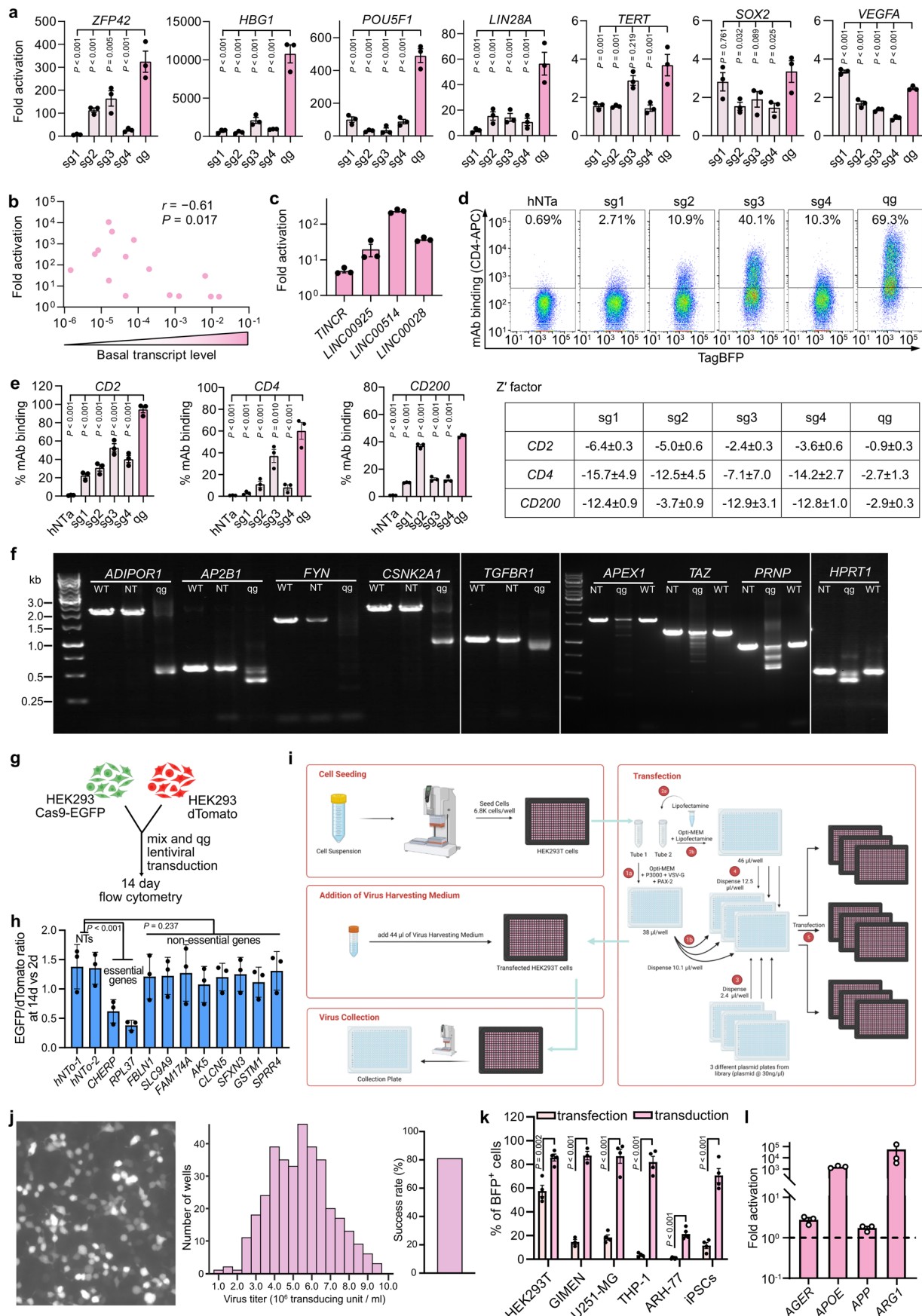

**Extended Data Fig. 2 | See next page for caption.**

**Extended Data Fig. 2 | Efficiency of qgRNAs in gene activation and ablation.**
**a**, Gene activation (qRT-PCR) in HEK293 cells 3 days post-transfection with dCas9-VPR and single (sg1-4) or four sgRNA (qg) plasmids, these are the additional genes tested beyond Fig. 1f. Dots (here and henceforth): independent experiment (Mean ± S.E.M.). **b**, Correlation of fold activation induced by qgRNAs with the basal transcript level among the 15 genes tested. Basal transcript level was normalized to the level of beta ACTIN. **c**, Activation of lncRNAs by qgRNA plasmids co-transfected with dCas9-VPR into HEK293 cells (3 days post-transfection). **d**, Surface expression of CD4 in HEK293 cells expressing dCas9-VPR and single sgRNA (sg1-4) or qgRNA (qg). hNTa, non-targeting control vector. Cell surface proteins were stained with fluorescent-conjugated antibodies and analyzed via flow cytometry. **e**, left, Surface expression of CD2, CD4, and CD200 in HEK293 cells after gene activation with single or qgRNAs; right, Z′ factor of single-sgRNAs or qgRNA vectors for CRISPR activation relative to non-targeting controls. **f**, An example of gel images of PCR amplicons amplified from qgRNA (qg) knockout plasmid-edited genomic DNAs. For each of the genes tested, a pair of primers flanking all the four sgRNA targeting sites was designed (Methods). PCR amplicons of *ADIPOR1*, *AP2B1*, *CSNK2A1*, *FYN*, *HPRT1*, *TGFBR1*, *APEX1*, *TAZ*, and *PRNP* conditions amplified from wildtype (WT) or non-targeting (NT) plasmid treated cells showed the expected sizes of 2095, 558, 2225, 1975, 514, 987, 1663, 1288, and 959 bp on agarose gels, whereas for amplicons from qg-edited cells in each condition, a great majority of them showed shorter sizes, indicating DNA deletions in the corresponding edited genes induced by qgRNA

knockout plasmids. **g**, EGFP and Cas9-expressing HEK293 cells (HEK293-Cas9-EGFP) were mixed with dTomato-expressing HEK293 (HEK293-dTomato) cells (~1:1 ratio) and transduced with qgRNA (qg) lentiviruses. **h**, EGFP/dTomato ratio in a HEK293-Cas9-EGFP and HEK293-dTomato co-culture two weeks after transduction of qgRNA lentiviruses. qgRNA plasmids targeting essential genes, non-essential genes, or non-targeting controls (NTs) were tested. EGFP/dTomato ratio at end point (day 14) was normalized to the ratio at day 2 post-transduction. **i**, A schematic of lentivirus production in 384-well plates (see methods in Supplementary Information). Created with BioRender.com. **j**, left, TagBFP⁺ cells of a typical well of 384-well plate post lentiviral transfection; middle, Titers of lentiviral particles packaged from an example of 384-well plate of library plasmids; Viral particles were produced in 384-well plates with HEK293T cells and transduced to a different population of HEK293T cells; TagBFP+ cells were quantified by flow cytometry 3 days post-transduction; right, The success rate for the lentiviral production of the T.gonfio library. **k**, Gene delivery efficiency of the qgRNA vector into poorly transfectable cells, measured by flow cytometry of TagBFP⁺ cells 3 days post-transduction. **l**, Gene activation in neurons derived from human induced pluripotent stem cells, measured by qRT-PCR after transduction of qgRNA lentiviruses (multiplicity of infection: 1.4). Target neurons stably expressed dCas9-VPR. Assays were performed at day 7 post-infection. In **a** and **e**, p values were determined by one-way ANOVA with Dunnett's multiple comparisons test; in **b**, p value was determined by Pearson correlation analysis; and in **h** and **k**, p values were determined by two-tailed Student's t-test.

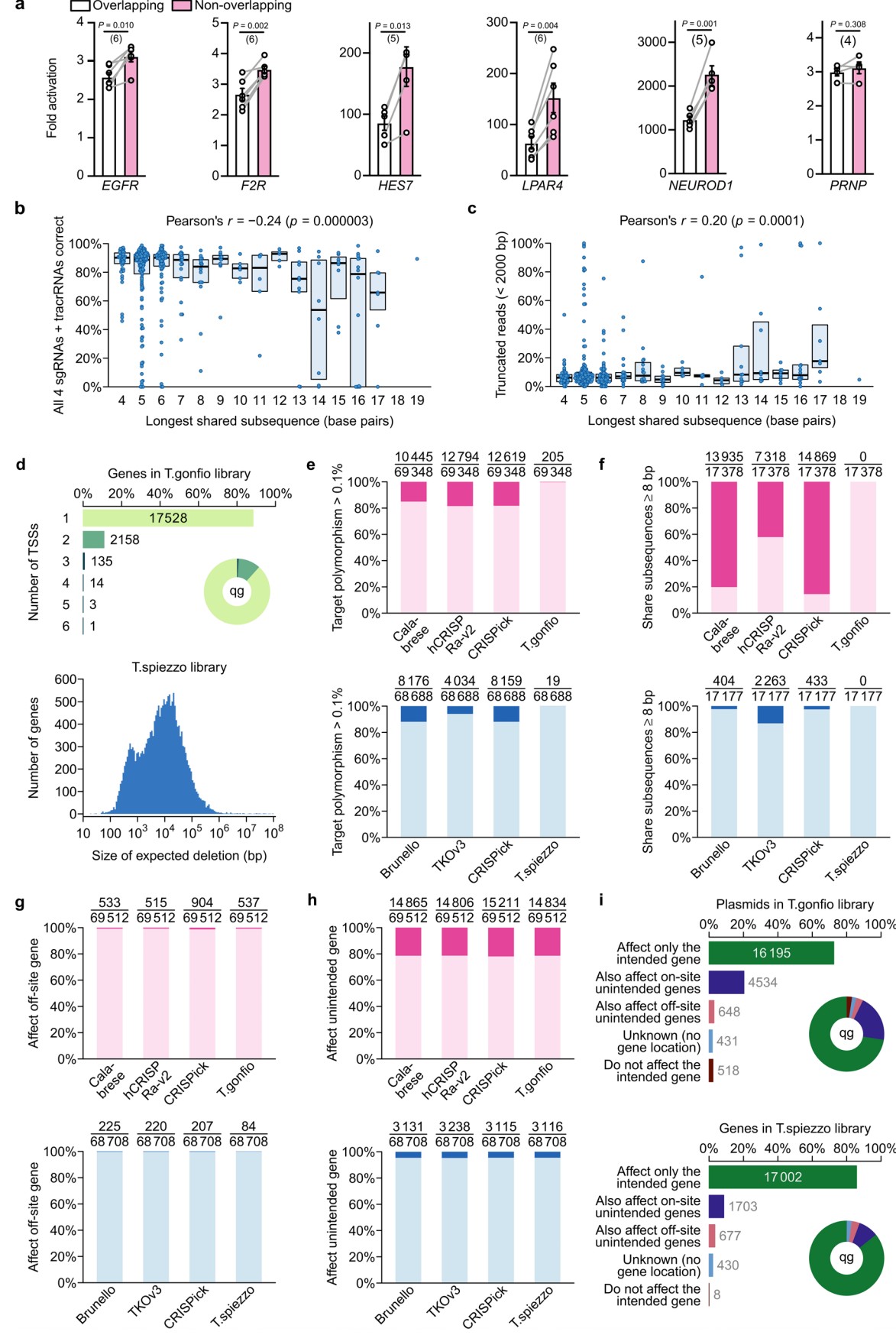

**Extended Data Fig. 3 | See next page for caption.**

**Extended Data Fig. 3 | The effect of sgRNA spacing and homology on qgRNA plasmids and other features of T.spiezzo and T.gonfio libraries.**
**a**, Comparison of the effect of overlapping and non-overlapping sgRNAs on gene activation in HEK293 cells (Two-tailed Student's t-test). **b**, Correlation between the extent of homology among the four sgRNAs and the percentage of correct plasmids. **c**, Correlation between the extent of homology and the frequency of shortened amplicon regions (indicating deletions). In **b** and **c**, the box plot represents the median and interquartile range. **d**, Summary of the number of transcription start sites (TSSs) per gene that are each targeted by a separate plasmid in the T.gonfio library (top), and the estimated size of deletions between the first and last cut sites of each qgRNA plasmid in the

T.spiezzo library (bottom). **e**, Percentage of sgRNAs that target genomic sites affected by a polymorphism with a frequency higher than 0.1% in the T.spiezzo and T.gonfio libraries in comparison with the top 4 sgRNAs from existing resources. **f**, Percentage of sgRNAs that share 8 or more base pairs of homology in the T.spiezzo and T.gonfio libraries in comparison with the top 4 sgRNAs from existing resources. **g** and **h**, Comparison of the percentage of sgRNAs predicted to target unintended genes at off-site locations (**g**) and all locations (**h**) – the latter include mostly sgRNAs with *on-site* unintended targets. **i**, All plasmids in the T.spiezzo and T.gonfio libraries were assigned to mutually exclusive categories, based on whether any of the 4 sgRNAs may target additional, unintended genes.

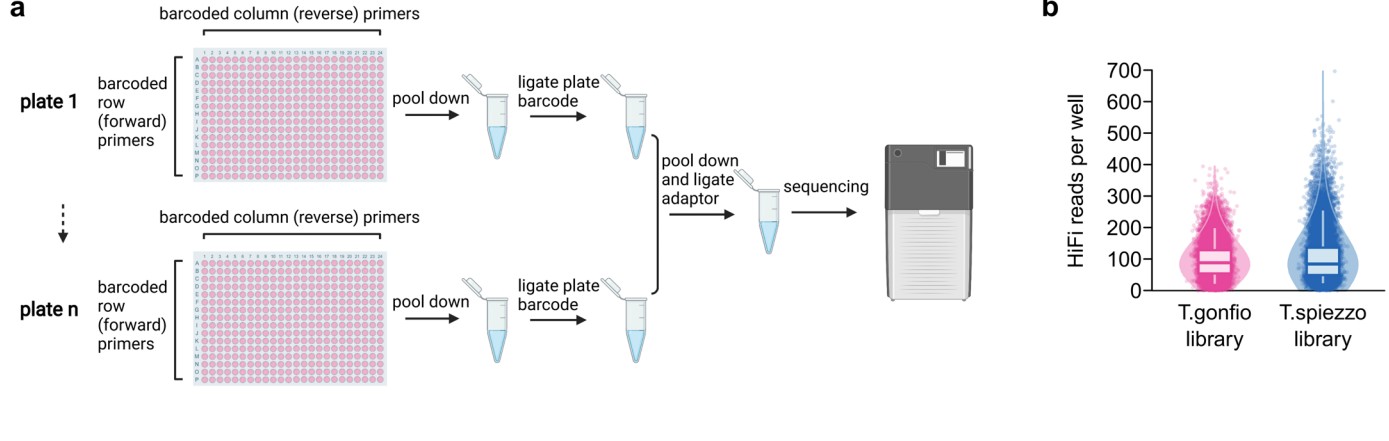

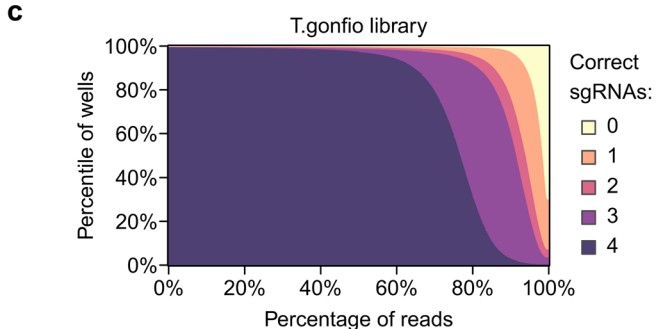

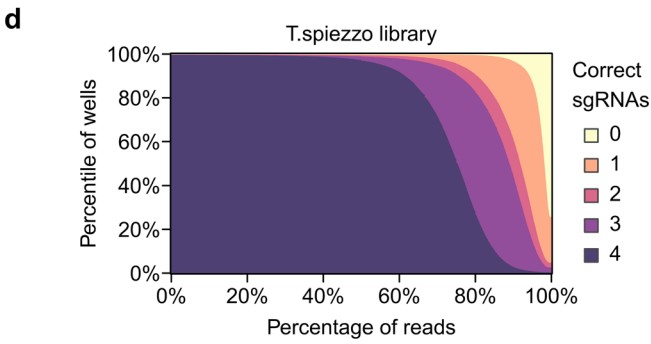

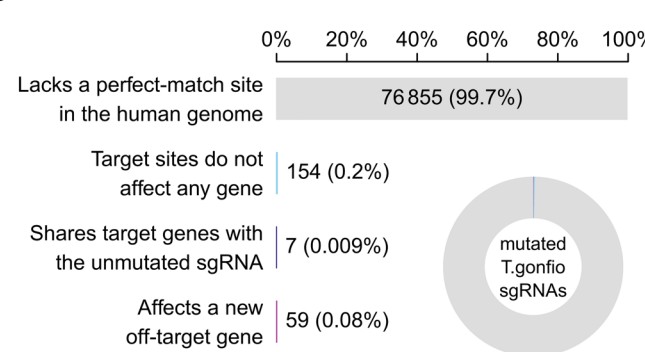

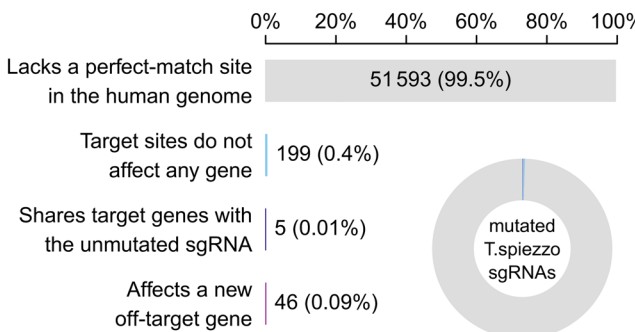

**Extended Data Fig. 4 | Genome-wide sequencing of the T.spiezzo and T.gonfio libraries. a**, SMRT long-read sequencing workflow (Created with BioRender. com): PCR was performed in each well of a 384-well plate using primers appended with row- and column-specific barcodes. All wells from one plate were pooled and ligated with plate-specific barcodes, and multiple plates were further pooled for sequencing. **b**, High-quality read count for each well in the T.spiezzo and T.gonfio libraries. The box represents the median and interquartile range; the whiskers indicate the 5th and 95th percentiles. **c** and **d**, Cumulative distribution of each well of plasmids with 0, 1, 2, 3, 4 entirely correct sgRNA and tracrRNA sequences, as well as an associated promoter sequence that was at least 95% correct, in the T.spiezzo and T.gonfio libraries. **e** and **f**, Predicted off-target effects for mutated sgRNAs in the T.spiezzo and T.gonfio libraries. Guide RNAs were considered to target a gene if they lay within coding sequences or exons (for CRISPR knockout plasmids) or within 1000 base pairs of a transcription start site (for CRISPR activation plasmids).

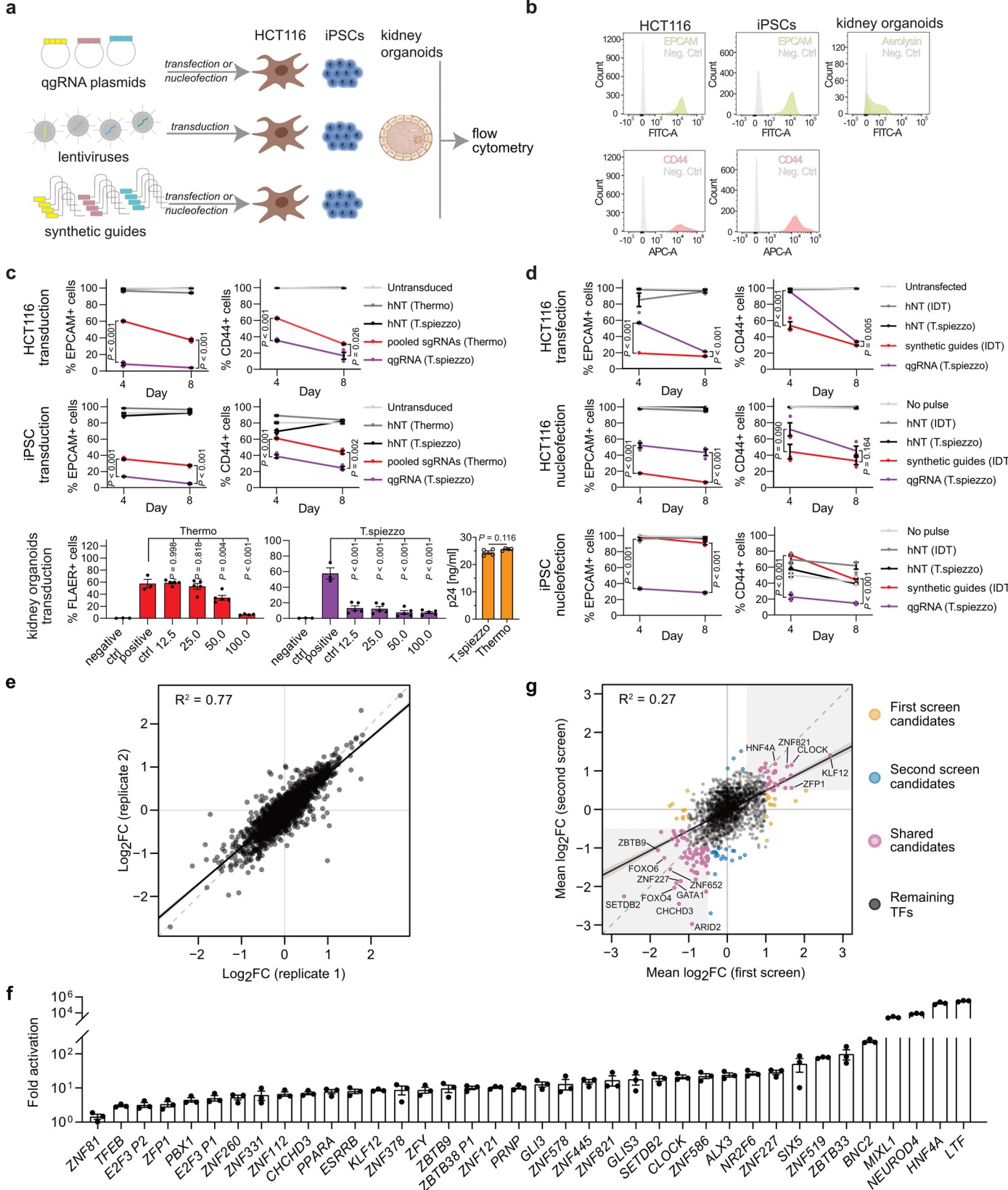

**Extended Data Fig. 5 | See next page for caption.**

**Extended Data Fig. 5 | Benchmarking of qgRNA ablation plasmids in cells and organoids. a**, Schematic of the experiment (Created with BioRender.com). qgRNA plasmids, synthetic guide RNAs, or lentivirally packaged sgRNAs were either transfected, nucleofected or transduced into Cas9 expressing HCT116, iPSCs and nephron progenitor cells which were matured to kidney organoids. **b**, Flow cytometry histograms of Cas9-expressing HCT116 and iPSC cells stained with antibodies to EPCAM (*green; top left and middle*) or CD44 (*red; bottom left and middle*), and single-cell-dissociated kidney organoids stained with fluorescently labelled aerolysin (*green, top right*). **c**, Top and middle, percentages of EPCAM+ or CD44+ HCT116-Cas9 (Top) and iPSC-iCas9 cells (middle) transduced with lentiviruses carrying the T.spiezzo qgRNA vector or a mixture of four individual lentiviruses (*Thermo*) targeting *EPCAM* or *CD44* or non-targeting (hNT) controls at 4 and 8 days post-transduction (*n*=3; error bars represent S.E.M.); bottom (left and middle), Percentage of fluorescently labelled aerolysin (FLAER) positive cells dissociated from kidney organoids transduced with T.spiezzo lentiviruses or four individual lentiviruses (*Thermo*) targeting *PIGA* at increasing viral volumes compared to the unstained (negative ctrl) and untransduced (positive Ctrl)

controls (*n*=4-5; error bars represent S.E.M.); bottom (right), p24 ELISA of supernatants containing *T.spiezzo* or *Thermo* lentiviruses targeting *PIGA* (*n*=4; error bars represent SEM). **d**, Percentages of EPCAM+ or CD44+ HCT116-Cas9 cells transfected (Top panels), electroporated (middle panels) or iPSC-iCas9 cells nucleofected (bottom panels) with *T.spiezzo* (5 µg) or 4 synthetic guide RNAs (*Integrated DNA Technologies*, 10 µM) targeting *EPCAM* or *CD44* at 4 and 8 days post-transfection (*n*=3; error bars: S.E.M.) **e**, Duplicate correlation across samples from the primary TF screen. **f**, Gene activation (qRT-PCR) of the 36 hits identified in Fig. 4e. **g**, Correlation analysis of the first and second TF screens, which were carried out with an interval of over one year. Candidates were determined based on a consistent log2 fold change (|Log2FC|≥1, *p*≤0.05). Candidates identified in one of the screens exhibiting a consistent trend in PrP$^C$ regulation with |log2FC|≥0.5 in the other screen were defined as shared candidates. In **c** and **d**, except the FLAER assay in the kidney organoids whose *p* values were determined by one-way ANOVA with Dunnett's multiple comparisons test, all other *p* values were determined by two-tailed Student's t-test.

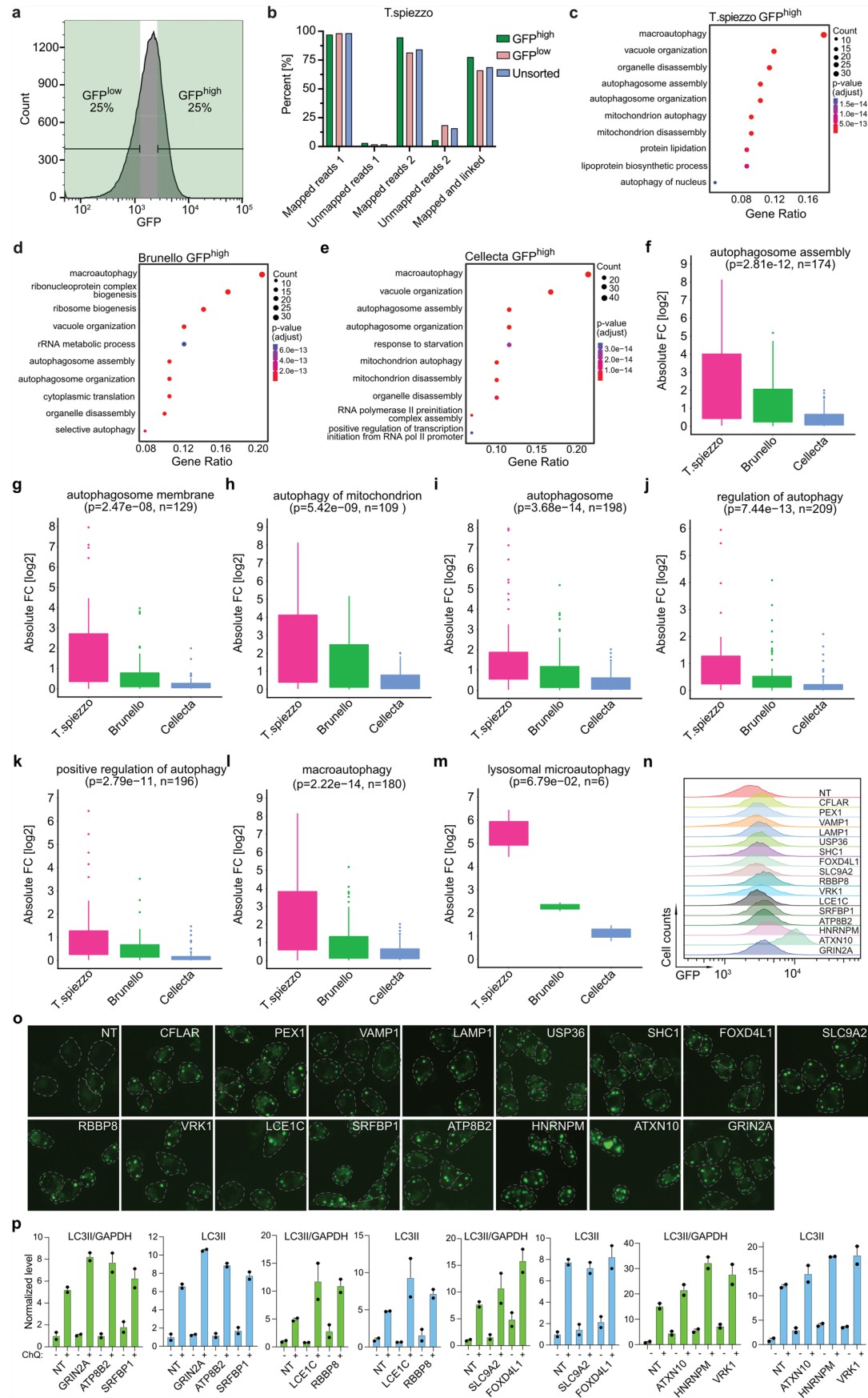

**Extended Data Fig. 6 | See next page for caption.**

**Extended Data Fig. 6 | Autophagy screens with the pooled T.spiezzo library and existing CRISPR knockout libraries. a**, Histogram showing the gating strategy to isolate GFP^high and GFP^low (upper and lower quartile of GFP fluorescence, respectively) cell populations. **b**, Percentages of sequencing reads in GFP^high, GFP^low, and unsorted samples from the T.spiezzo pooled screen that correctly aligned to sgRNA2 (mapped reads 1) or sgRNA3 (mapped reads 2), that did not align to sgRNA2 (unmapped reads 1) or sgRNA3 (unmapped reads 2) and those that aligned and had the correct linkage between sgRNA2 and sgRNA3 (mapped and linked). **c-e**, Overrepresentation analysis of the top 200 genes enriched in GFP^high cell populations from the T.spiezzo (**c**), Brunello (**d**), and Cellecta (**e**) screens. Gene counts and adjusted p-value are represented in each figure. The 10 most significant GO biological processes are shown. **f-m**, Autophagy-related gene sets including autophagosome assembly (GO:0000045, n=174, **f**), autophagosome membrane (GO:0000421, n=129, **g**), autophagy of mitochondrion (GO:0000422, n=109, **h**), autophagosome (GO:0005776, n=198, **i**), regulation of autophagy (GO:0010506, n=209, **j**), positive regulation of autophagy (GO:0010508, n=196, **k**), macroautophagy (GO:0016236, n=180, **l**),

and lysosomal microautophagy (GO:0016237, n=6, **m**) using absolute log$_2$ fold changes in GFP^high cell populations from the T.spiezzo, Brunello, and Cellecta screens. The p value was determined by two-way ANOVA. The box plot represents the interquartile range. **n**, An example of flow cytometry histograms of GFP-SQSTM1 intensity in H4-Cas9-GFP-SQSTM1 cells transduced with T.spiezzo qgRNA lentivirus against each of the 16 genes selected for validation or a non-targeting control (NT) lentivirus. N = 3 biological repeats. **o**, An example of GFP-SQSTM1 puncta in H4-Cas9-GFP-SQSTM1 cells transduced with T.spiezzo qgRNA lentivirus against each of the 16 genes selected for validation or NT controls. N = 3 biological repeats. Cells were demarcated by dashed lines according to the cytosolic GFP signal. **p**, Quantification of LC3II levels of cells and conditions described in Fig. 5i. All values were normalized to the mean of the two NT repeats (- ChQ) on the same blot. Both the LC3II and normalized LC3II (LC3II/GAPDH) levels were shown to determine whether consistent changes were observed for the two biological repeats of a defined gene to determine promising candidates for further validation.

**a**

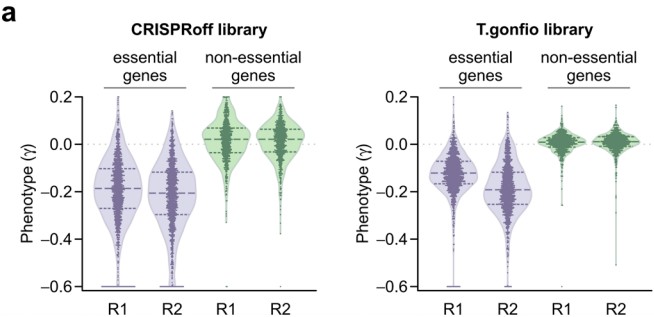

**b**

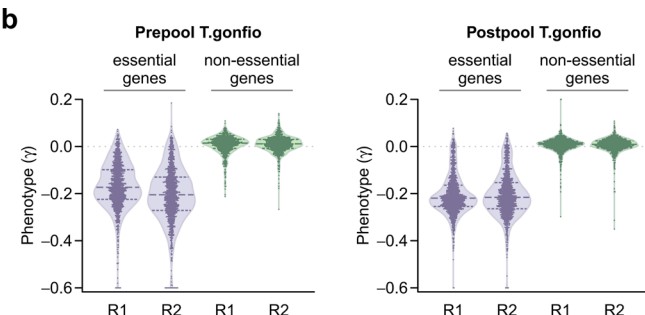

**Extended Data Fig. 7 | Genome-wide CRISPRoff screens. a** and **b**, Phenotype scores (γ) for essential and non-essential genes from screens with the CRISPRoff or the T.gonfio library (**a**) and the pre-pool or post-pool of the T.gonfio library (**b**). R1 and R2 represent each individual biological repeat of the screens. Values above 0.2 or below −0.6 were set to 0.2 and −0.6, respectively. The dashed lines indicate the first quartile, median and third quartile.

# Reporting Summary

## Statistics

For all statistical analyses, confirm that the following items are present in the figure legend, table legend, main text, or Methods section.

| n/a | Confirmed | |
|---|---|---|
| ☐ | ☒ | The exact sample size (*n*) for each experimental group/condition, given as a discrete number and unit of measurement |
| ☐ | ☒ | A statement on whether measurements were taken from distinct samples or whether the same sample was measured repeatedly |
| ☐ | ☒ | The statistical test(s) used AND whether they are one- or two-sided<br>*Only common tests should be described solely by name; describe more complex techniques in the Methods section.* |
| ☒ | ☐ | A description of all covariates tested |
| ☐ | ☒ | A description of any assumptions or corrections, such as tests of normality and adjustment for multiple comparisons |
| ☐ | ☒ | A full description of the statistical parameters including central tendency (e.g. means) or other basic estimates (e.g. regression coefficient) AND variation (e.g. standard deviation) or associated estimates of uncertainty (e.g. confidence intervals) |
| ☐ | ☒ | For null hypothesis testing, the test statistic (e.g. *F*, *t*, *r*) with confidence intervals, effect sizes, degrees of freedom and *P* value noted<br>*Give P values as exact values whenever suitable.* |
| ☒ | ☐ | For Bayesian analysis, information on the choice of priors and Markov chain Monte Carlo settings |
| ☒ | ☐ | For hierarchical and complex designs, identification of the appropriate level for tests and full reporting of outcomes |
| ☐ | ☒ | Estimates of effect sizes (e.g. Cohen's *d*, Pearson's *r*), indicating how they were calculated |

*Our web collection on statistics for biologists contains articles on many of the points above.*

## Software and code

Policy information about availability of computer code

| Data collection | For cell analysis and sorting with flow cytometers, data were obtained using the BD FACSDiva software. For the construction of our arrayed CRISPR libraries, we wrote a custom code pipeline that is available at https://github.com/Lukas-1/CRISPR_4sgRNA. All code is based on the R statistical programming environment, version 3.6.3, and Bioconductor suite, version 3.10.0. For TR-FRET measurement to determine PrPC level in cell lysates, EnVision Manager software (version 1.14.3049.528) was used. GFP-SQSTM1 images were taken using Zeiss fluorescent microscopy with ZEN 2012 software (version 1.1.2.0); YFP-LC3 images were taken with Leica confocal microscopy with LAS X software (version 4.5.0.25531). |
|---|---|
| Data analysis | For flow cytometry, data were analyzed with FlowJo (version 10.9.0). For the quality-control analysis with single-molecule sequencing of plasmids and for the arrayed and pooled CRISPRoff screens, the analysis was performed with custom code available at https://github.com/Lukas-1/CRISPR_4sgRNA/tree/master/6)%20Individual%20experiments using the R statistical programming environment, version 3.6.3. Western blot images, GFP-SQSTM1 and YFP-LC3 images were analyzed with Fiji (ImageJ, version 2.14.0). For plotting data and statistical analyses, GraphPad Prism (version 9.4.0) or Microsoft Excel was used. |

For manuscripts utilizing custom algorithms or software that are central to the research but not yet described in published literature, software must be made available to editors and reviewers. We strongly encourage code deposition in a community repository (e.g. GitHub). See the Nature Portfolio guidelines for submitting code & software for further information.

## Data

Policy information about <u>availability of data</u>

All manuscripts must include a <u>data availability statement</u>. This statement should provide the following information, where applicable:
- Accession codes, unique identifiers, or web links for publicly available datasets
- A description of any restrictions on data availability
- For clinical datasets or third party data, please ensure that the statement adheres to our <u>policy</u>

The complete sgRNA sequences, metadata and annotation for our T.spiezzo and T.gonfio libraries are included as Source Data File 1. The count and fold-change data from the CRISPR screens are available as Source Data Files 2–4. All additional experimental and sequencing data are available from the corresponding authors on reasonable request. Source data are provided with this paper.

## Research involving human participants, their data, or biological material

Policy information about studies with <u>human participants or human data</u>. See also policy information about <u>sex, gender (identity/presentation), and sexual orientation</u> and <u>race, ethnicity and racism</u>.

| | |
|---|---|
| Reporting on sex and gender | The study did not involve human research participants. |
| Reporting on race, ethnicity, or other socially relevant groupings | - |
| Population characteristics | - |
| Recruitment | - |
| Ethics oversight | - |

Note that full information on the approval of the study protocol must also be provided in the manuscript.

# Field-specific reporting

Please select the one below that is the best fit for your research. If you are not sure, read the appropriate sections before making your selection.

☒ Life sciences ☐ Behavioural & social sciences ☐ Ecological, evolutionary & environmental sciences

For a reference copy of the document with all sections, see nature.com/documents/nr-reporting-summary-flat.pdf

# Life sciences study design

All studies must disclose on these points even when the disclosure is negative.

| | |
|---|---|
| Sample size | No statistical method was used to predetermine sample sizes. The sample sizes in our experiments were determined according to established literature. |
| Data exclusions | For the calculation of the mean Z' factors in Extended Data Fig. 2e, one data point of sg1 of CD4 and one data point of sg1 of CD200 were excluded because the values were outliers of the assessment. The criteria was not pre-established. For all other figures or data, no data were excluded. |
| Replication | All experiments were performed multiple times, usually with 3–5 biological repeats, to reach a conclusive result. Genomic screens were performed 1–2 times and the top candidate genes were validated by various methods. Details on the number of experimental repeats and statistical analysis can be found in the figure legends. |
| Randomization | All samples were randomized to ensure no bias was introduced by investigators. |
| Blinding | The investigators were not blinded to the experimental group allocation. Most of the experiments were measured with quantitative readouts such as real-time quantitative PCR, flow cytometry and sequencing. Data analysis was performed while remaining blind to the sample assignments. |

# Reporting for specific materials, systems and methods

We require information from authors about some types of materials, experimental systems and methods used in many studies. Here, indicate whether each material, system or method listed is relevant to your study. If you are not sure if a list item applies to your research, read the appropriate section before selecting a response.

## Materials & experimental systems

| n/a | Involved in the study |
|---|---|
| ☐ | ☒ Antibodies |
| ☐ | ☒ Eukaryotic cell lines |
| ☒ | ☐ Palaeontology and archaeology |
| ☒ | ☐ Animals and other organisms |
| ☒ | ☐ Clinical data |
| ☒ | ☐ Dual use research of concern |
| ☒ | ☐ Plants |

## Methods

| n/a | Involved in the study |
|---|---|
| ☒ | ☐ ChIP-seq |
| ☐ | ☒ Flow cytometry |
| ☒ | ☐ MRI-based neuroimaging |

## Antibodies

| | |
|---|---|
| Antibodies used | Flow cytometry: APC anti-human CD47 Antibody (Biolegend, Cat. # 323124, Clone CC2C6); PE anti-human CD119 (IFN-γ R α chain) Antibody (Biolegend, Cat. # 308606, Clone GIR-208); APC anti-human CD146 Antibody (Biolegend, Cat. # 361016, Clone P1H12); PE anti-human CD29 Antibody (Biolegend, Cat. # 303004, Clone TS2/16); FITC anti-human CD81 (TAPA-1) Antibody (Biolegend, Cat. # 349504, Clone 5A6); APC anti-human CD151 (PETA-3) Antibody (Biolegend, Cat. # 350406, Clone 50-6), APC anti-human CD2 Antibody (Biolegend, Cat. # 300214, Clone RPA-2.10); APC anti-human CD4 Antibody (Biolegend, Cat. # 357408, Clone A161A1); APC anti-human CD200 (OX2) Antibody (Biolegend, Cat. # 329208, Clone OX-104); FITC Anti-EpCAM antibody [VU-1D9] (Abcam Cat. # ab112067); Alexa Fluor® 647 anti-mouse/human CD44 Antibody (Biolegend, Cat. # 103018, Clone IM7). <br><br> PrPC screen and validation: The anti-human PrPC antibodies POM1 and POM2 were produced in-house. The EU-POM2 antibody was produced by conjugating the POM2 antibody with europium (Eu, Perkin Elmer, Cat.# AD0013) in-house; the APC-POM1 antibody was produced by conjugating the POM1 antibody with allophycocyanin (APC, Abcam, Cat.# ab201807) in-house; Recombinant Anti-Vinculin antibody [EPR8185] (Abcam, Cat.# ab129002). <br><br> Autophagy hits validation: LC3B (D11) XP® Rabbit mAb #3868 (Cell Signaling, Cat. # 3868); Anti-GAPDH antibody produced in rabbit (Sigma-Aldrich, Cat. # G9545). |
| Validation | All commercial antibodies were validated by their suppliers: Biolegend, Abcam, Cell Siganling, and Sigma-Aldrich. Anti-PrPC antibodies including EU-POM2, APC-POM1, and POM2 were well established and validated in the Aguzzi laboratory and used in previous publications (Heinzer et al., PLoS Pathog 17, e1010013 (2021); (Pease et al., Brain Pathol 29, 232-244 (2019)). |

## Eukaryotic cell lines

Policy information about cell lines and Sex and Gender in Research

| | |
|---|---|
| Cell line source(s) | HEK293, H4, HEK293T, GIMEN, THP-1, ARH-77, and HCT116 cells were purchased from ATCC. U251-MG was purchased from Kerafast, Inc., Boston, MA, USA, AccessionID: CVCL_0021. iPSC cell line (Gm23280) was obtained from the Coriell Institute for Medical Research (https://www.coriell.org) and iNeurons were differentiated from the iPSCs. Kidney organoids were differentiated from NPCs which were established at the Novartis Institutes for Biomedical Research. |
| Authentication | Each cell line was handled and cultured separately, stored at early passages, and discarded after more than 20 passages. All these measures helped to preserve cell identity. |
| Mycoplasma contamination | iPSCs tested negative for mycoplasma once per month. All other cells were not tested for mycoplasma contamination. |
| Commonly misidentified lines (See ICLAC register) | No commonly misidentified cell lines were used. |

## Plants

| | |
|---|---|
| Seed stocks | The study did not involve plants. |
| Novel plant genotypes | - |
| Authentication | - |

# Flow Cytometry

## Plots

Confirm that:

☒ The axis labels state the marker and fluorochrome used (e.g. CD4-FITC).

☒ The axis scales are clearly visible. Include numbers along axes only for bottom left plot of group (a 'group' is an analysis of identical markers).

☒ All plots are contour plots with outliers or pseudocolor plots.

☒ A numerical value for number of cells or percentage (with statistics) is provided.

## Methodology

| | |
|---|---|
| Sample preparation | Suspension cells were directly collected via centrifugation. All other cells were dissociated with trypsin and then collected via centrifugation. Cells were washed once with PBS and analyzed in 500 µl of PBS. For antibody staining conditions, live cells were stained with the respective antibodies according to the manual and then analyzed. Non-treated cells were used as controls. |
| Instrument | Cells were analyzed using BD Canto II or LSR II Fortessa flow cytometers. Cells were sorted by BD FACSymphony S6 or FACSAria III. |
| Software | FACS data were collected using BD FACSDiva (BD Biosciences), and data analysis was performed using FlowJo (version 10.9.0). Statistical analysis and data visualization were conducted using GraphPad Prism (version 9.4.0). |
| Cell population abundance | Cell sorting was applied in the autophagy screen to isolate GFP-high and GFP-low cell populations. Cell sorting was also applied in CRISPRoff screens to enrich cells that were double positives for mScarletI and GFP (using the dual-guide CRISPRoff library) or mScarletI and TagBFP (using the T.gonfio library), respectively. For confirmation, a subsample of the sorted cells was reanalyzed using the same parameters with the corresponding sorters. |
| Gating strategy | Please refer to the Supplementary Fig. 1. |

☒ Tick this box to confirm that a figure exemplifying the gating strategy is provided in the Supplementary Information.

