## [Peer Review File · Nature Biomedical Engineering]

Arrayed CRISPR libraries for the genome-wide activation, deletion and silencing of human protein-coding genes

Corresponding author: Adriano Aguzzi

Editorial note

This document includes relevant written communications between the manuscript's corresponding author and the editor and reviewers of the manuscript during peer review. It includes decision letters relaying any editorial points and peer-review reports, and the authors' replies to these (under 'Rebuttal' headings). The editorial decisions are signed by the manuscript's handling editor, yet the editorial team and ultimately the journal's Chief Editor share responsibility for all decisions.

Any relevant documents attached to the decision letters are referred to as **Appendix #**, and can be found appended to this document. Any information deemed confidential has been redacted or removed. Earlier versions of the manuscript are not published, yet the originally submitted version may be available as a preprint. Because of editorial edits and changes during peer review, the published title of the paper and the title mentioned in below correspondence may differ.

Correspondence

Thu 04 Jan 2024

Decision on Article nBME-23-2583

Dear Prof Aguzzi,

Thank you again for submitting to *Nature Biomedical Engineering* your manuscript, "Quadruple-Guide Arrayed CRISPR for Indexed Human Genome-Wide Activation, Deletion, and Silencing", and for your patience in waiting for the reviewer feedback. The manuscript has been seen by three experts, whose reports you will find at the end of this message. You will see that the reviewers appreciate the work, and that they raise a number of technical criticisms that I am hoping you will be able to address. In particular, we would expect that a revised version of the manuscript provides:

- * Performance comparisons and analyses with additional target genes, as per the arguments of all reviewers.
- * Clear recognition of the lentivirus template-switching problem, as per the sensible suggestions of Reviewer #3.
- * Thorough methodological reporting, as per the pertinent points from all reviewers.

Please do also consider the following editorial points:

- * Ensure that all generated DNA constructs are deposited on Addgene.
- * To facilitate re-use (which is particularly relevant for methods such as yours), we strongly recommend that all the data be deposited in a relevant repository. The Data availability statement in the manuscript should be updated accordingly.* We discourage the use of acronyms that are also common words, especially when the acronyms appear to be 'forced' to match the words they embed. I suggest that, rather than 'APPEAL', you use 'ALP-PAC', which consistently matches the initials of your method's name.

* As noted by Reviewer #3, the first sentence of the introduction is unclear, and does not seem to be scientifically informative. I suggest that you use instead 'Genetic screens are fundamentally useful methods for biological and biomedical research1–3' or similar wording.

* Please remove the gridlines in all graphs with them; they are against house style.

* Rather than providing separate spreadsheet files as Supplementary Information, for usability considerations please provide a single file (or fewer files) with the data well structured and organized within tabs.

When you are ready to resubmit your manuscript, please upload the revised files, a point-by-point rebuttal to the comments from all reviewers, the reporting summary, and a cover letter that explains the main improvements included in the revision and responds to any points highlighted in this decision.

Please follow the following recommendations:

* Clearly highlight any amendments to the text and figures to help the reviewers and editors find and understand the changes (yet keep in mind that excessive marking can hinder readability).

* If you and your co-authors disagree with a criticism, provide the arguments to the reviewer (optionally, indicate the relevant points in the cover letter).

* If a criticism or suggestion is not addressed, please indicate so in the rebuttal to the reviewer comments and explain the reason(s).

* Consider including responses to any criticisms raised by more than one reviewer at the beginning of the rebuttal, in a section addressed to all reviewers.

* The rebuttal should include the reviewer comments in point-by-point format (please note that we provide all reviewers will the reports as they appear at the end of this message).

* Provide the rebuttal to the reviewer comments and the cover letter as separate files.

We hope that you will be able to resubmit the manuscript within 10 weeks from the receipt of this message. If this is the case, you will be protected against potential scooping. Otherwise, we will be happy to consider a revised manuscript as long as the significance of the work is not compromised by work published elsewhere or accepted for publication at *Nature Biomedical Engineering*.

We hope that you will find the referee reports helpful when revising the work, which we look forward to receive. Please do not hesitate to contact me should you have any questions.

Best wishes,

Pep

Pep Pàmies
Chief Editor, Nature Biomedical Engineering

Reviewer #1 (Report for the authors (Required)):

In this paper, Yin et al. constructed two genome-wide libraries, T. spiezzi and T. gonfio, targeting each

human gene with four non-overlapping gRNAs. The authors provide extensive evidence to support the quality of the libraries and their performance in Crispr-mediated knockout and overexpression experiments. By targeting each gene with multiple gRNAs, T. spiezzo and T. gonfio improve the potency of gene perturbation (particularly in CRISPRa experiments), enhance the sensitivity of pooled CRISPR-screens, and reduce the library size. In addition, the authors confirm that T. gonfio can be efficiently used in gene silencing experiments using the Crispr-off machinery.

The new libraries developed in this paper show improved performance when compared to previously described libraries and are therefore a resource that can be used by the scientific community to perform high quality CRISPR screenings at lower costs. However, the methods used by the authors to design and generate the library are not revolutionary.

The paper is overall well presented, although in many instances it would be useful to have more details on the experimental design to easily understand the figures.

Detailed comments:

- Resources generated in this paper are potentially useful for the community. However, it is not clear how these resources would be easily distributed or accessible and the conceptual advance or methodological novelty is limited.
- In figure 1, the performance of T. gonfio (f-g) and T. spiezzo (h-i) is compared to single gRNAs targeting very few genes (Figure 1 f-h). They also show data for other genes that have been proven to be difficult to activate (g) or silence (i). The number of these genes is still pretty limited and in g-i a direct comparison with single gRNAs is lacking. Are these data sufficient to convincingly prove better performances of T. gonfio and T. spiezzo? This is important in light of data in Extended figure 5 c-d that show very variable silencing results obtained with T.spiezzo targeting EPCAM and CD44 in different cell types and with different delivery methods.
- It seems to me that for KO experiments, T. spiezzo performs similarly to the best single gRNAs, while T. gonfio allows very high gene activation. Is that a general rule or is it just a case for the 3 targets tested? Already in the extended figure 2a-b, the difference between T. gonfio and the best gRNA is not as dramatic as figure 1f. Are more genes needed to conclude that T. gonfio reliably outperforms single gRNAs?
- What is the degree of activation for the genes in the transcription factor screen (at least for the hits listed in figure 4e)?

Minor technical criticisms:

- Figure 1b: for the single colony PCR it would be useful to have a schematic of the position of the primers (similarly to what shown in the bottom of extended figure 1) and the expected band size.
- For figure 1h, I found that this way of showing the data is very misleading, as it is not obvious that each dot represents the average of knockout for a single gRNA. I would rather the authors presented the data as in figure 1f or extended figure 2d.
- In the extended figure 2b, it seems unlikely that some of the single gRNA results are not significant.
- Extended figure 5c-d, statistics are missing. In addition, the panel in figure c bottom on P24 is not discussed anywhere in the paper.
- Lines 360-364: this paragraph is confusing if considering the statistics in figure 4h
- Figure 5g: it would be helpful to add a signal to visualize the cells (i.e. Dapi)
- Figure 5i: please add quantifications of the western blots as the conclusions described in the main text are not as obvious.

Reviewer #2 (Report for the authors (Required)):

Summary:

The authors construct genome-wide arrayed plasmid libraries for human gene ablation (T. Spiezzo), as well as activation/silencing (T. gonfio). Each plasmid encodes an array of 4 non-overlapping gRNAs (quadruple gRNAs or qgRNAs) whose expression is driven by different RNA polIII promoters. The APPEAL (Automated-liquied-Phase-Plasmid-assEmbly-And-cloning) cloning method was developed to generate each construct in the arrayed libraries. Subsequent assessment was that both libraries were of high quality by various metrics; for example, 90% of wells were >75% correct for each of the 4 individual sgRNAs. The qgRNA plasmids from the libraries were benchmarked for cell surface markers in cell lines, iPSCs and kidney organoids. A number of different screens were next performed for functional discovery including: CRISPRa arrayed screen for TFs controlling expression of the PRNP gene product, CRISPRko pooled screen for regulation of the autophagy reporter SQSTM1, and a pooled CRISPR-off screen for epigenetic silencing. Validation efforts were also made, particularly on the results of the autophagy CRISPRko screen where 5 novel autophagy modifiers were validated using orthogonal methods.

Comments:

This study presents construction methods and SpCas9-based quadruple guide-RNA (qgRNA) libraries for lentiviral-based arrayed phenotypic screens. Overall, and despite the large number of existing CRISPRko/i/a libraries, this may prove to be a useful resource. The arrayed nature of the library will provide researchers with the option to miniaturize their assays in multi-well format with individual knockdowns. Furthermore, the quadruple nature of the system reduces the number of wells required for genome-wide screens by 4-fold. However, aside from the PrPc arrayed screen, the study is all predicated on simple phenotypes and pooled screens. In fact, the PrPc screen could arguably have been carried out as a pooled screen. The advantage of the arrayed format is that one can image the cells and examine single cell phenotypes in a multiplexed quantitative fashion. This missed opportunity would certainly enhance the value of this resource and provide some context to its usability in more complex assays.

Specific comments:

- * Is there any meaning behind naming of the libraries T. Spiezzo and T. gonfio? Seems arbitrary.
- * A number of features are built into pYJA5 (e.g. PB, PuroR-T2A-TagBFP, etc.) that are not used to their capacity. For example, PB transposition could introduce many cassettes into the same cell, followed by expression of the qgRNAs which could lead to substantial off-target effects as the concentration of gRNAs increases. It would be appropriate to comment whether these features were tested and function as expected.
- * Figure 1f. Given that the general perception in the field is that CRISPRa is highly gRNA-dependent, it is surprising that the authors observe no effects with any of the individual gRNAs yet a robust activation with the qgRNA. Could the authors comment on whether they have observed single gRNA CRISPRa effects in their system? Have they tested all pairs to measure synergy? I suppose the results of the shared subsequence experiment (EDF 3b,c) provides some answer to this but doesn't explain why no single gRNA works to activate the target genes they are testing.
- * p3.line83. Define monospecific.
- * The methods for producing arrayed virus in 384-well plates are well described but it is difficult to fathom that this process is 100% efficient at generating lentivirus with titers $\sim 1e6$ IUs/mL. The methods (Methods, page 6) indicate that the percentage of TagBFP-positive infected cells were analysed by flow cytometry to estimate titers, but there was no data to review to assess the quality of the virus and the production process.
- * p8.lines171-175. This is not an accurate statement. It is well known that non-targeting gRNAs outcompete gRNAs targeting non-essential genes by virtue of some form of DNA damage response. EDF2f shows virtually no difference between non-targeting and targeting of non-essential genes with qgRNAs. Depending on the number of copies of each of the target genes in HEK293 cells, the qgRNAs targeting non-essential genes would generate 8 dsDNA cuts (possibly more), yet this seems to have no impact on cell fitness? This statement is particularly problematic in the face of iPSC and organoid cultures, where it is clear that multiple cuts significantly impact cellular fitness, I did not find any data in the manuscript that accurately describes the

effects of making cuts at 1, 2, 3, or 4 target sites to provide some pause to researchers hoping to take advantage of this resource.

Reviewer #3 (Report for the authors (Required)):

In this study, the authors describe a method for assembling arrayed CRISPR libraries (a method they call APPEAL). These arrayed CRISPR libraries consist of thousands of gene constructs, each of which encodes four Cas9 sgRNAs targeting one single gene. Although arrayed CRISPR libraries are not new, the novelty of this method is that the authors have removed the most labor-intensive steps of the standard molecular cloning workflow and optimized it for large-scale library construction. Thus, the authors could use APPEAL to generate libraries targeting tens of thousands of genes for upregulation, downregulation, or editing. Large, arrayed libraries like these address several important shortcomings with existing CRISPR screening methodologies, as outlined by the authors.

To demonstrate the usefulness of these large, arrayed CRISPR libraries, the authors present three use-cases. In the first, the authors performed an arrayed CRISPR activation screen to identify transcription factors that control levels of the prion protein PrPc and its gene PRNP. The authors demonstrate that their arrayed library enabled identification of several regulating transcription factors that have not been identified in previous CRISPR screens.

Second, the authors performed a pooled CRISPR knockout screen to identify regulators of autophagy. The robust gene editing enabled by the arrayed library enabled identification of novel autophagy regulators.

Third, the authors performed an epigenetic silencing screen. They compared their arrayed four-sgRNA library with previously published two-sgRNA library and found that the four-sgRNA arrayed library performed slightly better.

Lastly, the authors confirmed previous findings that lentivirus production with pooled CRISPR libraries leads to much undesired lentiviral template switching, but that this problem can be circumvented by first producing lentivirus from each individual library construct separately and then pooling all lentiviruses. While the method can help solve the problem, it does increase the tediousness of the protocol and may limit the scale of the pooled library.

The authors present the findings clearly and with a logical narrative. Experiments have been rigorously performed and include detailed control experiments and discussions of alternative explanations. The Methods section is detailed and appears to enable other researchers to replicate the methodology on their own. We did not review the authors' R code for sgRNA selection.

Overall, we think the study provides a new and powerful addition to the CRISPR screening toolkit, and as such represents a methodological advance. By addressing some important shortcomings with current CRISPR screening protocols, the APPEAL method will enable researchers to address questions that have been intractable with previous protocols. Because CRISPR screening is powerful for uncovering new biological insights and many researchers are interested in screening, methodological advances like these are bound to be useful to the research community. We therefore think that this manuscript will be valuable to many scientists.

We are enthusiastic about the publication of this study in Nature Biomedical Engineering. We do think that the lentivirus template switching problem requires some action, but apart from that, the study can be published with minor revisions.

Major comments

1. The lentivirus template switching problem. The authors confirm previous findings that lentiviral template switching is a problem for pooled CRISPR library generation (e.g., lines 388-394). In the last section of the paper (lines 475-489), the authors find that this problem can be circumvented by producing the lentiviral library using separate wells for each construct instead of in a pooled fashion. That's great. But in the pooled CRISPR screen aimed at identifying autophagy regulators presented earlier in the paper (lines 374-436), they have not yet applied this workaround, and so their library suffers from template switching artifacts. That such template switching occurred was discovered by the authors because they used sequencing

primers that spanned the array's sgRNA2 and sgRNA3 (though not sgRNA1 and sgRNA4). In their sequencing results, the authors find evidence of abundant lentiviral template switching between sgRNA2 and sgRNA3 (~30%).

As the authors note on lines 479-480, if they observe 30% template switching between sgRNA2 and sgRNA3, there is likely even higher template switching across the entire qgRNA cassette, though the authors cannot observe it using their sequencing primers. When analyzing the autophagy screen, the authors deal with this problem by subsetting for "reads that aligned correctly and where both sgRNAs mapped to the same plasmid".

This is where things get a bit unclear. The authors can subset for reads that contain both the correct sgRNA2 and sgRNA3. This indeed filters out reads where template switching had occurred between sgRNA2 and sgRNA3. But this likely leaves in constructs where template switching had occurred in other parts of the qgRNA cassette (i.e., between sgRNA1 and sgRNA2 or between sgRNA3 and sgRNA4), which the authors could not detect. This artifact probably obfuscates the results a great deal, but the authors do not mention it at all.

If the authors hadn't come up with the workaround described in the last section (lines 475-489), I would say that this flaw would invalidate the authors' proposed use of their arrayed libraries for pooled screens. As it is now, the authors present the results from a flawed screening experiment and, in a separate section of the manuscript, propose a workaround, though they do not mention that the results from their earlier screen were probably invalid because they hadn't yet applied this workaround.

This is not necessarily disastrous for the paper but I do think the authors need to address it somehow. The most rigorous way would be to redo the autophagy screen with a corrected library. But because that is a big experiment, another way could be to openly state (in the relevant Results section) that the library is likely riddled with template-switching artifacts and that the results from the autophagy screen may not be reliable. Can the authors estimate quantitatively how big this problem is likely to be in the autophagy dataset? The authors could perhaps also move the last section (describing the workaround) to follow immediately after the section describing autophagy screen to make it clear that it is intended to correct the flaw in the autophagy screen.

Minor comments

2. The qgRNA construct uses four separate Pol. 3 promoters and tracrRNAs (e.g., lines 99-100). The authors provide three references here but do not specify exactly which sequences were used from which reference. Exactly which tracrRNAs were used, for example? The authors provide DNA sequences for the empty pYJA5 backbone and the backbone with the qgRNA insert in the Supplement, but these sequences are not annotated and are therefore difficult to understand. The sequences should be annotated (by coloring the letters and providing a legend for DNA features, similar to how they annotated the amplification and sequencing primers). Moreover, it is crucial that the authors upload these and other relevant DNA constructs to Addgene to enable researchers to use this method in their own labs.

3. The authors name their main two arrayed libraries T.spiezzo and T.gonfio (line 85). These names are quite unintuitive for someone who doesn't speak Italian. And even if the authors later explain that gonfio means "swelling up", this doesn't really make me see how the name relates to gene activation and repression. And where does the "T." in the names come from? The authors are free to name their libraries what they want but the feedback is that it reduces legibility and may make it difficult for readers to remember what library the authors are talking about.

4. The APPEAL workflow includes magnetic bead-based minipreps using "custom-made equipment" (line 134). What is this custom equipment? There is no mention of such equipment in the Methods section. If there is special equipment that enables or facilitates the workflow, the authors should include a figure with photos of the setup and instructions for how to build the custom equipment.

5. The authors state that lentivirus-mediated CRISPRa with the qgRNA constructs was "generally efficient, and its extent depended on the basal expression levels of the target genes (Extended Data Fig. 2j)" (line 186). However, Extended Data Fig. 2j shows activation levels ranging from ~2-fold to ~10,000-fold and does not show how activation levels are related to basal expression levels. Fold Activation is a tricky metric and can be related to basal expression levels but the authors should be more specific in their claim on line 186. They should state that activation levels ranged from X-Y and provide some evidence that fold change is related to basal expression level.

6. In the Methods section (“Transformation and bacterial storage” and “Magnetic bead-based 96-well plasmids miniprep”), the authors state that they added tetracycline to the bacterial culture medium. Why was this done? There is no mention of a tetracycline resistance gene anywhere in the paper.
7. I am sympathetic to referencing classical works of science and philosophy, but honestly the first sentence of the paper was almost unintelligible to me (line 47). May I suggest rewriting to improve legibility?
8. As a suggestion, I think it would be useful to include (in the Discussion section), how the APPEAL-generated libraries uniquely enabled each of the three use-cases presented in the Results section, compared to previous methodologies. For example, the PrPc experiment “could not be done with previous methods because of such and such but is uniquely enabled by our method because of such and such”.

Mon 24 Jun 2024

Decision on Article NBME-23-2583A

Dear Prof Aguzzi,

Thank you for your revised manuscript, "Quadruple-Guide Arrayed CRISPR for Indexed Human Genome-Wide Activation, Deletion, and Silencing", and for your patience in waiting for the feedback. I sent the revised manuscript to the original reviewers; yet unfortunately Reviewer #2 has not yet provided any feedback, and at this point I am not confident that they will.

Nevertheless, in view of the comments from the other two reviewers, who are happy with the latest version of the manuscript, and of our assessment of your replies to the earlier points by Reviewer #2, I am pleased to write that we shall be happy to publish the manuscript in *Nature Biomedical Engineering*, provided that the points specified in the attached instructions file are addressed.

When you are ready to submit the final version of your manuscript, please upload the files specified in the instructions file.

We encourage authors to take up transparent peer review. If you are eligible and opt in to transparent peer review, we will publish, as a single supplementary file, all the reviewer comments for all the versions of the manuscript, your rebuttal letters, and the editorial decision letters. **If you opt in to transparent peer review, in the attached file please tick the box 'I wish to participate in transparent peer review'; if you prefer not to, please tick 'I do NOT wish to participate in transparent peer review'**. In the interest of confidentiality, we allow redactions to the rebuttal letters and to the reviewer comments. If you are concerned about the release of confidential data, please indicate what specific information you would like to have removed; we cannot incorporate redactions for any other reasons. More information on transparent peer review is available.

Best wishes,

Pep

Pep Pàmies
Chief Editor, Nature Biomedical Engineering

P.S. Nature Portfolio journals encourage authors to share their step-by-step experimental protocols on a protocol-sharing platform of their choice. Nature Portfolio's Protocol Exchange is a free-to-use and open resource for protocols; protocols deposited in Protocol Exchange are citable and can be linked from the published article. More details can be found at www.nature.com/protocolexchange/about.

Reviewer #1 (Report for the authors (Required)):

Overall, the authors addressed the majority of the concerns raised by the reviewers and - in this reviewer's opinion - the paper is suitable for publication in *Nature Biomedical Engineering*. We believe that for this study to be impactful, accessibility of the quadruple gRNA libraries is critical. We appreciate the authors' efforts to share and distribute the resources generated in this study with the scientific community.

Reviewer #3 (Report for the authors (Required)):

The authors have addressed all our comments satisfactorily. We especially thank the authors for the rigorous explanation to our comment about the template-switching problem. We support the publication of the study in its current form, apart from one small detail:

Both we and Reviewer #1 asked questions about activation fold-change, which the authors addressed with new experiments. We thank the authors for this. Given the new data on the magnitude of target gene activation, we think it is misleading that the abstract says that activation efficiency was “~10,000-fold”. Sure, the authors saw 10,000-fold activation for one single gene, HBG1 (Extended Data Fig. 1a). But HBG1 (a hemoglobin subunit) is normally not expressed at all in HEK293 cells and can therefore easily be activated by tens of thousands-fold using a single gRNA (personal experience with HBG1 activation in HEK293 cells). Instead, judging by Fig. 1f, Extended Data Fig. 2a, and Extended Data Fig. 5f, the median activation level is more like 10-20-fold. If the authors want the abstract to mention the typical fold-change activation they achieve with their library, they should calculate the median (not mean because not normally distributed) fold change from all their CRISPRa experiments and present that number, and not “10,000”. We personally think fold-change can anyway be a misleading metric if presented on its own, and that we should move away from striving to present impressive-sounding fold-changes.

Nature Biomedical Engineering is a Transformative Journal. Authors may publish their research with us through the traditional subscription access route, or make their paper immediately open access through payment of an article-processing charge. More information about publication options is available.

You may need to take specific actions to comply with funder and institutional open-access mandates. If the work described in the accepted manuscript is supported by a funder that requires immediate open access (as outlined, for example, by Plan S) and your manuscript was originally submitted on or after January 1st 2021, then you will need to select the gold OA route. Authors selecting subscription publication will need to accept our standard licensing terms (including our self-archiving policies), and these will supersede any other terms that the author or any third party may assert apply to any version of the manuscript.

Rebuttal 1

Point-to-point response to reviewers' comments

Reviewer #1 (Report for the authors (Required)):

In this paper, Yin et al. constructed two genome-wide libraries, T. spiezzo and T. gonfio, targeting each human gene with four non-overlapping gRNAs. The authors provide extensive evidence to support the quality of the libraries and their performance in Crispr-mediated knockout and overexpression experiments. By targeting each gene with multiple gRNAs, T. spiezzo and T. gonfio improve the potency of gene perturbation (particularly in CRISPRa experiments), enhance the sensitivity of pooled CRISPR-screens, and reduce the library size. In addition, the authors confirm that T. gonfio can be efficiently used in gene silencing experiments using the Crispr-off machinery.

The new libraries developed in this paper show improved performance when compared to previously described libraries and are therefore a resource that can be used by the scientific community to perform high quality CRISPR screenings at lower costs. However, the methods used by the authors to design and generate the library are not revolutionary.

The paper is overall well presented, although in many instances it would be useful to have more details on the experimental design to easily understand the figures.

We appreciate the positive evaluation of our manuscript.

Detailed comments:

1. Resources generated in this paper are potentially useful for the community. However, it is not clear how these resources would be easily distributed or accessible and the conceptual advance or methodological novelty is limited.

Since >20 years, the Aguzzi lab is committed to sharing all its tools and reagents with the academic community. In that spirit, the plasmids including the backbone vector pYJA5 (Addgene, #217778) for cloning of quadruple sgRNA (qgRNA) vector, two non-targeting control plasmids of each library (Addgene, # 217779- 217782, which can be used as controls and provide the three constant regions for qgRNA assembly, and the CRISPRoff-mScarletI plasmid (Addgene, #217783) have now been deposited to Addgene and are available without restrictions for non-commercial purposes. This is now specified in the revised manuscript (Methods; DNA constructs section).

However, Addgene has declined to serve as a distribution hub for the arrayed T.gonfio and T.spiezzo collections (42,000 individual plasmids) because their maintenance would exceed Addgene's bandwidth. We therefore opted to distribute the tools directly. Recognizing that large-scale arrayed genome-perturbation screens require significant logistics, we grant access to the scientific community through four mechanisms:

- a) Direct distribution of the libraries to academic institutions wishing to conduct experiments locally. The cost of regenerating plasmids and producing lentiviruses is covered through joint grant applications. Thus far, three such

agreements have been stipulated with partner institutions, and several more are being prepared.

- b) We received a grant of the Swiss Universities association (CHF 500'000) which allows us to host scientists interested in performing arrayed screens in our laboratory. We made a public call for applications and selected the 10 most promising ones with the help of an independent Scientific Advisory Board. These projects will be executed within the next 3 years.
- c) Provisioning T.gonfio and T.spiezzo to academic core facilities. This approach is highly sustainable and extends access to all scientists belonging to the respective academic cluster. Crucially, core facilities can address the technical challenges of high-throughput screens (including the availability of complex liquid-handling equipment) in the most cost-efficient manner. We are currently negotiating agreements with several core facilities in Europe and the USA.
- d) The University of Zurich has provided seed funding for the establishment of an on-site core facility for genetic screening with our reagents. This facility is structured as a non-profit entity. Its use is not restricted to local scientists but will service interested parties worldwide.

These channels guarantee broad accessibility and reduce the cost of library maintenance, thereby benefiting the global scientific community.

We respectfully disagree with the reviewer's statement that the "conceptual advance or methodological novelty is limited" for the following reasons:

- **Conceptual advance: innovative high-throughput cloning.** Arrayed libraries require the generation of tens of thousands of individually collected plasmids. This is extremely labor-intensive and arguably the main reason for the dearth of such libraries. The APPEAL method (now ALPA cloning, according to the suggestion of the editor) addresses the technical challenge of generating large-scale arrayed individual plasmids, thus representing a conceptual and practical breakthrough.
- **Methodological novelty: expansion of the CRISPR screening territory.** Pooled CRISPR libraries (all sgRNA plasmids are pooled in 1-2 pools) are confined to selectable phenotypes. Arrayed libraries (each individual sgRNA plasmid is localized into a separate well of microplates) expand the screening space to phenotypes inaccessible to pooled CRISPR libraries, such as cell-cell interactions, morphological, and secretory phenotypes inter alia. There are some commercial CRISPR ablation libraries, but they are based on outdated, inefficient designs. Crucially, there are no arrayed CRISPR activation/silencing libraries available at present.
- **Methodological novelty: unprecedented targeting efficiency through qgRNA targeting with four nonoverlapping sgRNAs.** Furthermore, sgRNAs were designed to tolerate DNA polymorphisms among >10,000 individuals, which is crucial for use with primary human cells.

2. In figure 1, the performance of T. gonfio (f-g) and T. spiezzo (h-i) is compared to single gRNAs targeting very few genes (Figure 1 f-h). They also show data for other genes that

have been proven to be difficult to activate (g) or silence (i). The number of these genes is still pretty limited and in g-i a direct comparison with single gRNAs is lacking. Are these data sufficient to convincingly prove better performances of T. gonfio and T. spiezzo? This is important in light of data in Extended figure 5 c-d that show very variable silencing results obtained with T.spiezzo targeting EPCAM and CD44 in different cell types and with different delivery methods.

We have addressed these concerns as detailed below:

1. For gene activation, we had originally compared single guides with their qgRNA version in three cases. We have now expanded our comparisons to the 12 protein-coding genes that were studied in Figure 2 of Koner mann et al. Again, qgRNAs performed at least as well, and often better, than the best-performing single sgRNAs (Fig. 1f and Extended Data Fig. 2a). The activation differential was most pronounced for genes with low basal expression levels (Extended Data Fig. 2b). This is relevant because such genes are often non-responsive to single RNA guides.
2. To test if the synthetic assays translate well to real-life scenarios, we compared the performance of conventional vs qgRNAs in parallel autophagy screens using the T.spiezzo, Brunello, and Celecta libraries. Here we show a modified version of Fig. 5b in which we projected the fold-changes observed with T.spiezzo and Brunello, respectively. For the same known autophagy modifiers, Brunello achieved a change of approx. 2.5 log₂ (approx. 1.3-fold) whereas T.spiezzo achieved 6-8 log₂ (up to 3-fold).
3. Not only did T.spiezzo recognize known autophagy genes with higher signal/noise ratio than any of the 1sgRNA libraries, but it also identified many novel *bona fide* autophagy modifiers – which we validated as such using orthogonal techniques (Fig. 5 and Extended Data Fig. 6).
4. Finally, we compared the epigenetic silencing efficacy of T. gonfio and a dual-sgRNA library (CRISPRoff) in a genome-wide essentialome screen. T.gonfio showed slightly higher phenotypic scores and reduced variability than CRISPRoff (Fig. 6f-h), resulting in fewer false positives as documented by a receiver-operating characteristic analysis (Fig. 6i).

3. It seems to me that for KO experiments, T. spiezzo performs similarly to the best single gRNAs, while T. gonfio allows very high gene activation. Is that a general rule or is it just a case for the 3 targets tested? Already in the extended figure 2a-b, the difference between T. gonfio and the best gRNA is not as dramatic as figure 1f. Are more genes needed to conclude that T. gonfio reliably outperforms single gRNAs?

In Extended Figure 2a-b (now Extended Data Fig. 2d-e), we had analyzed the protein abundance of CD2, CD4, and CD200 on the surface of living cells by antibody staining after single sgRNAs vs qgRNAs activation. Surface protein expression can be affected

by many rate-limiting steps and is not directly comparable to Fig. 1f which reports qPCR results. However, we observed that even in these cases, qgRNAs reduced the variegation of expression among treated cells (Extended Data Fig. 2d-e) which was observed with single-guide activation, thereby improving the reliability of genetic screens.

4. What is the degree of activation for the genes in the transcription factor screen (at least for the hits listed in figure 4e)?

We have compared the activation of all 36 hits and *PRNP* (positive control) to nontargeting control qgRNA in lentivirally transduced cells (hits in Fig. 4e). As now shown in Extended Data Fig. 5f, the enhancement of expression was generally conspicuous (range: 1.47-fold to 310,906-fold).

Minor technical criticisms:

5. Figure 1b: for the single colony PCR it would be useful to have a schematic of the position of the primers (similarly to what shown in the bottom of extended figure 1) and the expected band size.

We have modified Fig. 1a as suggested.

6. For figure 1h, I found that this way of showing the data is very misleading, as it is not obvious that each dot represents the average of knockout for a single gRNA. I would rather the authors presented the data as in figure 1f or extended figure 2d.

Corrected (now Fig. 1g).

7. In the extended figure 2b, it seems unlikely that some of the single gRNA results are not significant.

Our intention was to compare the difference between single sgRNAs and qgRNAs. We have now updated all such comparisons (Fig. 1f, g and Extended Data Figure 2a, e).

8. Extended figure 5c-d, statistics are missing. In addition, the panel in figure c bottom on P24 is not discussed anywhere in the paper.

The statistics have now been calculated and incorporated (Extended Data Fig. 5c-d). A sentence was introduced to explain the use of p24 as a surrogate marker of viral titers (lines 335-336).

9. Lines 360-364: this paragraph is confusing if considering the statistics in figure 4h

We have now updated the paragraph. Please see lines 365-369 in the revised manuscript.

10. Figure 5g: it would be helpful to add a signal to visualize the cells (i.e. Dapi)

We have now used the cytosolic GFP signals to trace cell contours with dashed lines (Fig. 5g and Extended Data Fig. 6o).

11. Figure 5i: please add quantifications of the western blots as the conclusions described in the main text are not as obvious.

We have quantified the LC3II intensity of Fig. 5i, and the new data has been incorporated into the Extended Data Fig. 6p in the revised manuscript.

Reviewer #2 (Report for the authors (Required)):

Summary:

The authors construct genome-wide arrayed plasmid libraries for human gene ablation (T. Spiezzo), as well as activation/silencing (T. gonfio). Each plasmid encodes an array of 4 non-overlapping gRNAs (quadruple gRNAs or qgRNAs) whose expression is driven by different RNA polIII promoters. The APPEAL (Automated-liquied-Phase-Plasmid-assEmbly-And-cLoning) cloning method was developed to generate each construct in the arrayed libraries. Subsequent assessment was that both libraries were of high quality by various metrics; for example, 90% of wells were >75% correct for each of the 4 individual sgRNAs. The qgRNA plasmids from the libraries were benchmarked for cell surface markers in cell lines, iPSCs and kidney organoids. A number of different screens were next performed for functional discovery including: CRISPRa arrayed screen for TFs controlling expression of the PRNP gene product, CRISPRko pooled screen for regulation of the autophagy reporter SQSTM1, and a pooled CRISPR-off screen for epigenetic silencing. Validation efforts were also made, particularly on the results of the autophagy CRISPRko screen where 5 novel autophagy modifiers were validated using orthogonal methods.

We thank the reviewer for the positive evaluation of our manuscript and for appreciating our arrayed individual plasmid cloning method and the arrayed libraries.

Comments:

This study presents construction methods and SpCas9-based quadruple guide-RNA (qgRNA) libraries for lentiviral-based arrayed phenotypic screens. Overall, and despite the large number of existing CRISPRko/i/a libraries, this may prove to be a useful resource. The arrayed nature of the library will provide researchers with the option to miniaturize their assays in multi-well format with individual knockdowns. Furthermore, the quadruple nature of the system reduces the number of wells required for genome-wide screens by 4-fold. However, aside from the PrPc arrayed screen, the study is all predicated on simple phenotypes and pooled screens. In fact, the PrPc screen could arguably have been carried out as a pooled screen. The advantage of the arrayed format is that one can image the cells and examine single cell phenotypes in a multiplexed quantitative fashion. This missed opportunity would certainly enhance the value of this resource and provide some context to its usability in more complex assays.

We are currently performing complex screens based on single-cell phenotyping, and will report them in due time. A particularly exciting image-based screen aims at describing modifiers of synuclein phosphorylation occurring after exposure to exogenous synuclein fibers.

A rationale for performing the PrP^C screen in an arrayed format was that we avail of a miniaturized, sensitive high-throughput assay for the detection of PrP^C in cell lysates using fluorescence resonance energy transfer, which we used to perform a genome-wide siRNA screen (Novel regulators of PrP^C biosynthesis revealed by genome-wide RNA interference | PLOS Pathogens). This provided a foundation for assessing the performance of our T. gonfio library.

We agree that in principle the PrP^C screen could have been performed in a pooled format, we felt that the arrayed format would (1) massively increase the sensitivity, (2) provide a precise, quantitative readout, and (3) showcase the feasibility of large-scale, cell-lysate based arrayed screens which may not always be amenable to single-cell methods such as PerturbSeq or FACS-based screens.

Finally, the PrP^C screen yielded the surprising insight that arrayed screens, when performed with optimized assays (such as our FRET assay), are not much more laborious than pooled screens. This is because arrayed screens deliver hits that are typically robust and survive orthogonal validation, thereby reducing follow-up efforts.

Specific comments:

1. Is there any meaning behind naming of the libraries T. Spiezzo and T. gonfio? Seems arbitrary.

Many CRISPR libraries were jocularly given Italian names, with “Brunello” and “Dolcetto” referring to Italian wines. In that spirit, we ran a naming contest for our libraries on Twitter, and the current names were adjudicated as the best ones. “T.spiezzo” is derived from a famous sentence uttered by the fictional character Ivan Drago in the 1985 movie “Rocky IV”, meaning “I will break you into two pieces”, which in our view aptly symbolizes the gene-ablation power of our library. Analogously, “T.gonfio” means “I will make you swell”, which suggests strong activation of the target genes. The instant popularity of these two libraries, with already >20 labs receiving the full arrays or sublibraries thereof, appears to indicate that our naming convention was well received.

2. A number of features are built into pYJA5 (e.g. PB, PuroR-T2A-TagBFP, etc.) that are not used to their capacity. For example, PB transposition could introduce many cassettes into the same cell, followed by expression of the qgRNAs which could lead to substantial off-target effects as the concentration of gRNAs increases. It would be appropriate to comment whether these features were tested and function as expected.

Because of the significant effort of constructing >42'000 plasmids, we have engineered the vectors for maximal future versatility. The backbone features of our qgRNA vector were inherited from the lenti-PB vector designed by Metzakopian et al. We modified the

lenti-PB vector to be compatible with qgRNA cloning and named the new vector pYJA5 (Methods; DNA constructs). These features were validated in the previous study and confirmed in our hands as well. We did not measure how many copies of PB transposioned cassettes were integrated into the host cells and whether the effect of increased concentration of sgRNAs leads to substantial off-targets. We have added these points (lines 532-533 and 579-581) in Discussion.

3. Figure 1f. Given that the general perception in the field is that CRISPRa is highly gRNA-dependent, it is surprising that the authors observe no effects with any of the individual gRNAs yet a robust activation with the qgRNA. Could the authors comment on whether they have observed single gRNA CRISPRa effects in their system? Have they tested all pairs to measure synergy? I suppose the results of the shared subsequence experiment (EDF 3b,c) provides some answer to this but doesn't explain why no single gRNA works to activate the target genes they are testing.

We did observe a moderate degree of gene activation with single sgRNAs. This becomes evident if the activation is presented on a logarithmic scale. The source data will be provided to the journal. We retained the linear scale data presentation of the two genes to be consistent with that of CXCR4 and of the additional 12 genes which we have now tested (Question 2, reviewer #1) (Fig. 1f and Extended Data Figure 2a). We did not test all pairs of single sgRNAs to measure their synergy.

4. p3.line83. Define monospecific.

Deleted in the revised manuscript.

5. The methods for producing arrayed virus in 384-well plates are well described but it is difficult to fathom that this process is 100% efficient at generating lentivirus with titers $\sim 1e6$ IUs/mL. The methods (Methods, page 6) indicate that the percentage of TagBFP-positive infected cells were analyzed by flow cytometry to estimate titers, but there was no data to review to assess the quality of the virus and the production process.

Our quality control steps for viral production include assessment of plasmid transfection efficiency and lentiviral titration. A transfection efficiency $\geq 50\%$ (as assessed by TagBFP⁺ cells) typically resulted in acceptable lentiviral production, which we define as $\geq 90\%$ of wells reaching a titer of $\geq 10^6$ transduction units/ml (determined by flow cytometry in HEK293T cells). Across the entire T.gonfio library, these criteria were satisfied in 81.5% of plates. We have now incorporated these data into the revised manuscript (Extended Data Fig. 2j, lines 182-184, and Methods; Virus collection and Virus titration sections).

6. p8.lines171-175. This is not an accurate statement. It is well known that non-targeting gRNAs outcompete gRNAs targeting non-essential genes by virtue of some

form of DNA damage response. EDF2f shows virtually no difference between non-targeting and targeting of non-essential genes with qgRNAs. Depending on the number of copies of each of the target genes in HEK293 cells, the qgRNAs targeting non-essential genes would generate 8 dsDNA cuts (possibly more), yet this seems to have no impact on cell fitness? This statement is particularly problematic in the face of iPS and organoid cultures, where it is clear that multiple cuts significantly impact cellular fitness, I did not find any data in the manuscript that accurately describes the effects of making cuts at 1, 2, 3, or 4 target sites to provide some pause to researchers hoping to take advantage of this resource.

The sensitivity of cell fitness to DNA double-strand breaks may depend on the genomic region and on the cell line, as suggested by a systematic study (Álvarez et al.). Thus, we have added the following paragraph to the discussion (lines 544-548): “*Gene ablation by enzymatically active Cas9 can induce genotoxic double-strand DNA breaks. Although we did not observe qgRNA-associated genotoxicity in T.spiezzo-treated HEK293 cells, other studies suggest that toxicity can be cell-line and genomic-region dependent. In such contexts, T.gonfio-mediated epigenetic silencing may offer advantages over T.spiezzo-mediated gene ablation.*”

Reviewer #3 (Report for the authors (Required)):

In this study, the authors describe a method for assembling arrayed CRISPR libraries (a method they call APPEAL). These arrayed CRISPR libraries consist of thousands of gene constructs, each of which encodes four Cas9 sgRNAs targeting one single gene. Although arrayed CRISPRs libraries are not new, the novelty of this method is that the authors have removed the most labor-intensive steps of the standard molecular cloning workflow and optimized it for large-scale library construction. Thus, the authors could use APPEAL to generate libraries targeting tens of thousands of genes for upregulation, downregulation, or editing. Large, arrayed libraries like these address several important shortcomings with existing CRISPR screening methodologies, as outlined by the authors.

To demonstrate the usefulness of these large, arrayed CRISPR libraries, the authors present three use-cases. In the first, the authors performed an arrayed CRISPR activation screen to identify transcription factors that control levels of the prion protein PrPc and its gene PRNP. The authors demonstrate that their arrayed library enabled identification of several regulating transcription factors that have not been identified in previous CRISPR screens.

Second, the authors performed a pooled CRISPR knockout screen to identify regulators of autophagy. The robust gene editing enabled by the arrayed library enabled identification of novel autophagy regulators.

Third, the authors performed an epigenetic silencing screen. They compared their arrayed four-sgRNA library with previously published two-sgRNA library and found that the four-sgRNA arrayed library performed slightly better.

Lastly, the authors confirmed previous findings that lentivirus production with pooled CRISPR libraries leads to much undesired lentiviral template switching, but that this problem can be circumvented by first producing lentivirus from each individual library construct separately and then pooling all lentiviruses. While the method can help solve the problem, it does increase the tediousness of the protocol and may limit the scale of the pooled library.

The authors present the findings clearly and with a logical narrative. Experiments have been rigorously performed and include detailed control experiments and discussions of alternative explanations. The Methods section is detailed and appears to enable other researchers to replicate the methodology on their own. We did not review the authors' R code for sgRNA selection.

Overall, we think the study provides a new and powerful addition to the CRISPR screening toolkit, and as such represents a methodological advance. By addressing some important shortcomings with current CRISPR screening protocols, the APPEAL method will enable researchers to address questions that have been intractable with previous protocols. Because CRISPR screening is powerful for uncovering new biological insights and many researchers are interested in screening, methodological advances like these are bound to be useful to the research community. We therefore think that this manuscript will be valuable to many scientists.

We are enthusiastic about the publication of this study in Nature Biomedical Engineering. We do think that the lentivirus template switching problem requires some action, but apart from that, the study can be published with minor revisions.

We appreciate the reviewer's encouraging assessment of our manuscript.

Major comments

1. The lentivirus template switching problem. The authors confirm previous findings that lentiviral template switching is a problem for pooled CRISPR library generation (e.g., lines 388-394). In the last section of the paper (lines 475-489), the authors find that this problem can be circumvented by producing the lentiviral library using separate wells for each construct instead of in a pooled fashion. That's great. But in the pooled CRISPR screen aimed at identifying autophagy regulators presented earlier in the paper (lines 374-436), they have not yet applied this workaround, and so their library suffers from template switching artifacts.

That such template switching occurred was discovered by the authors because they used sequencing primers that spanned the array's sgRNA2 and sgRNA3 (though not sgRNA1 and sgRNA4). In their sequencing results, the authors find evidence of abundant lentiviral template switching between sgRNA2 and sgRNA3 (~30%).

As the authors note on lines 479-480, if they observe 30% template switching between sgRNA2 and sgRNA3, there is likely even higher template switching across the entire qgRNA cassette, though the authors cannot observe it using their sequencing primers.

When analyzing the autophagy screen, the authors deal with this problem by subsetting for “reads that aligned correctly and where both sgRNAs mapped to the same plasmid”.

This is where things get a bit unclear. The authors can subset for reads that contain both the correct sgRNA2 and sgRNA3. This indeed filters out reads where template switching had occurred between sgRNA2 and sgRNA3. But this likely leaves in constructs where template switching had occurred in other parts of the qgRNA cassette (i.e., between sgRNA1 and sgRNA2 or between sgRNA3 and sgRNA4), which the authors could not detect. This artifact probably obfuscates the results a great deal, but the authors do not mention it at all.

If the authors hadn't come up with the workaround described in the last section (lines 475-489), I would say that this flaw would invalidate the authors' proposed use of their arrayed libraries for pooled screens. As it is now, the authors present the results from a flawed screening experiment and, in a separate section of the manuscript, propose a workaround, though they do not mention that the results from their earlier screen were probably invalid because they hadn't yet applied this workaround.

This is not necessarily disastrous for the paper but I do think the authors need to address it somehow. The most rigorous way would be to redo the autophagy screen with a corrected library. But because that is a big experiment, another way could be to openly state (in the relevant Results section) that the library is likely riddled with template-switching artifacts and that the results from the autophagy screen may not be reliable. Can the authors estimate quantitatively how big this problem is likely to be in the autophagy dataset? The authors could perhaps also move the last section (describing the workaround) to follow immediately after the section describing autophagy screen to make it clear that it is intended to correct the flaw in the autophagy screen.

We thank the reviewer for thinking deeply about template switching in the context of our libraries. As our sequencing data show, template switching occurs at a very high frequency. However, its impact on screen performance seems to be limited, as detailed below:

When scoring for rare phenotypes, template switching is unlikely to invalidate screens. This is because that the template switches are random and involve different genes in each cell, most of which will not produce a phenotype and, consequently, won't be enriched in the screen. Our screens utilized ≥ 300 cells per qgRNA. Template switching will cause hundreds of random genes to be targeted in addition to those predicted by their barcode. However, because of their randomness and their low likelihood to modify the phenotype, they will be mostly averaged out. This introduces noise and may reduce the sensitivity of the screen, yet we found that it will not necessarily change the ranking of bona fide hits.

As an example, let's imagine a synthetic-lethality screen in which 70% of all guides are template-switched. If synthetic-lethal genes represent only a small segment of the whole genome, most switches will have no phenotypic effect and will not result in false-positive hits. Instead, intact (i.e. unswitched) vectors will exert their full effect onto cells and result in identifiable hits. This will forcefully bias the screen towards true

positives.

This theoretical example is corroborated by our experimental evidence Autophagy is a highly regulated process: the number of core autophagy genes ($n = 185$) is much smaller than that of the core essential genes ($n = 1'246$). This may be one reason why the autophagy screen was so successful despite our switching-prone “prepooling” strategy.

We conclude that postpooled libraries are necessary when (1) cell numbers are critically small, (2) when no drastic selective pressure can be applied, and (3) when genes with small effects must be identified. Conversely, prepooled T.spiezzo and T.gonfio libraries (which are much easier to produce) are highly effective when a high coverage of qgRNAs is possible and/or when the phenotypes are drastic and can be strongly selected (e.g. essentialome screens).

Our “prepooling” and “postpooling” head-to-head essentialome screen demonstrates a minimal effect of template switching on hits identification. My (AA) own prediction was that postpooling would turn out to be much more powerful than prepooling, precisely because it completely eliminates any template switches. We indeed observed a higher phenotypic scores of the essential genes, however, the ranking of the hits largely remains (Fig. 6k, l). Thus, in absolute numbers, the complete suppression of template switching yielded a much lower advantage than I (AA) had predicted. Whenever confirmatory screens are laborious and expensive, this improvement may prove extremely useful. However, it’s important to note that this favorable result may primarily apply to screens in which a less selective pressure is applied.

The results of the autophagy screen supports the real-life performance improvement of the T.spiezzo library. We disagree with the reviewer’s conclusion that we “do not mention that the results from [our] earlier screen were probably invalid because [we] hadn’t yet applied this workaround”. The screen not only identified recognized known autophagy genes with higher signal/noise ratio than any of the 1-sgRNA libraries tested in parallel, but it also identified many *bona fide* autophagy modifiers which we confirmed with orthogonal assays (Fig. 5). Although we agree that the use of the postpooled library may have yielded an even higher performance, we contend that the potency of the qgRNA approach renders the prepooled library a useful and sensitive tool. We expect that the true advantage of postpooling will become evident over time, as many laboratories are using the postpooled libraries in a variety of screening scenarios.

Thus, per the suggestion of the reviewer, we have stated in Results (lines 399-403) that “*This method does not detect template switches occurring in sgRNA1 and sgRNA4. However, the success of the screens reported here suggests that the randomness of the switches and the high number of targeted cells (300 cells/qgRNA) attenuates the impact of such switches. Thus, the sgRNA2-sgRNA3 sequencing strategy provides an acceptable compromise between quality assurance and cost-effectiveness.*” Also, we added a chapter (lines 613-623) discussing the effect of lentiviral template switch artifact in the Discussion.

We decided to not move the last section to follow immediately the autophagy screen section because the last section addresses the issue of the T.gonfio essentialome screen and provides a quantitative measurement of the impact of lentiviral template switching on pooled CRISPR screens.

Minor comments

2. The qgRNA construct uses four separate Pol. 3 promoters and tracrRNAs (e.g., lines 99-100). The authors provide three references here but do not specify exactly which sequences were used from which reference. Exactly which tracrRNAs were used, for example? The authors provide DNA sequences for the empty pYJA5 backbone and the backbone with the qgRNA insert in the Supplement, but these sequences are not annotated and are therefore difficult to understand. The sequences should be annotated (by coloring the letters and providing a legend for DNA features, similar to how they annotated the amplification and sequencing primers). Moreover, it is crucial that the authors upload these and other relevant DNA constructs to Addgene to enable researchers to use this method in their own labs.

We have added the referenced tracrRNAs sequences in the Extended Data Fig. 1a. We have also annotated the DNA sequences in the Supplement Information section in the revised manuscript. We have deposited these vectors to Addgene. See also our response to Question 1 of reviewer 1, and Methods, DNA constructs section.

3. The authors name their main two arrayed libraries T.spiezzo and T.gonfio (line 85). These names are quite unintuitive for someone who doesn't speak Italian. And even if the authors later explain that gonfio means "swelling up", this doesn't really make me see how the name relates to gene activation and repression. And where does the "T." in the names come from? The authors are free to name their libraries what they want but the feedback is that it reduces legibility and may make it difficult for readers to remember what library the authors are talking about.

Many CRISPR libraries were jocularly given Italian names, with Brunello and Dolcetto referring to Italian wines. In that spirit, we ran a naming contest for our libraries on Twitter, and the current names were adjudicated as the best ones. Their rising popularity, with >20 labs requesting the full arrays or sublibraries thereof, appears to indicate that our naming convention was well received.

4. The APPEAL workflow includes magnetic bead-based minipreps using "custom-made equipment" (line 134). What is this custom equipment? There is no mention of such equipment in the Methods section. If there is special equipment that enables or facilitates the workflow, the authors should include a figure with photos of the setup and instructions for how to build the custom equipment.

The equipment is a magnetic stand designed in-house and produced by 3D printing. It allows for efficient separation of plasmid-bound magnetic beads from the liquid mixture during the plasmid preparation procedure. We have now incorporated this information in the Methods; Magnetic bead-based 96-well plasmids miniprep section.

5. The authors state that lentivirus-mediated CRISPRa with the qgRNA constructs was “generally efficient, and its extent depended on the basal expression levels of the target genes (Extended Data Fig. 2j)” (line 186). However, Extended Data Fig. 2j shows activation levels ranging from ~2-fold to ~10,000-fold and does not show how activation levels are related to basal expression levels. Fold Activation is a tricky metric and can be related to basal expression levels but the authors should be more specific in their claim on line 186. They should state that activation levels ranged from X-Y and provide some evidence that fold change is related to basal expression level.

We now show a correlation analysis in Extended Data Fig. 2b (Question 2; reviewer #1) and have updated the sentence in question to “We observed conspicuous gene activation of all tested genes ranging from ~2-fold to ~10,000-fold” (lines 189-190).

6. In the Methods section (“Transformation and bacterial storage” and “Magnetic bead-based 96-well plasmids miniprep”), the authors state that they added tetracycline to the bacterial culture medium. Why was this done? There is no mention of a tetracycline resistance gene anywhere in the paper.

Our competent *E. coli* cells NEB Stable (C3040I, New England Biolabs) express tetracycline resistance. Thus, double antibiotic selection (trimethoprim and tetracycline, through the qgRNA vector and the host bacteria, respectively) was utilized to eliminate potential contaminations during the cloning procedure. This is now explained in Methods; Transformation and bacterial storage section.

7. I am sympathetic to referencing classical works of science and philosophy, but honestly the first sentence of the paper was almost unintelligible to me (line 47). May I suggest rewriting to improve legibility?

Changed to "*Genetic screens are well-established tools of biomedical research*" (Line 46).

8. As a suggestion, I think it would be useful to include (in the Discussion section), how the APPEAL-generated libraries uniquely enabled each of the three use-cases presented in the Results section, compared to previous methodologies. For example, the PrPc experiment "could not be done with previous methods because of such and such but is uniquely enabled by our method because of such and such".

We would be delighted to expand on the advantages of arrayed-screen technologies. However, our manuscript is already quite long, and we have pointed out such advantages at various points in the manuscript. Therefore, we would respectfully propose that we abstain from the specific discussion suggested by the reviewer, and instead let the data (including those that are being generated by colleagues who have received our libraries) speak for themselves.